# The Utrecht Finite Volume Ice-Sheet Model (UFEMISM version 2.0) – part 1: description and idealised experiments

Constantijn J. Berends[1], Victor Azizi[2], Jorge A. Bernales[1], Roderik S. W. van de Wal[1,3]

[1]Institute for Marine and Atmospheric research Utrecht, Utrecht University, Utrecht, the Netherlands
[2]Netherlands eScience Center, Amsterdam, the Netherlands
[3]Faculty of Geosciences, Department of Physical Geography, Utrecht University, Utrecht, the Netherlands

*Correspondence to*: Constantijn J. Berends (c.j.berends@uu.nl)

**Abstract.** Projecting the anthropogenic mass loss of the Greenland and Antarctic ice sheets requires models that can accurately describe the physics of flowing ice, and its interactions with the atmosphere, the ocean, and the solid Earth. As the uncertainty
in many of these processes can only be explored by running large numbers of simulations to sample the phase-space of possible physical parameters, the computational efficiency and user-friendliness of such a model are just as relevant to its applicability as is its physical accuracy. Here, we present and verify version 2.0 of the Utrecht Finite Volume Ice-Sheet Model (UFEMISM). UFEMISM is a state-of-the-art finite-volume model which applies an adaptive grid in both space and time. Since the first version was published two years ago, v2.0 has added more accurate approximations to the Stokes flow, more sliding laws,
different schemes for calculating the ice thickness rates of change, a more numerically stable time-stepping scheme, more flexible and powerful mesh generation code, and a more generally applicable discretisation scheme. The parallelisation scheme has changed from a shared-memory architecture to distributed memory, enabling the user to utilise more computational resources. The version control system (git) includes automated unit tests and benchmark experiments, to aid with model development, as well as automated installation of the required libraries, improving both user comfort and reproducibility of
results. The in/output (I/O) now follows the NetCDF-4 standard, including automated remapping between regular grids and irregular meshes, reducing user workload for pre- and post-processing. These additions and improvements make UFEMISM v2.0 a powerful, flexible ice-sheet model, that can be used for long palaeoglaciological applications, as well as large ensemble simulations for future projections of ice-sheet retreat, and which is ready to be used for coupling within Earth system models.

## 1 Introduction

One of the most worrisome, and at the same time most uncertain, possible long-term consequences of anthropogenic climate change is mass loss of the Greenland and Antarctic ice sheets, leading to global sea-level rise (Oppenheimer et al., 2019; van de Wal et al., 2019; Fox-Kemper et al., 2021). Projections for the Antarctic contribution to sea-level rise in 2100 under RCP8.5, which were studied in the Ice-Sheet Model Intercomparison for CMIP6 (ISMIP6), range from −2.5 cm to +17 cm (Seroussi et al., 2020), with a possible high-end value of +59 cm (van de Wal et al., 2022) and consequently much more on longer time
scales. Part of this large uncertainty stems from poorly constrained physical properties and processes in the Antarctic ice sheet

system, including subglacial conditions (e.g. Kazmierczak et al., 2022; Berends et al., 2023a), basal sliding (Sun et al., 2020), interactions between the ice shelf and the ocean in the sub-shelf cavity (e.g. Burgard et al., 2022; Berends et al., 2023b), calving (e.g. Crawford et al., 2021) and ice-dynamical processes (e.g. Rückamp et al., 2022). However, even in idealised experiments where all these quantities are known perfectly, different ice-sheet models can predict rates of sea-level rise that differ by a

factor of three (Cornford et al., 2020). This represents the uncertainty arising from the numerical models themselves, which disagree on how to translate the physical equations to computer code. These model differences include the way the momentum balance (typically represented by the Stokes equations) is approximated, the choice of grid, the numerical treatment of discontinuities in basal friction and melt rates at the grounding line, and the way the model is initialised. Sampling both this model-intrinsic uncertainty, and the uncertainty in the physical properties and processes of the actual ice sheet, require ice-

sheet models that have the computational power, and the flexibility, to perform large numbers of simulations, at an adequate resolution to capture these processes. To meet this challenge, many research groups working on ice-sheet modelling have recently directed their efforts at creating new, more powerful ice-sheet models (e.g. Pattyn, 2017; Hoffman et al., 2018; Quiquet et al., 2018; Lipscomb et al., 2019; Robinson et al., 2020; Berends et al., 2022).

Here, we present version 2.0 of the Utrecht Finite Volume Ice-Sheet Model (UFEMISM). The main distinguishing feature of the model is its dynamic adaptive mesh. This approach was pioneered by Durand et al. (2009) and Gladstone et al. (2010), and has since been applied in BISICLES (Cornford et al., 2013), ISSM (dos Santos et al., 2019), and (in glacier-scale applications) in Elmer/ice (Todd et al., 2018). This structure allows the model to resolve the grounding line at high (< 5 km) resolutions during multi-millennial simulations. Since the publication of v1.0, many new features have been added to UFEMISM, and

many existing features have been improved in terms of power, flexibility, and user-friendliness. In Sect. 2, we provide the physical equations for ice flow that are solved by the model. This includes several approximations to the Stokes equations for the momentum balance (Sect. 2.2), several sliding laws (Sect. 2.3), a new numerical scheme for treating basal friction at the grounding line (Sect. 2.4), different temporal discretisation schemes to calculate the ice geometry rates of change (Sect. 2.5), and a new adaptive time-stepping scheme (Sect. 2.6). In Sect. 3, we describe several improvements that were made to the

model code. This includes a change from a shared-memory to a distributed-memory implementation (Sect. 3.1), and a thoroughly reworked I/O module that now follows the NetCDF-4 standard (Unidata, 2023) and is much more flexible and user-friendly (Sect. 3.2). It also includes a version control system that includes automated unit tests and benchmark experiments to aid in developing robust code, and automated installation of external libraries to improve user-friendliness and reproducibility of results (Sect. 3.3). In Sect. 4, we present results of a number of idealised-geometry experiments to verify the

new model physics and numerics.

This paper, part 1, focuses on the basic mathematics and physics of the model, and their verification in idealised benchmark experiments. Part 2, which is submitted for review and publication separately (Bernales et al, *in prep.*), focuses on model additions required for the application of UFEMISM to realistic ice sheets such as those in Greenland and Antarctica. It includes

descriptions of the routines for inverting for subglacial bed roughness and for ocean temperatures in shelf cavities, different sub-shelf melt parameterisations, initialisation approaches, and future projections of mass loss.

## 2 Model description

### 2.1 General

UFEMISM is a large-scale ice-sheet model. It solves different approximations of the Stokes equations to calculate the flow
velocities of the ice. These are combined with the ice accumulation/loss rates at the surface, basal, and lateral boundaries of the ice sheet to find the thinning/thickening rates of the ice, which are integrated through time to find the evolution of the ice sheet. Note that hereafter, we will refer to UFEMISM version 1.0 as "v1.0", and to the version 2.0 presented here as "v2.0".

The main distinguishing feature of UFEMISM compared to many other ice-sheet models is the use of a dynamic adaptive grid.
The two-dimensional plane on which the model operates is discretised as an irregular triangular mesh, an example of which is shown in Fig. 1.

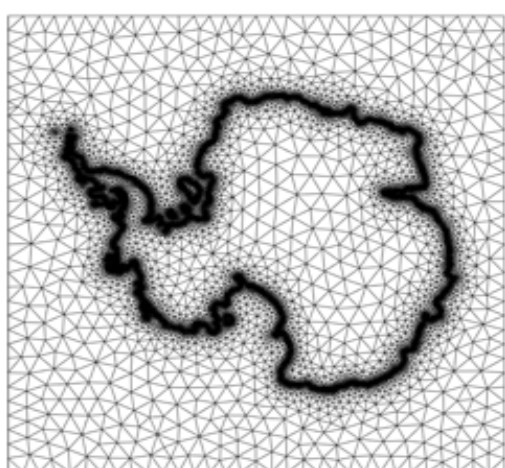

**Fig. 1: A demo mesh generated by UFEMISM for the Antarctic ice sheet, using a 10 km grounding-line resolution, and up to 200 km for the ice-sheet interior and the open ocean.**

Earlier research in ice-sheet modelling has shown that the accuracy of a numerical model is particularly sensitive to the spatial resolution of the grid around the grounding line (Durand et al., 2009; Gladstone et al., 2012; Pattyn et al., 2012). There, the discontinuous basal friction, which is non-zero underneath the grounded ice but zero underneath the floating ice, causes strong gradients in englacial stresses and therefore the ice geometry. Although different solutions have been presented in earlier literature to produce accurate results at coarser resolutions (see Sect. 2.4), the resolutions required at the grounding line are
still much higher than those needed in the slow-moving interior of the ice sheet, in order to achieve the same level of accuracy in the ice thickness evolution. As the demand for both the temporal coverage and number of ice-sheet simulations increases,

computational efficiency becomes a more important property of ice-sheet models. Using a uniform high resolution over the entire ice sheet, while it is only needed in the relatively small area around the grounding line, is therefore undesirable. UFEMISM solves this problem by using a mesh that has a high resolution only where needed, and a low resolution where

possible. This is the "adaptive" part of the mesh. However, as the ice-sheet geometry changes over the course of a model simulation, the location of the grounding line changes as well. This means that, after a while, the grounding line might no longer be located within the high-resolution area of the mesh. A possible solution would be to use a mesh with a high resolution over a wider area, enveloping the expected future migration of the grounding line. While this is a feasible approach for century-scale simulations, doing this for the multi-millennial applications for which UFEMISM is also intended would mean creating

a mesh with a very large high-resolution area, thus offsetting the benefits of the adaptive mesh. Instead, UFEMISM periodically checks the mesh fitness to the modelled ice-sheet geometry and, if needed, automatically creates a new mesh that conforms to the new ice-sheet geometry (with a high-resolution area around the new grounding-line position), remapping the model data from the old mesh to the new one. This is the "dynamic" part of the mesh. Berends et al. (2021) showed that this approach results in good computational performance, with no significant loss of accuracy.

While the general principles of the dynamic adaptive mesh have not changed significantly in v2.0 with respect to v1.0, the way these principles are implemented has changed in several ways. The new mesh generation code, the scheme used to discretise the partial differential equations of the model on the mesh, and the scheme used to remap data from one mesh to another, are presented in Appendices A, B, and C, respectively.

## 2.2 Momentum balance

UFEMISM v2.0 includes solvers for several different approximations to the Stokes equations, which neglect increasingly more terms in these equations. Of these approximations, the Blatter-Pattyn approximation (BPA; Blatter, 1995; Pattyn, 2003), which is described in Sect. 2.2.1, neglects the fewest terms. The depth-integrated viscosity approximation (DIVA; Goldberg, 2011; Sect. 2.2.2), the shallow shelf approximation (SSA; Morland, 1987; Sect. 2.2.3), the shallow ice approximation (SIA; Morland and Johnson, 1980; Sect. 2.2.4), and the hybrid SIA/SSA (Bueler and Brown, 2009; Sect. 2.2.5) can all be derived by neglecting

more and more terms. For a comprehensive description of the Stokes equations and a derivation of the different approximations, we recommend reading Greve and Blatter (2009).

**Table 1: Model symbols, units, and default values where applicable**

| Symbol | Description | Units | Value |
|--------|-------------|-------|-------|
| $A$ | Glen's flow law factor | $Pa^{-n}\ yr^{-1}$ | |
| $b$ | Bedrock elevation | m | |
| $\beta$ | Basal friction coefficient | $Pa\ m^{-1}\ yr$ | |

| | | | |
|---|---|---|---|
| $\dot{\varepsilon}_e$ | Effective strain rate | yr$^{-1}$ | |
| $g$ | Gravitational acceleration | m s$^{-2}$ | 9.81 |
| $H$ | Ice thickness | m | |
| $\eta$ | Effective viscosity | Pa yr | |
| $n$ | Glen's flow law exponent | | 3 |
| $\rho$ | Density of ice | kg m$^{-3}$ | 910 |
| $s$ | Surface elevation | m | |
| $u$ | Horizontal ice velocity vector | m yr$^{-1}$ | |
| $u$ | Horizontal ice velocity in x-direction | m yr$^{-1}$ | |
| $v$ | Horizontal ice velocity in y-direction | m yr$^{-1}$ | |
| $w$ | Vertical ice velocity | m yr$^{-1}$ | |
| $\zeta$ | Scaled vertical coordinate | | $0 - 1$ |

### 2.2.1 Blatter-Pattyn Approximation

The BPA arises from the Stokes equations by assuming hydrostatic equilibrium and neglecting the stresses arising from horizontal variations in the vertical velocity (i.e. $\frac{\partial w}{\partial x} \ll \frac{\partial u}{\partial z}, \frac{\partial w}{\partial y} \ll \frac{\partial v}{\partial z}$; Pattyn, 2003). This means that the pressure $p$ and the vertical velocity $w$ disappear as degrees of freedom from the momentum balance, so that only the horizontal velocities $u, v$ remain to be solved for. The BPA produces ice velocities that are generally very close to those from the Stokes equations (Pattyn et al., 2008), so that it is generally able to describe the large-scale evolution of continental ice-sheet-shelf systems such

as the Antarctic Ice Sheet. Deviations from the full Stokes solution are more noticeable in e.g. thermo-mechanically coupled problems of ice-streams (Schoof and Mantelli, 2021), advection problems of tracers (Jouvet et al., 2021) and flow at ridges and domes (Seddik et al., 2011). While less computationally expensive to solve than the Stokes equations, the BPA is still significantly slower than the DIVA or the hybrid SIA/SSA, owing to the fact that, where those approximations either parameterise or neglect vertical variations in horizontal velocities or strain rates, the BPA solves for such variations explicitly.

This requires the model to discretise the vertical dimension as well, whereas the DIVA and the hybrid SIA/SSA operate in the two-dimensional plane, yielding a system of linear equations that is larger by a factor of the number of vertical layers in the model.

The set of partial different equations that must be solved in order to find the 3-D horizontal ice velocities $u, v$ reads:


$$\frac{\partial}{\partial x}\left[2\eta\left(2\frac{\partial u}{\partial x}+\frac{\partial v}{\partial y}\right)\right]+\frac{\partial}{\partial y}\left[\eta\left(\frac{\partial u}{\partial y}+\frac{\partial v}{\partial x}\right)\right]+\frac{\partial}{\partial z}\left[\eta\frac{\partial u}{\partial z}\right]=\rho g\frac{\partial s}{\partial x}, \tag{1a}$$

$$\frac{\partial}{\partial y}\left[2\eta\left(2\frac{\partial v}{\partial y}+\frac{\partial u}{\partial x}\right)\right]+\frac{\partial}{\partial x}\left[\eta\left(\frac{\partial v}{\partial x}+\frac{\partial u}{\partial y}\right)\right]+\frac{\partial}{\partial z}\left[\eta\frac{\partial v}{\partial z}\right]=\rho g\frac{\partial s}{\partial y}. \tag{1b}$$

The effective viscosity $\eta$ is related to the effective strain rate $\dot{\varepsilon}$ by Glen's flow law (Glen, 1955):

$$\eta = \frac{1}{2} A^{-1/n} \dot{\varepsilon}_e^{\frac{1-n}{n}}. \tag{2}$$

The flow factor can be set to a uniform fixed value (as is done in the idealised experiments presented here), or can be calculated from the ice temperature, following the Arrhenius-type relation provided in Berends et al. (2021).

The effective strain rate $\dot{\varepsilon}_e$ is given by:

$$\dot{\varepsilon}_e = \left[ \left( \frac{\partial u}{\partial x} \right)^2 + \left( \frac{\partial v}{\partial y} \right)^2 + \frac{\partial u}{\partial x}\frac{\partial v}{\partial y} + \frac{1}{4}\left( \frac{\partial u}{\partial y} + \frac{\partial v}{\partial x} \right)^2 + \frac{1}{4}\left( \frac{\partial u}{\partial z} \right)^2 + \frac{1}{4}\left( \frac{\partial v}{\partial z} \right)^2 \right]^{1/2}. \tag{3}$$

At the ice surface, the zero-stress boundary condition reads:

$$2\frac{\partial s}{\partial x}\left( 2\frac{\partial u}{\partial x} + \frac{\partial v}{\partial y} \right) + \frac{\partial s}{\partial y}\left( \frac{\partial u}{\partial y} + \frac{\partial v}{\partial x} \right) - \frac{\partial u}{\partial z} = 0, \tag{4a}$$

$$2\frac{\partial s}{\partial y}\left( 2\frac{\partial v}{\partial y} + \frac{\partial u}{\partial x} \right) + \frac{\partial s}{\partial x}\left( \frac{\partial v}{\partial x} + \frac{\partial u}{\partial y} \right) - \frac{\partial v}{\partial z} = 0. \tag{4b}$$

A similar dynamical boundary condition at the ice base includes a basal friction term:

$$2\frac{\partial b}{\partial x}\left( 2\frac{\partial u}{\partial x} + \frac{\partial v}{\partial y} \right) + \frac{\partial b}{\partial y}\left( \frac{\partial u}{\partial y} + \frac{\partial v}{\partial x} \right) - \frac{\partial u}{\partial z} + \frac{\beta}{\eta} u = 0, \tag{5a}$$

$$2\frac{\partial b}{\partial y}\left( 2\frac{\partial v}{\partial y} + \frac{\partial u}{\partial x} \right) + \frac{\partial b}{\partial x}\left( \frac{\partial v}{\partial x} + \frac{\partial u}{\partial y} \right) - \frac{\partial v}{\partial z} + \frac{\beta}{\eta} v = 0. \tag{5b}$$

UFEMISM currently does not include a stress boundary condition at the ice front for any of the momentum balance approximations. Instead, it uses the "infinite slab" approach, where the momentum balance is solved on the entire grid, including ice-free cells (which the solvers assumes are covered by a very thin [0.1 m by default] layer of ice), and a simple Neumann boundary condition is applied at the domain boundary. While we have not (yet) tested this in UFEMISM, earlier experiments with IMAU-ICE (Berends et al., 2022) showed that the effect of this approach on the velocity solution is generally very small. It should also be noted that this approach is only used in the momentum balance; when solving for conservation of mass, the model does account for ice-free vertices, so that a calving front is explicitly included.

In order to solve the BPA, the vertical dimension must be discretised as well. This is not straightforward, as the surface and base of the ice are generally not flat, and evolve over time, so that these surfaces will generally move in between grid points in the vertical. In UFEMISM, as in most other ice-sheet models that solve a three-dimensional version of the momentum balance, this problem is solved by introducing a terrain-following coordinate transformation, which is described in Appendix D.

### 2.2.2 Depth-integrated viscosity approximation

The DIVA, which is the default option for the momentum balance approximation in v2.0, arises by neglecting the stresses that arise from vertical variations in the horizontal strain rates in the BPA (i.e. $\frac{\partial}{\partial z}\left( \frac{\partial u}{\partial x}, \frac{\partial u}{\partial y}, \frac{\partial v}{\partial x}, \frac{\partial v}{\partial y} \right) \approx 0$), and integrating the resulting equations vertically. This means that, whereas the BPA is solved in three dimensions, the DIVA operates in the two-dimensional plane, greatly reducing the computational expense of solving it. In the Ice-Sheet Model Intercomparison Project

for Higher-Order Models (ISMIP-HOM; Pattyn et al., 2008) experiments, the DIVA produces velocities that agree well with the Stokes solution down to horizontal scales for basal topographical features of about 20 km (Berends et al., 2022; Robinson et al., 2022; this study, Sect. 4.1).


The partial differential equations of the DIVA read:

$$\frac{\partial}{\partial x}\left[2\eta H\left(2\frac{\partial u}{\partial x}+\frac{\partial v}{\partial y}\right)\right]+\frac{\partial}{\partial y}\left[\eta H\left(\frac{\partial u}{\partial y}+\frac{\partial v}{\partial x}\right)\right]-\beta_{eff}u=\rho gH\frac{\partial s}{\partial x}, \tag{6a}$$

$$\frac{\partial}{\partial y}\left[2\eta H\left(2\frac{\partial v}{\partial y}+\frac{\partial u}{\partial x}\right)\right]+\frac{\partial}{\partial x}\left[\eta H\left(\frac{\partial v}{\partial x}+\frac{\partial u}{\partial y}\right)\right]-\beta_{eff}v=\rho gH\frac{\partial s}{\partial y}. \tag{6b}$$

Here, $\beta_{eff}$ is a term describing both basal friction and vertical shear stress:

$$\beta_{eff}=\frac{\beta}{1+\beta F_2}. \tag{7}$$

The integral term $F_2$, which can be thought of as a (scaled) depth-integral of the inverse viscosity, is defined as:

$$F_n=\int_b^h \frac{1}{\eta}\left(\frac{s-z}{H}\right)^n dz. \tag{8}$$

Note that, in Eq. 7, $n=2$; Eq. 8 lists the general form because elsewhere in the DIVA, $F_1$ appears as well. A comprehensive derivation of these and other required equations, including a step-by-step approach for how to solve them numerically, can be

found in Lipscomb et al. (2019).

While the mathematical derivation is too cumbersome to include here, it can be shown that, in the absence of horizontal strain (i.e. $\frac{\partial u}{\partial x}, \frac{\partial u}{\partial y}, \frac{\partial v}{\partial x}, \frac{\partial v}{\partial y}=0$), the DIVA is identical to the SIA. In a preliminary experiment, we used the DIVA to perform the moving-margin experiment from EISMINT-1 (Huybrechts et al., 1996), which describes a roughly Greenland-sized, idealised, circular, polythermal (though not thermomechanically coupled) ice sheet lying on a flat bed, achieving a steady state through

a simple, spatially variable mass balance. The resulting ice sheet, which is dominated by vertical shearing, was nearly identical to that produced by the SIA, with only a few meters difference in ice thickness, concentrated near the ice divide and the ice margin.

A more practical advantage of the DIVA that was previously pointed out by Robinson et al. (2022) is that the system of linear equations that must be solved (Eq. 6) is almost identical to that of the SSA (Eq. 9). Ice-sheet models that already contain code

to solve the SSA can therefore be altered to solve the DIVA instead with relatively little effort, including only a few simple calculations to evaluate Eq. 7, altering the friction term that enters into the system of linear equations.

### 2.2.3 Shallow shelf approximation

The SSA arises by neglecting all vertical variations in the BPA (i.e. $\frac{\partial u}{\partial z}, \frac{\partial v}{\partial z}\approx 0$), leaving only the membrane stresses, and then vertically integrating the result. This is generally accepted to be a valid approximation in areas of negligible basal shear stress,

such as ice shelves, as well as well-lubricated, fast-flowing ice streams.

The partial differential equations of the SSA read:

$$\frac{\partial}{\partial x}\left[2\eta H\left(2\frac{\partial u}{\partial x}+\frac{\partial v}{\partial y}\right)\right]+\frac{\partial}{\partial y}\left[\eta H\left(\frac{\partial u}{\partial y}+\frac{\partial v}{\partial x}\right)\right]-\beta u=\rho g H\frac{\partial s}{\partial x}, \tag{9a}$$

$$\frac{\partial}{\partial y}\left[2\eta H\left(2\frac{\partial v}{\partial y}+\frac{\partial u}{\partial x}\right)\right]+\frac{\partial}{\partial x}\left[\eta H\left(\frac{\partial v}{\partial x}+\frac{\partial u}{\partial y}\right)\right]-\beta v=\rho g H\frac{\partial s}{\partial y}. \tag{9b}$$

Neglecting the same strain rates reduces the expression for the effective strain rate that is used in Glen's flow law to:

$$\dot{\varepsilon}_e=\left[\left(\frac{\partial u}{\partial x}\right)^2+\left(\frac{\partial v}{\partial y}\right)^2+\frac{\partial u}{\partial x}\frac{\partial v}{\partial y}+\frac{1}{4}\left(\frac{\partial u}{\partial y}+\frac{\partial v}{\partial x}\right)^2\right]^{1/2}. \tag{10}$$

It should be noted that v1.0 further simplified Eq. 9 by neglecting the gradients in the effective viscosity (after expanding the derivative outside the square brackets using the product rule). While this made the numerical solver more stable (and also significantly faster, requiring fewer non-linear viscosity iterations to converge), it was later discovered that this could lead to
significant errors in the velocity and the ice thickness evolution. V2.0 therefore solves the SSA (and the DIVA) without this simplification, gaining physical accuracy at the cost of computational performance. Including these additional terms necessitated a change in the discretisation scheme, so that in v2.0 the ice velocities are defined on the triangle centres, whereas in v1.0 they were defined on the edges. The new discretisation scheme is presented in more detail in Appendix B.

### 2.2.4 Shallow ice approximation

The SIA arises by neglecting the membrane stresses in the BPA (i.e. $\frac{\partial u}{\partial x},\frac{\partial u}{\partial y},\frac{\partial v}{\partial x},\frac{\partial v}{\partial y}\approx 0$), leaving only the vertical shear strain rates. This is generally accepted to be a valid approximation for the thick, slow-moving ice in the interior of the Greenland and Antarctic ice sheets, where the flow is dominated by deformation due to vertical shearing, rather than by basal sliding. These assumptions simplify the Stokes equations to:

$$\frac{\partial}{\partial z}\left(\eta\frac{\partial u}{\partial z}\right)=\rho g\frac{\partial s}{\partial x}. \tag{11a}$$

$$\frac{\partial}{\partial z}\left(\eta\frac{\partial v}{\partial z}\right)=\rho g\frac{\partial s}{\partial y}. \tag{11a}$$

Similarly, the effective strain rate that is used in Glen's flow law reduces to:

$$\dot{\varepsilon}_e=\left[\frac{1}{4}\left(\frac{\partial u}{\partial z}\right)^2+\frac{1}{4}\left(\frac{\partial v}{\partial z}\right)^2\right]^{1/2}. \tag{12}$$

Substituting Eq. 12 into Eq. 11, and assuming a stress-free boundary condition at the ice surface and a no-slip boundary condition at the ice base, leads to the following analytical solution for the vertical profile of the horizontal ice velocity:

$$u(z)=-2(\rho g)^n|\nabla s|^{n-1}\nabla s\int_b^z A\big(T(\zeta)\big)(s-\zeta)^n d\zeta. \tag{13}$$

Note that it is not possible to include a sliding law in UFEMISM v2.0 when using only the SIA; for this, the hybrid SIA/SSA must be used (see Sect. 2.2.5).

### 2.2.5 Hybrid shallow ice / shallow shelf approximation

In this approach, the SIA and SSA are solved separately, following the approach proposed by Bueler and Brown (2009). Based on the observation that the flow regime in most areas of an ice sheet is generally dominated by either vertical shear (described by the SIA) or by basal sliding (described by the SSA), the two solutions are then simply added together to find an approximation of the flow of the entire ice sheet. This approach produces accurate results in terms of large-scale ice flow (e.g. Bueler and Brown, 2009; Berends et al., 2022), but starts to deviate significantly from the Stokes solution earlier than the DIVA as the length scale decreases (Berends et al., 2022; this study).

### 2.2.6 Vertical velocities

The assumption that glacial ice is incompressible is expressed mathematically as:

$$\frac{\partial u}{\partial x} + \frac{\partial v}{\partial y} + \frac{\partial w}{\partial z} = 0. \tag{14}$$

The BPA, the DIVA, and the (hybrid) SIA/SSA only solve for the horizontal velocities $u, v$. From those, the horizontal divergence $\frac{\partial u}{\partial x} + \frac{\partial v}{\partial y}$ can be calculated. Integrating Eq. 14 in the vertical dimension then yields the vertical velocity $w$:

$$w(z) = w(z = b) - \int_b^z \left( \frac{\partial u}{\partial x}(\zeta) + \frac{\partial v}{\partial y}(\zeta) \right) d\zeta. \tag{15}$$

Here too, the terrain-following coordinate transformation must be applied before evaluating the vertical integral. The way this is done in UFEMISM is described in Appendix E.

### 2.3 Sliding laws

UFEMISM v2.0 includes a number of different sliding laws for the user to choose from, which relate the basal shear stress $\tau_b$ to the basal velocity $u_b$ through the basal friction coefficient $\beta = \frac{|\tau_b|}{|u_b|}$. All sliding laws are presented here as they are coded in the model, with the basal friction coefficient $\beta$ expressed as a function of the basal speed $u_b = |u_b|$. The first option is a Weertman-type sliding law (Weertman, 1957):

$$\beta = C_w u_b^{\frac{1}{m}-1}. \tag{16}$$

Here, $C_w$ represents the (spatially variable) subglacial bed roughness.

The second option is a Coulomb-type sliding law (Iverson et al., 1998):

$$\beta = N tan\varphi u_b^{-1}. \tag{17}$$

Here, $N$ is the effective pressure between the ice and the bedrock, which is equal to the ice overburden pressure minus the subglacial water pressure. Currently, the subglacial water pressure is defined simply as 96% of the ice overburden pressure, following Winkelmann et al. (2011), optionally scaled with a bedrock elevation-dependent parameterisation developed for

Antarctica by Martin et al. (2011); the addition of a more elaborate subglacial hydrology model to UFEMISM is planned for future work. The (spatially variable) till friction angle $\varphi$ is a measure for the subglacial bed roughness.

The third option is a Budd-type sliding law, proposed by Bueler and van Pelt (2015):

$$\beta = N \tan\varphi \frac{u_b^{q-1}}{u_0^q}. \tag{18}$$

Here, $u_0$ is a transition velocity, with a default (configurable) value of 100 m yr[-1]. Note that this is a Budd-type sliding law

(i.e. a power-law dependence on velocity, scaled with the effective pressure) for the current choice of exponent $q = 0.3$; for $q = 1$, this becomes a regularised Coulomb sliding law, with no dependence on velocity. This sliding law was the only option in UFEMISM 1.0 (Berends et al., 2021).

The fourth option is the hybrid sliding law proposed by Tsai et al. (2015), as formulated by Asay-Davis et al. (2016):

$$\beta = \min\left(\alpha^2 N, \beta^2 u_b^{\frac{1}{m}}\right) u_b^{-1}. \tag{19}$$

Note that here, the (spatially variable) subglacial bed roughness is described by two separate parameters: $\alpha^2$ for the Coulomb-type part of the friction, and $\beta^2$ (which is not the square of the basal friction coefficient $\beta$, but a confusingly named separate entity, which we maintain for the sake of consistency with earlier literature) for the Weertman-type part.

The final option is the hybrid sliding law proposed by Schoof (2005), as formulated by Asay-Davis et al. (2016):

$$\beta = \frac{\beta^2 u_b^{\frac{1}{m}} \alpha^2 N}{[\beta^{2m} u_b + (\alpha^2 N)^m]^{\frac{1}{m}}} u_b^{-1}. \tag{20}$$

Note that the terms on the right-hand side of Eq. 20 are again the $\beta^2$ term from Eq. 19. In the idealised-geometry experiments presented here, the bed roughness is spatially uniform. For applications to realistic ice sheets, UFEMISM v2.0 includes routines for inverting the bed roughness by nudging. These are presented in part 2 of this work (Bernales et al., *in prep.*).

## 2.4 Sub-grid friction scaling

UFEMISM v1.0 used a grounding-line flux condition (Schoof, 2007; Pollard and DeConto, 2012) to improve grounding-line
migration. While the flux condition generally seems to produce more accurate results in unbuttressed geometries (e.g. Pattyn et al., 2012), extending this solution to geometries where buttressing plays a significant role has proved problematic (Reese et al., 2018). Furthermore, while the implementation of this scheme in v1.0 performed well in idealised geometries, it frequently resulted in numerical instability in the more complex geometries encountered in e.g. the Antarctic Ice Sheet. Therefore, in UFEMISM v2.0 the flux condition has been replaced by a sub-grid friction scaling scheme, following the approach used in
ISSM (Seroussi et al., 2014), PISM (Feldmann et al., 2014), Elmer/ice (Gagliardini et al., 2016), CISM (Leguy et al., 2021), and IMAU-ICE (Berends et al., 2022). Here, the area fraction of each mesh triangle (where the velocities are defined) that is covered by grounded ice, is calculated by bilinearly interpolating the thickness above floatation on the three vertices spanning the triangle. The basal friction coefficient $\beta$ that is calculated using the sliding law, is then multiplied with this grounded

fraction, before being used to solve the momentum balance. This approach is much more numerically stable, does not require any special treatment to include buttressing, and works well in both idealised and realistic geometries.

## 2.5 Conservation of energy

The way the heat equation inside the ice is approximated and discretised is unchanged from UFEMISM v1.0. The approximation, which is based on Greve (1997), includes terms describing horizontal and vertical advection, vertical diffusion, and internal strain heating, with the annual mean temperature of the ocean and atmosphere and the geothermal heat flux serving as boundary conditions. Horizontal diffusion and the possible formation of liquid water inside the ice column are neglected. The governing equations and their discretisation (which uses an explicit scheme for the horizontal advective terms, and an implicit scheme for the vertical advective and diffusive terms) are provided, and verified in the EISMINT-1 benchmark experiments (Huybrechts et al., 1996), by Berends et al. (2021). Unless otherwise specified by the user, the ice temperature affects the ice flow factor through an Arrhenius-type relation, following Huybrechts (1992).

## 2.6 Conservation of mass

After the momentum balance has been solved to find the ice velocities, the condition of conservation of mass can be used to find the rates of change of the ice geometry. Conservation of ice mass for a shallow layer of incompressible ice in the 2-D plane is expressed mathematically as:

$$\frac{\partial H}{\partial t} = -\nabla \cdot (\overline{\boldsymbol{u}}H) + m. \tag{21}$$

Here, $m$ is the net mass balance, including terms at the ice base, the ice surface, and the lateral boundaries, while $\overline{\boldsymbol{u}}$ is the vertically averaged, horizontally vector-valued ice velocity. Since UFEMISM always assumes a uniform, constant ice density, vertical variations in the horizontal velocities are not needed to solve the continuity equation. Eq. 21 is discretised spatially using the finite volume scheme that lent UFEMISM its name, which is derived in Appendix F, resulting in the following equation:

$$\frac{\partial H^i}{\partial t} = -M_{\text{divQ}}H^i + m^i. \tag{22}$$

Here, the ice thickness vector $H^i$ contains the values of $H$ on all the vertices $i$. $M_{\text{divQ}}$ is a matrix whose coefficients depend on the mesh geometry and the ice velocities, which can be multiplied with the ice thickness vector $H^i$ to find the ice flux divergence $\nabla \cdot (uH)^i = M_{\text{divQ}}H^i$. UFEMISM v2.0 offers three different options to discretise Eq. 22 temporally: an explicit scheme, an implicit scheme, and a semi-implicit scheme. In all three cases, the thickness rate of change $\frac{\partial H}{\partial t}$ is discretised using a simple first-order scheme. In the explicit scheme, all terms on the right-hand side of Eq. 22 are defined at time $t$:

$$\frac{H^{i,t+\Delta t}-H^{i,t}}{\Delta t} = -M_{\text{divQ}}H^{i,t} + m^{i,t}. \tag{23}$$

Rearranging the terms yields the following expression, which can be evaluated to find $H^{i,t+\Delta t}$:

$$H^{i,t+\Delta t} = \left(I - M_{\text{divQ}}\Delta t\right)H^{i,t} + m^{i,t}\Delta t. \tag{24}$$

In the implicit scheme, the ice thickness on the right-hand side of Eq. 22 is defined at time $t + \Delta t$:


$$\frac{H^{i,t+\Delta t} - H^{i,t}}{\Delta t} = -M_{\text{divQ}} H^{i,t+\Delta t} + m^{i,t}. \tag{25}$$

Rearranging the terms yields the following matrix equation that can be solved for $H^{i,t+\Delta t}$:

$$\left(I + M_{\text{divQ}}\Delta t\right) H^{i,t+\Delta t} = H^{i,t} + m^{i,t}\Delta t. \tag{26}$$

Lastly, the semi-implicit scheme is derived by defining the ice thickness on the right-hand side of Eq. 22 as the weighted average of $H^{i,t}$ and $H^{i,t+\Delta t}$:


$$\frac{H^{i,t+\Delta t} - H^{i,t}}{\Delta t} = -M_{\text{divQ}}(f_s H^{i,t+\Delta t} + (1 - f_s)H^{i,t}) + m^{i,t}. \tag{27}$$

Here, using a coefficient $f_s = 0$ implies a fully explicit scheme, $f_s = 1$ implies a fully implicit scheme, $0 < f_s < 1$ implies a semi-implicit scheme, and $f_s > 1$ implies an over-implicit scheme. Rearranging the terms yields the following matrix equation that can be solved for $H^{i,t+\Delta t}$:

$$\left(I + f_s M_{\text{divQ}}\Delta t\right) H^{i,t+\Delta t} = \left(I - \Delta t(1 - f_s)M_{\text{divQ}}\right) H^{i,t} + m^{i,t}\Delta t. \tag{28}$$

Note that the (semi-)implicit schemes are only implicit in terms of the ice thickness. The flux divergence is still computed based on the velocity solution at time step, making even the implicit scheme technically semi-implicit. Recent work by Bueler (2023) has looked into the possibilities of fully implicit solvers for the coupled momentum-mass conservation equations, but this has not (yet) been implemented in UFEMISM.

## 2.7 Time stepping

In v2.0, we use the predictor/corrector (PC) time-stepping scheme by Robinson et al. (2020). Whereas the SIA and SSA both have well-defined critical time steps, no such condition has yet been derived for the DIVA or the BPA. The predictor/corrector scheme essentially operates by calculating two solutions of $H^{i,t+\Delta t}$: one with an explicit solution of the ice velocity $u$, and one with a pseudo-implicit solution. The difference between the two solutions of $H^{i,t+\Delta t}$ is a measure for the temporal discretisation error, which can be used to adapt the time step; if the error is found to be increasing, the time step is decreased, and vice versa.

Robinson et al. (2022) showed that this scheme is particularly suitable to the DIVA (and, by extension, the BPA), where the error is less sensitive to larger time steps than in the hybrid SIA/SSA, due to the weaker dependence of the velocity on the local surface slope.

A time step in the PC scheme consists of three parts: the predictor step, the update step, and the corrector step. First, in the

predictor step, the "predicted" ice thickness is calculated, based on the current ice thickness and the current velocity solution:

$$H_{\text{pred}}^{t+\Delta t} = H^t + \Delta t^t \left[ \left(1 + \frac{\zeta_t}{2}\right) \frac{\partial H}{\partial t}(H^t, u^t) - \frac{\zeta_t}{2} \frac{\partial H}{\partial t}(H^{t-\Delta t}, u^{t-\Delta t}) \right]. \tag{29}$$

Here, $\zeta_t = \frac{\Delta t^t}{\Delta t^{t-\Delta t}}$ is the ratio between the current and the previous time steps.

Then, in the update step, a new ice velocity solution is calculated for the predicted ice thickness:

$$u^{t+\Delta t} = u\big(H_{\text{pred}}^{t+\Delta t}\big). \tag{30}$$

Lastly, in the corrector step, the "corrected" ice thickness is calculated, based on the current ice thickness and the new velocity solution:

$$H_{\text{corr}}^{t+\Delta t} = H^t + \frac{\Delta t^t}{2}\left[\frac{\partial H}{\partial t}(H^t, u^t) + \frac{\partial H}{\partial t}\big(H_{\text{pred}}^{t+\Delta t}, u^{t+\Delta t}\big)\right]. \tag{31}$$

The discretisation error $\tau$ in the ice thickness is estimated based on the difference between the predicted and the corrected ice thicknesses:

$$\tau^{t+\Delta t} = \frac{\zeta_t\big(H_{\text{corr}}^{t+\Delta t} - H_{\text{pred}}^{t+\Delta t}\big)}{(3\zeta_t + 3)\Delta t^t}. \tag{32}$$

The time step is then adapted based on the maximum discretisation error:

$$\Delta t^{t+\Delta t} = \left(\frac{\epsilon}{\max|\tau^{t+\Delta t}|}\right)^{(k_I + k_p)}\left(\frac{\epsilon}{\max|\tau^t|}\right)^{-k_p}\Delta t^t. \tag{33}$$

Here, $\epsilon$ is the target truncation error in the ice thickness rate of change (configurable, default value 3 m/yr), and $k_I = 0.2$ and $k_p = 0.1$ are tuning parameters (values taken from Robinson et al., 2020).

It should be noted that Eqs. 29 and 31 involve different realisations of the ice thickness rate of change $\frac{\partial H}{\partial t}$. However, the equations for the different ice thickness schemes (Eqs. 24, 26, and 28) yield $H^{t+\Delta t}$, for a given value of $\Delta t$. In the model, the current ice thickness $H^t$ is subtracted from that, and the remainder divided by $\Delta t$, to find $\frac{\partial H}{\partial t}$. Later on, $\frac{\partial H}{\partial t}$ is then adapted by the predictor-corrector scheme to yield the "final" value of $\frac{\partial H}{\partial t}$ that is used by the model.

## 3 Code

### 3.1 Parallelisation

A major change in v2.0 with respect to v1.0 is the switch from a shared-memory architecture, where all parts of the memory are accessible via a common bus to all computing cores, to a distributed-memory architecture, which involves communication between memory-separated computing nodes. Memory access within shared-memory nodes outperforms message passing between separated-memory nodes, which implies that, all else being equal, v2.0 would be (slightly) slower than v1.0. However, the shared-memory architecture can only run on the number of cores within a single multi-core, shared-memory node (typically 32 or 64 on many high-performance scientific computing systems). The distributed-memory architecture is not limited in this way, allowing the user to scale up to far larger numbers of cores if necessary. With distributed-memory MPI, the code path and the communication paradigm stay the same whether running on a single-node or a multi-node configuration. However, inter-nodal communication is usually much slower than intra-nodal communication which might cause an observable slowdown in the algorithm when moving to multiple nodes.

Solving the matrix equation representing the momentum balance is currently the most computationally demanding part of the model by far, often accounting for more than 80 % of the total computation time of a simulation (when using the DIVA; the hybrid SIA/SSA and the BPA are more expensive to solve, and would account for an even larger fraction). UFEMISM uses the PETSc library (Balay et al., 2021) for this. Most of the other operations that require data exchange between processes (e.g.

remapping, calculating gradients, etc.) are represented by matrix operations, which are also handled by PETSc. In cases where the user requires a process to access data from another process, UFEMISM offers a set of standardised routines that, in turn, use the MPI API to facilitate this. For example, gathering a distributed array to a single process would require allocating memory on that process (possibly after performing an MPI reduction to determine how much memory is needed), calling one of the MPI_gather routines, and finally (optionally) deallocating the memory for the distributed array. We have combined

these steps into a single subroutine to ease the workload of an aspiring UFEMISM developer. Currently, MPI_gather routines are only used for I/O, and for boundary communications when necessary for domain-wide computations that are not supported by PETSc.

We have performed some simple simulations to assess the scalability of UFEMISM v2.0. These consist of the spin-up phase

of the (modified, plan-view) MISMIP experiment, using the DIVA with an 8-km resolution at the grounding line, for a period of 10,000 years. These simulations were run on the Snellius supercomputer on the AMD Rome 7H12 nodes (of 128 cores each). These preliminary results show that v2.0 does not yet scale well when using more than 32 cores, as shown in Fig. 2. Reducing the grounding-line resolution to 2 km to increase the size of the problem does not result in better scaling. This likely has to do with the way data communication between processes is handled by PETSc, which could be improved by paying more

attention to the way the model domain is partitioned over the processes, and the way PETSc decides which data should be communicated. These improvements are reserved for future work. Another contributing factor could be that the model set-up used for the scaling test was too 'small' (i.e. had too few vertices), so that the communication latencies between cores begin to dominate the total computation time. This is supported by the slowdown observed at 64 cores. Unfortunately, the time spent on communications is not (yet) measured separately in v2.0. However, it should be noted that v2.0 in its current form is already

capable of performing multi-millennial simulations of the Antarctic Ice Sheet, solving the DIVA with a grounding-line resolution of < 5 km across selected basin-scale regions (e.g. the Amundsen Sea drainage basin), on a dual-core, consumer-grade laptop (Macbook Pro M2 2023), within 24 hours of wall-clock time (Bernales et al., *in prep.*). Large-scale practical applications of the model are therefore already feasible even without these future improvements.

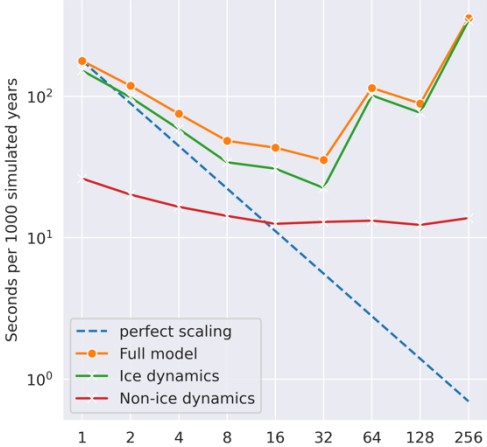

**Figure 2: Strong scaling for UFEMISM v2.0 with the 10,000-yr initialisation phase of the (modified, plan-view) MISMIP experiment, run with the DIVA (see also Sect. 4.2). The domain consists of approximately 13,000 triangles. The full model is the sum of the ice dynamics and non-ice dynamics components. I/O was disabled for these scaling tests. With more than 32 cores a slowdown instead of a speedup is visible**

### 3.2 I/O

All output files of v2.0 are now NetCDF-4 standard (Unidata, 2023), and all input files are NetCDF too (replacing the small number of text-based files that v1.0 in/outputted). UFEMISM's NetCDF input routines automatically interact with the routines for remapping data between Cartesian grids (typical of ice-sheet-specific data, e.g. BedMachine; Morlighem et al., 2019), lon/lat-grids (typical of global climate model output), and triangular meshes (e.g. output from other UFEMISM simulations). The user can provide input data in any of those formats, and UFEMISM will automatically detect the type of grid (by parsing the names of the dimensions of the NetCDF file), choose the appropriate remapping function for that grid, and remap the data to the model mesh. The sparse matrices representing the remapping operators (commonly known as 'weights'; see Appendix C) are stored in memory, so that if more data is read from the same input file later on, the matrix is reused instead of needing to calculate it again. All of this is done automatically, requiring no user intervention. Currently, $2^{nd}$-order conservative remapping is used by default; with a single keyword, this can be changed to e.g. bilinear or nearest-neighbour interpolation.

Input files that do not cover the entire computational domain are extrapolated on a nearest-neighbour basis; routines for applying a user-defined missing value, or doing a linear or Gaussian extrapolation instead exist, and can be easily integrated here. Projection parameters specified in the header of the NetCDF file are not read; UFEMISM assumes that input grids use the same projection as the model itself (i.e. the ISMIP standard projections for Greenland and Antarctica). Converting between different projections therefore must be done by the user before providing files to the model.

UFEMISM produces output on both the model mesh, and on a Cartesian grid (with a used-defined resolution). The former is useful for detailed post-processing or visualisation, while the latter can be conveniently used for cursory inspection of model output using any NetCDF viewing software, as well as for coupling to other models where square-grid input is more

convenient. The user can specify in the model configuration files which data fields should be included in the output files; the full list of the 100+ fields (both 2-D and 3-D) that the user can choose from, can be found in the NetCDF-output module.

Adding a new field requires about 10 lines of new code. The standard output includes all the data required for a "perfect restart" (so that e.g. running one simulation of 200 years yields identical results to two subsequent simulations of 100 years), which is necessary for script-based coupling to other models. Additionally, UFEMISM generates a separate NetCDF file with time series of domain-integrated quantities (e.g. mass balance components, ice volume).

### 3.3 Version control

UFEMISM is maintained on GitHub (https://github.com/IMAU-paleo/UFEMISM2.0). GitHub Actions (https://docs.github.com/en/actions) have been set up to automatically perform all the unit tests that have been built in for the routines interfacing with OpenMPI and PETSc, the NetCDF I/O routines, mesh generation, remapping, and PDE discretisation. This enables the user to quickly diagnose any problems occurring in the model. A number of benchmark experiments have been set up similarly, which are automatically run when Git branches are merged. Figures for these experiments, following the

style of the publications where these benchmark experiments were first presented (e.g. Pattyn et al., 2008 for the ISMIP-HOM experiments) are created automatically. The UFEMISM GitHub repository also features integration with the nix package manager (https://nixos.org/). This should allow the user to install all the required libraries (OpenMPI, PETSc, NetCDF) with their transient dependencies, using the exact version numbers for each of them, with a single command.

### 4 Idealised-geometry experiments

### 4.1 ISMIP-HOM

The Ice-Sheet Model Intercomparison Project for Higher-Order Models (ISMIP-HOM; Pattyn et al., 2008) contains several experiments to benchmark the velocities produced by the momentum balance in an idealised geometry. These experiments describe a slab of ice on a sloping bed. In experiments A and B, no sliding is allowed, and periodic undulations are superimposed on the flat bed slope, either in both the along-slope and cross-slope directions (experiment A), or in only the

along-slope direction (experiment B). In experiments B and C, the bedrock remains a flat slope, but sliding is now allowed, with the basal friction coefficient varying periodically in both the along-slope and cross-slope directions (experiment C), or in only the along-slope direction (experiment D). Six different versions of each experiment exist, differing in the wavelength of the bedrock undulations or the friction variations, with values ranging between 160 km and 5 km. Decreasing the wavelength increases the aspect ratio of the ice geometry, making the more simplified momentum balance approximations such as the SIA

and SSA less accurate. The experimental setup is described in full by Pattyn et al. (2008).

The velocities calculated by UFEMISM v2.0 for ISMIP-HOM experiments A and C using the hybrid SIA/SSA, the DIVA, and the BPA are compared to the ISMIP-HOM model ensemble by Pattyn et al. (2008) in Figs. 3 and 4.

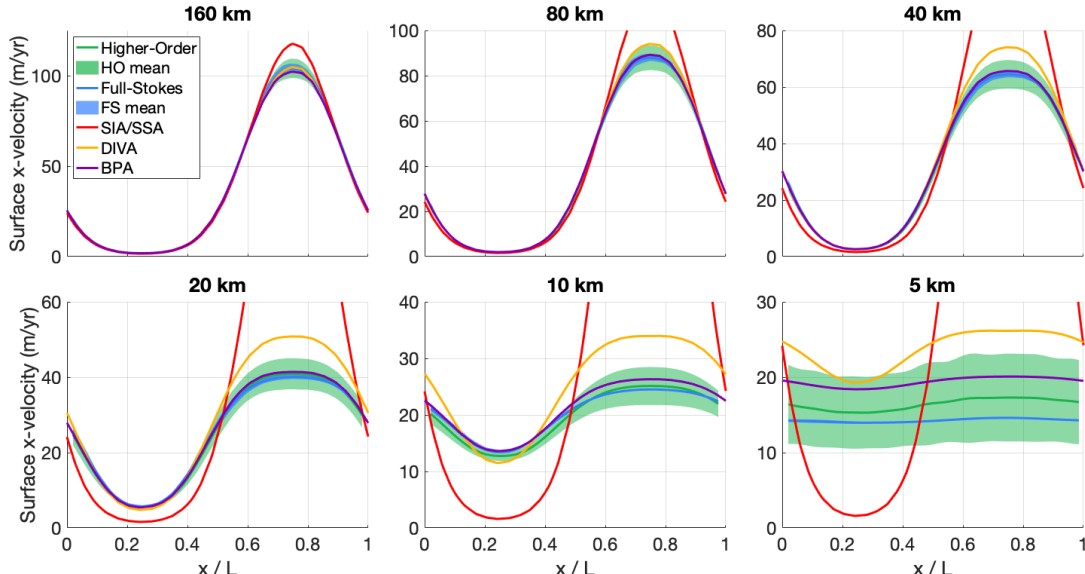

**Figure 3: Ice surface velocities calculated by UFEMISM with the hybrid SIA/SSA (red), the DIVA (yellow), and the BPA (purple) in the six different versions of ISMIP-HOM experiment A (periodic bedrock undulations in both directions), compared to the model ensemble by Pattyn et al. (2008), which is divided into the higher-order model ensemble (green), and the full-Stokes model ensemble (blue).**

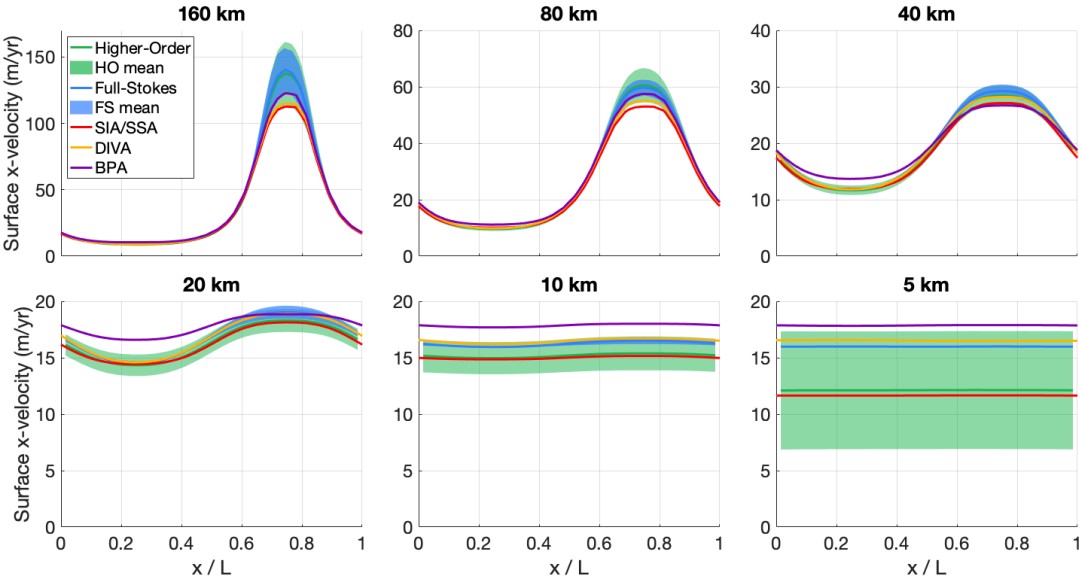

**Figure 4: Ice surface velocities calculated by UFEMISM with the hybrid SIA/SSA (red), the DIVA (yellow), and the BPA (purple) in the six different versions of ISMIP-HOM experiment C (flat sloping bed, periodic variations in friction in both directions), compared to the model ensemble by Pattyn et al. (2008), which is divided into the higher-order model ensemble (green), and the full-Stokes model ensemble (blue).**

In experiment C (Fig. 4), which concerns sliding over a bed with spatially varying roughness, all three approximations result in velocities that agree well with the ensemble, with only the BPA solution lying (slightly) outside the ensemble range, differing

from the full Stokes solution by up to 13 %. In experiment A (Fig. 3), which concerns viscous, non-sliding flow over an undulating bed, the hybrid SIA/SSA starts to diverge from the ensemble with the increasing aspect ratio of the geometry at spatial scales of about 80 km. UFEMISM's solution to the BPA lies within the ensemble for all spatial scales. The DIVA produces a relative velocity error of about 17 % in the L = 40 km experiment, which increases to 25 % and 40 % for the L = 20 km and L = 10 km experiments, respectively. The choice of what level of error is acceptable is, to some extent, subjective. Considering the inter-model spread in ensembles of realistic experiments (e.g. ISMIP6-Antarctica; Seroussi et al., 2020), and the fact that ISMIP-HOM Experiment A has rather extreme subglacial topography, we believe it is justified to use the DIVA in settings where subglacial topographical features have a typical length scale larger than 20 km. Of course, when it comes computationally feasible to use the BPA in large-scale realistic experiments, this is to be preferred.

## 4.2 MISMIP

To demonstrate the effectiveness of our sub-grid basal friction scaling scheme at resolving grounding-line migration, we performed an experiment along the lines of the Marine Ice-Sheet Intercomparison Project (MISMIP; Pattyn et al., 2012). The original experiment describes a flowline over a simple linear slope, which is subjected to a spatially uniform positive mass balance. Rather than transforming this 1-D flowline into a 2-D flowband, we have opted to extrude the 1-D geometry radially to create a circular, cone-shaped island. This results in the formation of a circular, dome-shaped ice sheet, which flows radially outward, feeding into an ice shelf that extends outward to infinity. While this means the resulting grounding-line position no longer matches the (semi-)analytical solution provided by Pattyn et al. (2012), it offers the advantage of checking the full 2-D stress balance (instead of only the x-component). The experimental protocol consists of step-wise decreasing/increasing the flow parameter $A$ in Glen's flow law, resulting in an advance/retreat of the grounding line. After being spun up to a steady state, a single advance-retreat cycle should, physically, result in the same grounding-line position as before. Any remaining difference in position, i.e. grounding-line hysteresis where there should be none, must therefore be a numerical path-dependency, which the original MISMIP study showed could be significant (up to several hundred kilometres) in models that do not pay special attention to the way the discontinuous friction at the grounding line is handled (Pattyn et al., 2012). We performed simulations with grounding-line resolutions of 10, 8, 5, and 4 km, using the DIVA. We start with a 10,000-yr spin-up phase, with a uniform flow factor of $A = 10^{-16} Pa^{-3} yr^{-1}$. We then decrease the flow factor to $A = 10^{-17} Pa^{-3} yr^{-1}$ for a period of 10,000 years, resulting in an advance of the grounding line by about 200 km. Finally, we revert the flow factor back to its original value, causing the grounding line to retreat again. While the original experiment involves several more decreases of the flow factor before moving on to the step-wise increases, only a single decrease/increased step is sufficient to assess the level of unwanted numerical grounding-line hysteresis, which is what we aim to investigate here. The results of this experiment are shown in Fig. 5; panel A shows transects of the ice sheet at the end of each of the three phases (spin-up, advance, retreat) for the 10 km simulation, while panel B shows the position of the grounding line over time for all three resolutions. The difference in grounding-line position between the end of the spin-up phase at 10 kyr, and the end of the retreat phase at 30 kyr is smaller than twice the grounding-line resolution in all simulations. Note that all these simulations were

performed with the dynamic adaptive mesh; whereas in v1.0, a mesh update would result in a small but noticeable "jump" in the grounding-line position (Berends et al., 2021, their Fig. 10b; note that that study used the hybrid SIA/SSA instead of the

DIVA, the flux condition scheme instead of the sub-grid friction scaling scheme, and much coarser resolutions of 64 – 16 km). Some improvements to the remapping scheme in v2.0 (see Appendix C) have greatly reduced this problem. Lastly, the sub-grid friction scaling scheme in v2.0 results in a more symmetrical, circular grounding line (not shown) than the flux condition scheme in v1.0. A well-known (but, as far as we are aware, never published) issue with flux condition schemes in square-grid models is the "octagonal" grounding line that can sometimes appear (in square-grid models; on unstructured grids, the grid

dependency is often less obvious); a similar undesirable dependency on the grid geometry could sometimes be seen in v1.0.

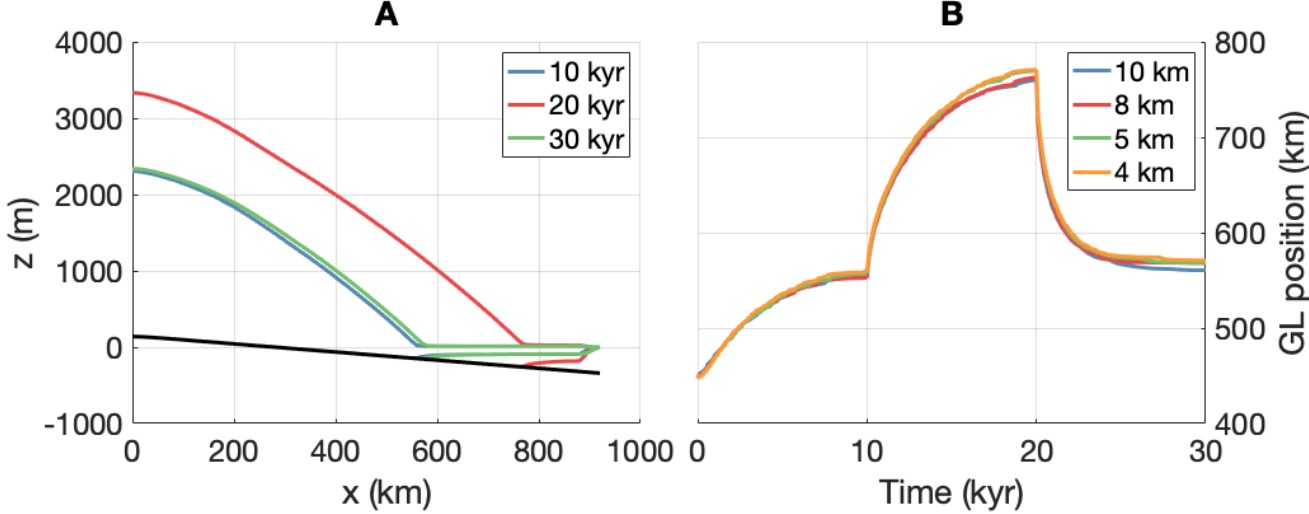

**Figure 5: A) Transects of the ice sheet at the end of each of the three phases (spin-up [blue], advance [red], retreat [green]) for the 10 km simulation. B) Grounding-line position over time, using grounding-line resolutions of 10 km (blue), 8 km (red), 5 km (green), and 4 km (orange).**

**4.3 MISMIP+**

The third Marine Ice-Sheet Model Intercomparison Project (MISMIP+; Asay-Davis et al., 2016) investigates the retreat of an ice stream feeding into a buttressed shelf. In the steady state, the ice stream flows down an 80-km wide, ~500 km long fjord. The grounding line rests on a retrograde slope, which is kept stable by the strongly buttressed ice shelf. In the experiment, the ice sheet starts from a steady state, and is subjected to a strong sub-shelf melt forcing. The resulting loss of buttressing causes

the grounding line to retreat by about 50 km over the course of the 100-yr simulation. The experimental set-up is described by Asay-Davis et al. (2016), while the results of the intercomparison are presented by Cornford et al. (2020). The resulting grounding-line retreat was found to vary by about a factor 3 between different models. A large part of this spread was attributed to (small) differences in initial conditions, as well as the choice of sliding law (the experimental protocol allows one to choose between three different sliding laws).

We have performed MISMIP+ experiment "ice1r" (100 years of increased-melt forcing) with UFEMISM v2.0, using the Schoof sliding law (Eq. 20, chosen here because, of the three options in the MISMIP+ protocol, we find it results in the best numerical stability) and the DIVA, at grounding-line resolutions ranging from 5 km to 500 m. Glen's flow law parameter $A$ has been tuned separately for each simulation to achieve a stable mid-stream grounding-line position at $x = 450$ km. The results of these simulations are compared to the model ensemble results by Cornford et al. (2020) in Fig. 6. The UFEMISM results

lie well within the Cornford et al. (2020) ensemble range. Note that these simulations were all performed with the dynamic adaptive mesh. In the 500 m simulation, the mesh was updated about once every model year on average, at no significant computational expense (as the computation time is dominated by solving the momentum balance). The solution does not seem to converge to a unique value with increasing resolution, which might be explained by the fact that the flow factor A is tuned for each experiment individually. When we repeat the simulations with the same flow factor for every resolution (not shown),

the resulting grounding-line retreat curves in Fig. 6b are more parallel, but at the cost of an increased spread in initial positions (though still mostly within +- 5 km of the 450 km target).

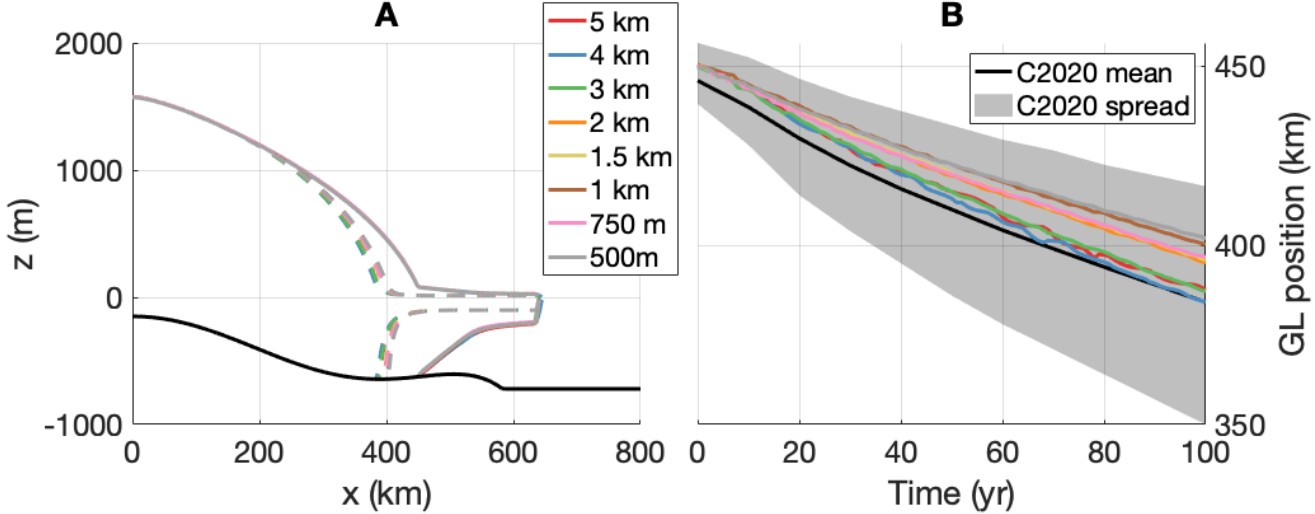

**Figure 6: A) Mid-stream transects of the ice sheet at the beginning (solid lines) and the end (dashed lines) of the 100-yr retreat simulation at different resolutions (see legend). B) Mid-stream grounding-line position over time at different resolutions (see legend),**
**compared to the Cornford et al. (2020) model ensemble (mean shown by solid black line, spread shown by grey shaded area).**

## 5 Discussion and conclusions

We have presented version 2.0 of UFEMISM and verified it in a number of benchmark experiments with idealised geometries. We have shown that the model is able to solve different approximations to the Stokes equations, and to integrate the resulting thinning rates through time to model the evolving ice geometry on a dynamic adaptive mesh. The results lie within the

published model ensembles for all these experiments. These verified model capabilities provide the groundwork for the realistic applications presented in part 2 of this work (Bernales et al., *in prep.*).

The numerical stability of the model has been greatly improved. This includes the new time-stepping scheme, as well as the switch from a simple successive over-relaxation scheme to PETSc for solving the matrix equations. While these changes have generally improved the computational performance of the model, a direct comparison between v1.0 and v2.0 is complicated by the changes that have been made to the model physics and discretisation, such as the un-simplification of the SSA, the change from a grounding-line flux condition to a sub-grid friction scaling scheme, and the change from defining velocities on the edges to the triangle centres. Comparing the performance is further complicated by the absence of several new features in v1.0 that are required for realistic simulations of the Greenland or Ice Sheet. E.g., v1.0 lacks the modules for inverting the basal friction and the sub-shelf melt, so that it cannot start from the same steady state as v2.0. Initialising the model with a spin-up instead, using simple parameterisations for the basal friction and melt, would lead to a very different, generally smoother initial ice geometry, which would artificially increase the stability of the model and inflate its performance.

The ISMIP-HOM experiments presented here, as well as the work by Rückamp et al. (2022), demonstrate the importance of considering the model's approximation to the Stokes equations when moving to high resolutions. At the high resolutions that UFEMISM can now achieve, topographical features can be resolved that would invalidate the underlying assumptions of the DIVA. However, solving the BPA can easily require 50 times more computation time than solving the DIVA, which would be unfeasible for many practical applications. Improving the model's performance when using large numbers of cores, as mentioned before, could be a way to solve this problem. Another approach could be to reduce the size of the physical problem by moving to regional ice-sheet modelling, limiting the model domain to a single drainage basin. In preparation for such an approach, the code of UFEMISM's routines for solving the ice thickness equation has been written in such a way as to easily allow the user to define regions where the ice thickness should not change.

The infinite-slab approach used by UFEMISM to simplify the momentum balance at the calving front, while not expected to greatly affect the solution, is outdated and should be replaced by a proper stress boundary condition in future work.

The current version of the model does not yet scale well, which is a major remaining point of improvement. We suspect part of this problem lies with the way PETSc is implemented in UFEMISM, and consequently, the way it handles inter-process communication. Although the (simple) mesh partitioning scheme that was created for version 1.0 (Berends et al., 2021) generally results in good load balancing, we suspect that currently, a lot of computation time is wasted by PETSc determining what data it should communicate (i.e., figuring out the non-zero structures of the different sub-matrices), when this information can already be determined a priori from the mesh connectivity. However, even with this sub-optimal performance, the model is already capable of performing high-resolution (< 5 km), multi-millennial simulations of the Antarctic Ice Sheet (Bernales et al., *in prep.*), within a few hours on a consumer-grade (dual-core) laptop (although moving to even higher resolutions would

currently still require the user to wait for several days for the simulation to complete). Improving this part of the model's performance should be the focus of future work.

## Appendix A: Mesh generation

UFEMISM uses an extended version of Ruppert's algorithm (Ruppert, 1995) to iteratively refine a simple initial mesh until it meets the requirements of the ice-sheet geometry. In Ruppert's original algorithm, the mesh is inspected to find "bad" triangles,

which are triangles whose smallest internal angle lies below a certain threshold value (typically 25°). These triangles are then "split", meaning that a new vertex is added at that triangle's circumcentre, and the Delaunay triangulation is updated to include the new vertex. In UFEMISM, Ruppert's algorithm is extended to additionally mark as "bad" those triangles whose longest leg exceeds the maximum resolution for the area of the domain where that triangle lies. For example, if the grounding line passes through a triangle whose longest leg exceeds the user-defined maximum grounding-line resolution, that triangle is

marked as "bad", even if it meets Ruppert's original smallest-angle criterion.

While the general functionality of the mesh generation code has not fundamentally changed from v1.0, the way meshes are refined is quite different now. In v2.0, the mesh generation code is provided with data fields of bedrock elevation and ice thickness, which can be defined either on a square grid or on a mesh. This geometry is then "reduced" to obtain a list of [x,y] points that together span the grounding line (and similarly for the calving front, etc.). This is illustrated in Fig. A1.

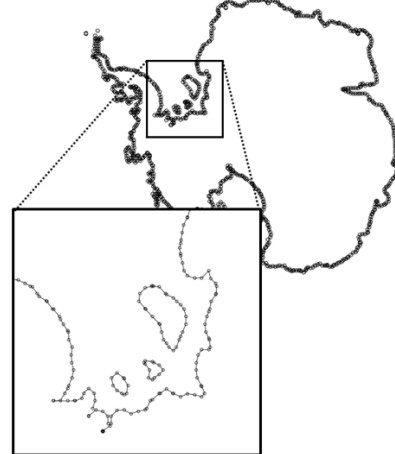


**Figure A1: The grounding line of the Ice Sheet can be represented by a series of short line segments. This grounding line was created from the BedMachine Antarctica dataset (Morlighem et al., 2019) at 40 km resolution, so that the individual segments are at most $40\sqrt{2}$ km long.**

This line is provided as input to the mesh generation code, which simply checks which triangles cross with any section of the

line, and splits them if necessary. An advantage of this approach is that the code paths for generating a mesh based on an ice-sheet geometry that is provided on a square grid (e.g. BedMachine; Morlighem et al., 2019), and for a geometry provided on

a mesh (e.g. during a mesh update in a UFEMISM simulation) are identical from the point where these geometries are reduced to lines.

In addition to the line-based mesh refinement code, v2.0 also contains point-based and polygon-based refinement routines. The point-based routine can be used to obtain a high-resolution at a certain location of interest, for example an ice-core site. The polygon-based routine can be used to increase the mesh resolution over a certain ice-sheet section, e.g. the Pine Island Glacier drainage basin. The point-based and line-based refinement are illustrated in Fig. A2.

Input mesh          Refinement          Output mesh

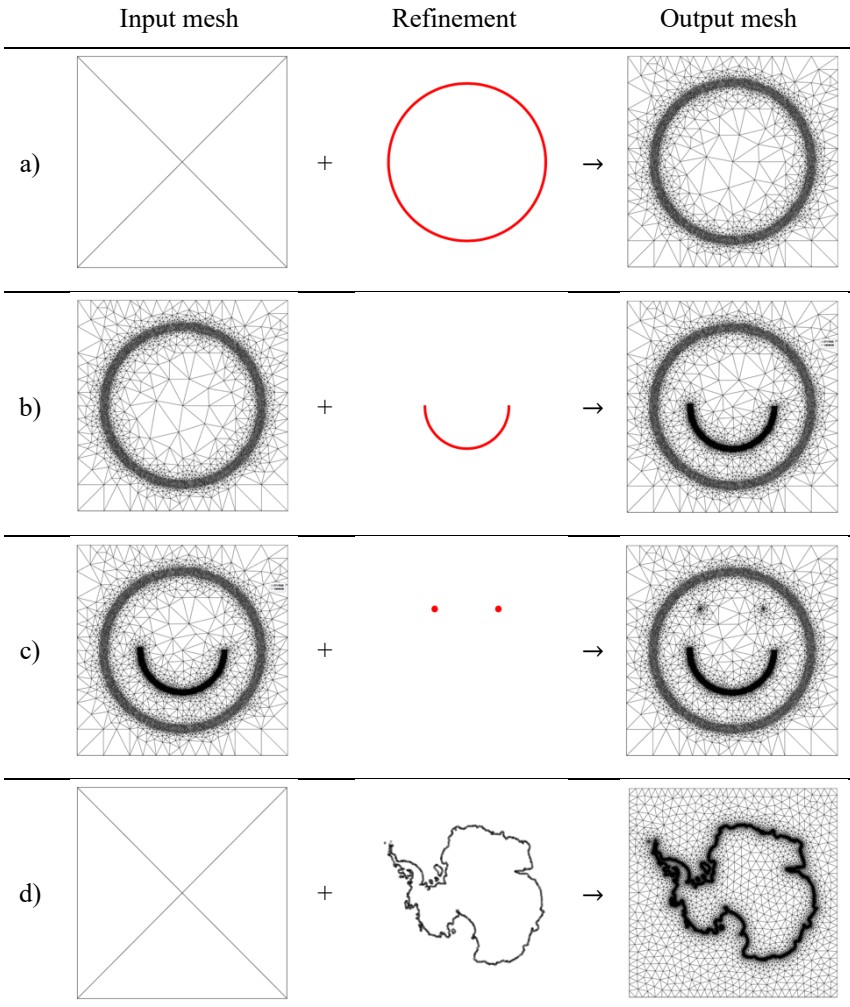

Figure A2: Each row shows how the mesh refinement algorithm refines an existing mesh (first column) with a refinement forcing (second column) to produce a new mesh (third column). a) Starting with the 5-vertex, 4-triangle "dummy" mesh, the line-refinement algorithm is provided with a series of short line segments spanning a simple circle. b) The mesh is further refined (to an even higher resolution) over a series of short line segments spanning a half-circle. c) The mesh is further refined over two points. d) A dummy mesh is refined over a series of line segments spanning the Antarctic grounding line, yielding a mesh that would be more suitable for the ice-sheet model.

Through the config file, the user can set separate maximum resolutions for the entire domain, for grounded ice, floating ice, for (a band of specified width around) the grounding line, the calving front, the grounded ice margin, and the coastline.

**Appendix B: Discretisation**

The discretisation scheme used in v1.0, described in Berends et al. (2021), which was based on neighbour functions, has been replaced by a least squares-based scheme based on Syrakos et al. (2017). The advantage of this new scheme is that it is easily extended to work on different Arakawa grids (a benefit, since due to the change in the definition of the velocities from mesh edges to mesh triangles, v2.0 makes a lot more use of staggering than v1.0 did) and to higher orders of accuracy, and that it can be coded much more elegantly.

Let $f: R^2 \rightarrow R$ be a function defined on the model domain, and let $f_a, f_b, f_c$ be its discretised approximations on respectively the mesh vertices (equivalent to the Arakawa-A grid), triangles (B-grid), and edges (C-grid). For convenience, the discretised approximations to the partial derivatives of $f$ on the different grids are written as $f_{x,a} = \left(\frac{\partial f}{\partial x}\right)_a$, $f_{yy,c} = \left(\frac{\partial^2 f}{\partial x^2}\right)_c$, etc. These partial derivatives can be expressed as linear combinations of $f_a, f_b, f_c$, e.g.:

$$f_{x,a} = M_{x,a,a} f_a. \tag{B1}$$

Here, $M_{x,a,a}$ is an $nV$-by-$nV$ matrix (with $nV$ being the number of vertices in the mesh). In the notation convention used here, $M$ has three subscript indices. The first indicates the operation represented by $M$: $x$ for $\frac{\partial}{\partial x}$, $yy$ for $\frac{\partial^2}{\partial y^2}$, etc., and $m$ for mapping $f$ between the different Arakawa grids. The second and third indices represent the source and destination Arakawa grids, respectively. E.g., $M_{m,a,b}$ maps a data field from the vertices to the triangles.

**B1: First-order, regular grid**

Syrakos et al. (2017) describe a (weighted) least-squares scheme for discretising the gradient operator on an unstructured grid. Let $f_a^i, f_{x,a}^i, f_{y,a}^i$ be the values of the function $f$ and its first partial derivatives on vertex $i$. The value $f_a^j$ of $f$ on vertex $j$, which neighbours vertex $i$, can then be expressed as a Taylor expansion of $f$ around $i$:

$$f_a^j = f_a^i + \Delta x_j f_{x,a}^i + \Delta y_j f_{y,a}^i + O\left(\Delta x_j^2, \Delta y_j^2\right). \tag{B2}$$

Here, $\Delta x_j, \Delta y_j$ is the displacement between vertices $j$ and $i$, as illustrated in Fig. B1.

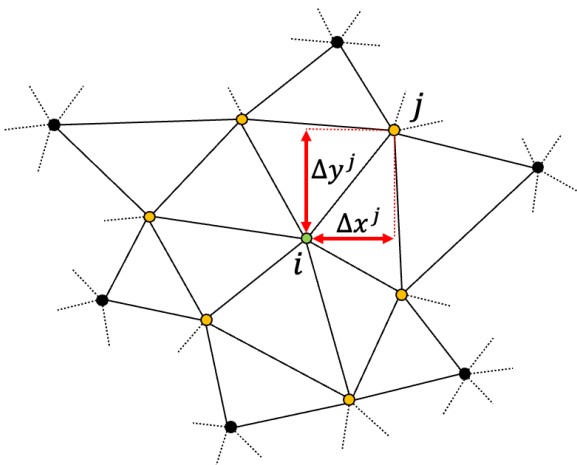


**Figure B1: illustration showing part of a mesh. Vertex $i$ is indicated by the green dot, while its six neighbours are shown in orange. Vertex $j$ is one of its neighbours, with the displacement $\Delta x^j, \Delta y^j$ shown by the red arrows.**

If $i$ has $n$ neighbours, this results in the following system of $n$ linear equations (defining $\Delta f_a^j \equiv f_a^j - f_a^i$, dropping the truncation error $O(\Delta x_j^2, \Delta y_j^2)$, and introducing the vertex weights $w_j$ for the weighted least-squares approximation):

$$
\underbrace{\begin{bmatrix} w_1 & 0 & \cdots & 0 \\ 0 & w_2 & \cdots & 0 \\ \vdots & \vdots & \ddots & \vdots \\ 0 & 0 & \cdots & w_n \end{bmatrix}}_{W} \underbrace{\begin{bmatrix} \Delta f_a^1 \\ \Delta f_a^2 \\ \vdots \\ \Delta f_a^n \end{bmatrix}}_{b} = \underbrace{\begin{bmatrix} w_1 & 0 & \cdots & 0 \\ 0 & w_2 & \cdots & 0 \\ \vdots & \vdots & \ddots & \vdots \\ 0 & 0 & \cdots & w_n \end{bmatrix}}_{W} \underbrace{\begin{bmatrix} \Delta x_1 & \Delta y_1 \\ \Delta x_2 & \Delta y_2 \\ \vdots & \vdots \\ \Delta x_n & \Delta y_n \end{bmatrix}}_{A} \underbrace{\begin{bmatrix} f_{x,a}^i \\ f_{y,a}^i \end{bmatrix}}_{z}.
$$

(B3)


Using matrix notation, this equation reads $Wb = WAz$, which can be solved for $z$:

$$
z = (A^T W^T W A)^{-1} A^T W^T W b = Q\beta_b.
$$

(B4)

Here, we have grouped the $A$ and $W$ terms into $Q = (A^T W^T W A)^{-1}$ and $\beta_b = A^T W^T W b$. The symmetric 2-by-2 matrix $A^T W^T W A$, which needs to be inverted to find $Q$, is expressed as:

$$
A^T W^T W A = \sum_{c=1}^n w_c^2.
$$

(B5)


Here, $c$ loops over all vertices that are connected to $i$ (the orange vertices in Fig. B1). The second term, $\beta_b$, is expressed as:

$$
\beta_b = \sum_{c=1}^n w_c^2 \begin{bmatrix} \Delta x_c \Delta f_a^c \\ \Delta y_c \Delta f_a^c \end{bmatrix}.
$$

(B6)

Once $Q$ has been calculated by inverting $A^T W^T W A$, the first partial derivative $f_{x,a}^i$ of $f$ on $i$ can be expressed as:

$$
f_{x,a}^i = Q(1,1) \sum_{c=1}^n (w_c^2 \Delta x_c \Delta f_a^c) + Q(1,2) \sum_{c=1}^n (w_c^2 \Delta y_c \Delta f_a^c).
$$

(B7)


Since we defined $\Delta f_a^j \equiv f_a^j - f_a^i$, this can be rewritten to read:

$$
f_{x,a}^i = -f_a^i \sum_{c=1}^n [w_c^2 (Q(1,1)\Delta x_c + Q(1,2)\Delta y_c)] + \sum_{c=1}^n f_a^c [w_c^2 (Q(1,1)\Delta x_c + Q(1,2)\Delta y_c)].
$$

(B8)

This means that the coefficients of the operator matrix $M_{x,a,a}$ are given by:

$$M_{x,a,a}^{i,j} = \begin{cases} -\sum_{c=1}^{n}[w_c^2(Q(1,1)\Delta x_c + Q(1,2)\Delta y_c)] & \text{if } i = j, \\ w_j^2(Q(1,1)\Delta x_j + Q(1,2)\Delta y_j) & \text{if } j \text{ is connected to } i, \\ 0 & \text{otherwise.} \end{cases} \tag{B9}$$

Similarly, the coefficients for $M_{y,a,a}$ are given by:

$$M_{y,a,a}^{i,j} = \begin{cases} -\sum_{c=1}^{n}[w_c^2(Q(2,1)\Delta x_c + Q(2,2)\Delta y_c)] & \text{if } i = j, \\ w_j^2(Q(2,1)\Delta x_j + Q(2,2)\Delta y_j) & \text{if } j \text{ is connected to } i, \\ 0 & \text{otherwise.} \end{cases} \tag{B10}$$

The weights $w_j$ depend on the distance between $j$ and $i$:

$$w_j = \frac{1}{|r_j - r_i|^q}. \tag{B11}$$

Following Syrakos et al. (2017), we choose $q = \frac{3}{2}$.

**B2: First-order, staggered grid**

The derivation in section B1 holds for the case where both the function $f$ and its gradients $f_x, f_y$ are defined on the same grid, so that $f_i$ is known. However, if for example we want to calculate the first partial derivative of $f$ on the mesh triangles $f_{x,b}$ when $f$ itself is defined on the mesh vertices ($f_a$), then this condition does not hold, and a slightly different derivation is needed.

Consider the Taylor series described by Eq. B2. We once again write out the system of linear equations for $f$ on the collection of neighbouring points, but this time we do not introduce $\Delta f$, so that we obtain the following expression:

$$\underbrace{\begin{bmatrix} w_1 & 0 & \cdots & 0 \\ 0 & w_2 & \cdots & 0 \\ \vdots & \vdots & \ddots & \vdots \\ 0 & 0 & \cdots & w_n \end{bmatrix}}_{W} \underbrace{\begin{bmatrix} f_a^1 \\ f_a^2 \\ \vdots \\ f_a^n \end{bmatrix}}_{b} = \underbrace{\begin{bmatrix} w_1 & 0 & \cdots & 0 \\ 0 & w_2 & \cdots & 0 \\ \vdots & \vdots & \ddots & \vdots \\ 0 & 0 & \cdots & w_n \end{bmatrix}}_{W} \underbrace{\begin{bmatrix} 1 & \Delta x_1 & \Delta y_1 \\ 1 & \Delta x_2 & \Delta y_2 \\ \vdots & \vdots & \vdots \\ 1 & \Delta x_n & \Delta y_n \end{bmatrix}}_{A} \underbrace{\begin{bmatrix} f_b^i \\ f_{x,b}^i \\ f_{y,b}^i \end{bmatrix}}_{z}. \tag{B12}$$

Following the same derivation as before, the symmetric 3-by-3 matrix $A^T W^T W A$ that needs to be inverted to find $Q$ is now given by:

$$A^T W^T W A = \sum_{c=1}^{n} w_c^2 \begin{bmatrix} 1 & \Delta x_c & \Delta y_c \\ \Delta x_c^2 & & \Delta y_c^2 \end{bmatrix}. \tag{B13}$$

Similarly, $\beta_b$ is now given by:

$$\beta_b = \sum_{c=1}^{n} w_c^2 \begin{bmatrix} f_a^c \\ \Delta x_c \Delta f_a^c \\ \Delta y_c \Delta f_a^c \end{bmatrix}. \tag{B14}$$

This leads to the following expression for the coefficients of the matrices $M_{m,a,b}, M_{x,a,b}, M_{y,a,b}$:

$$M_{m,a,b}^{i,j} = \begin{cases} -\sum_{c=1}^{n}[w_c^2(Q(1,1) + Q(1,2)\Delta x_c + Q(1,3)\Delta y_c)] & \text{if } i = j, \\ w_j^2(Q(1,1) + Q(1,2)\Delta x_j + Q(1,3)\Delta y_j) & \text{if } j \text{ is connected to } i, \\ 0 & \text{otherwise,} \end{cases} \tag{B15}$$

$$M_{x,a,b}^{i,j} = \begin{cases} -\sum_{c=1}^{n}[w_c^2(Q(2,1) + Q(2,2)\Delta x_c + Q(2,3)\Delta y_c)] & \text{if } i = j, \\ w_j^2\big(Q(2,1) + Q(2,2)\Delta x_j + Q(2,3)\Delta y_j\big) & \text{if } j \text{ is connected to } i, \\ 0 & \text{otherwise,} \end{cases} \tag{B16}$$

$$M_{y,a,b}^{i,j} = \begin{cases} -\sum_{c=1}^{n}[w_c^2(Q(3,1) + Q(3,2)\Delta x_c + Q(3,3)\Delta y_c)] & \text{if } i = j, \\ w_j^2\big(Q(3,1) + Q(3,2)\Delta x_j + Q(3,3)\Delta y_j\big) & \text{if } j \text{ is connected to } i, \\ 0 & \text{otherwise.} \end{cases} \tag{B17}$$

**B3: Second-order, regular grid**

Here, we extend the discretisation scheme by Syrakos et al. (2017) to include the second-order partial derivatives $f_{xx}, f_{xy}, f_{yy}$.

First, we extend the Taylor expansion of $f$ around $i$ to include the second-order terms:

$$f_a^j = f_a^i + \Delta x_j f_{x,a}^i + \Delta y_j f_{y,a}^i + \tfrac{1}{2}\Delta x_j^2 f_{xx,a}^i + \Delta x_j \Delta y_j f_{xy,a}^i + \tfrac{1}{2}\Delta y_j^2 f_{yy,a}^i + O\big(\Delta x_j^3, \Delta y_j^3\big). \tag{B18}$$

Writing out the system of linear equations for all neighbours of $i$ now yields the following expression:

$$\underbrace{\begin{bmatrix} w_1 & 0 & \cdots & 0 \\ 0 & w_2 & \cdots & 0 \\ \vdots & \vdots & \ddots & \vdots \\ 0 & 0 & \cdots & w_n \end{bmatrix}}_{W} \underbrace{\begin{bmatrix} f_a^1 \\ f_a^2 \\ \vdots \\ f_a^n \end{bmatrix}}_{b} = \underbrace{\begin{bmatrix} w_1 & 0 & \cdots & 0 \\ 0 & w_2 & \cdots & 0 \\ \vdots & \vdots & \ddots & \vdots \\ 0 & 0 & \cdots & w_n \end{bmatrix}}_{W} \underbrace{\begin{bmatrix} \Delta x_1 & \Delta y_1 & \tfrac{1}{2}\Delta x_1^2 & \Delta x_1 \Delta y_1 & \tfrac{1}{2}\Delta y_1^2 \\ \Delta x_2 & \Delta y_2 & \tfrac{1}{2}\Delta x_2^2 & \Delta x_2 \Delta y_2 & \tfrac{1}{2}\Delta y_2^2 \\ \vdots & \vdots & \vdots & \vdots & \vdots \\ \Delta x_n & \Delta y_n & \tfrac{1}{2}\Delta x_n^2 & \Delta x_n \Delta y_n & \tfrac{1}{2}\Delta y_n^2 \end{bmatrix}}_{W} \underbrace{\begin{bmatrix} f_{x,a}^i \\ f_{y,a}^i \\ f_{xx,a}^i \\ f_{xy,a}^i \\ f_{yy,a}^i \end{bmatrix}}_{z}. \tag{B19}$$

The symmetric 5-by-5 matrix $A^T W^T W A$ that needs to be inverted to find $Q$ is now given by:

$$A^T W^T W A = \sum_{c=1}^{n} w_c^2 \begin{bmatrix} \Delta x_c^2 & \Delta x_c \Delta y_c & \tfrac{1}{2}\Delta x_c^3 & \Delta x_c^2 \Delta y_c & \tfrac{1}{2}\Delta x_c \Delta y_c^2 \\ & \Delta y_c^2 & \tfrac{1}{2}\Delta x_c^2 \Delta y_c & \Delta x_c \Delta y_c^2 & \tfrac{1}{2}\Delta y_c^3 \\ & & \tfrac{1}{4}\Delta x_c^4 & \tfrac{1}{2}\Delta x_c^3 \Delta y_c & \tfrac{1}{4}\Delta x_c^2 \Delta y_c^2 \\ & & & \Delta x_c^2 \Delta y_c^2 & \tfrac{1}{2}\Delta x_c \Delta y_c^3 \\ & & & & \tfrac{1}{4}\Delta y_c^4 \end{bmatrix}. \tag{B20}$$

Similarly, $\beta_b$ is now given by:

$$\beta_b = \sum_{c=1}^{n} w_c^2 \begin{bmatrix} \Delta x_c \Delta f_a^c \\ \Delta y_c \Delta f_a^c \\ \tfrac{1}{2}\Delta x_c^2 \Delta f_a^c \\ \Delta x_c \Delta y_c \Delta f_a^c \\ \tfrac{1}{2}\Delta y_c^2 \Delta f_a^c \end{bmatrix}. \tag{B21}$$

Expressions for the coefficients of $M_{x,a,a}, M_{y,a,a}, M_{xx,a,a}, M_{xy,a,a}, M_{yy,a,a}$ (which are now fourth-order accurate operators) can be derived similar as before.

## Appendix C: Remapping

Because of the dynamic adaptive grid, data fields must often be remapped between square grids and (different) irregular triangular meshes. Extensive preliminary experiments have shown that only second-order conservative remapping results in accurate model results (e.g., ice thickness over time that matches the analytical solution in the Halfar dome experiment). Less accurate remapping schemes (nearest-neighbour, bilinear, biquadratic, binning, Gaussian interpolation) all result in much more diffusion during each remapping operation, and additionally violate conservation of mass and energy when remapping ice thickness and temperature, as these schemes are generally not conservative.

The mathematical theory behind conservative remapping is described by Jones (1999), and is relatively straightforward. However, Jones (1999) derived the equations in spherical coordinates, whereas UFEMISM uses Cartesian coordinates. Furthermore, UFEMISM uses a slightly different scheme, which conserves both local and global integrated values (the definition of "conservative" used by Jones), as well as extreme values (an important property, as we do not want to end up with negative ice thickness after remapping). We will therefore provide a full derivation here.

### C1: Theory

Let there exist two meshes that both cover the same domain $\Omega$: a source mesh (indicated from here by the subscript $s$) and a destination mesh (subscript $d$). Suppose the source mesh is the one that existed before a mesh update, and the destination mesh is the newly generated mesh. Let $f_{s^a}$ be a discrete function defined on the vertices of the source mesh. The remapping problem then consists of finding a new discrete function $f_{d^a}$, defined on the vertices of the destination mesh, such that:

$$\iint_\Omega f_{d^a} dA = \iint_\Omega f_{s^a} dA, \tag{C1}$$

$$\iint_{A_d^i} f_{d^a} dA = \iint_{A_d^i} f_{s^a} dA, \text{ where } A_d^i \text{ are the Voronoi cells of the vertices of the destination mesh,} \tag{C2}$$

$$\min(f_{s^a}) \le f_{d^a} \le \max(f_{s^a}), \tag{C3}$$

Here, Eq. C1 implies conservation of the global integrated value, Eq. C2 implies conservation of local integrated values, and Eq. C3 implies conservation of extreme values.

Let $f(x, y)$ be a piecewise bilinear function, which is obtained from the discrete source function on the source mesh triangles $f_{s^b}$ by bilinearly interpolating inside the triangles:

$$f(x, y) = f_{s^b} + (x - x_{s^b}) \left(\frac{\partial f}{\partial x}\right)_{s^b} + (y - y_{s^b}) \left(\frac{\partial f}{\partial y}\right)_{s^b}. \tag{C4}$$

Here, $x_{s^b}, y_{s^b}$ are the coordinates of the geometric centre of source mesh triangle $s^b$. Note that $f_{s^b}$ can be obtained from $f_{s^a}$ using the operator matrices derived in Appendix B:

$$f_{s^b} = M_{m,s^a,s^b} f_{s^a}, \tag{C5}$$

$$\left(\frac{\partial f}{\partial x}\right)_{s^b} = M_{x,s^a,s^b} f_{s^a}, \tag{C6}$$

$$\left(\frac{\partial f}{\partial y}\right)_{s^b} = M_{y,s^a,s^b} f_{s^a}. \tag{C7}$$

The discrete function $f_{d^a}$ on the vertices of the destination mesh is found by simply averaging $f(x,y)$ over the Voronoi cells $A_{d^a}$ of the vertices of the destination mesh:

$$f_{d^a} = \frac{1}{A_{d^a}} \iint_{A_{d^a}} f(x,y) dA. \tag{C8}$$

Note that, as Eq. C8 implies that $min(f(x,y)) \leq f_{d^a} \leq max(f(x,y))$, and Eq. C4 implies that $min(f_{s^a}) \leq f(x,y) \leq max(f_{s^a})$, this implies that $min(f_{s^a}) \leq f_{d^a} \leq max(f_{s^a})$, thus satisfying the conservation of extreme values required by Eq. C3. Substituting Eq. C4 into Eq. C8 yields:

$$f_{d_a} = \frac{1}{A_{d^a}} \sum_{s^b} \left[ \iint_{A_{s^b d^a}} \left( f_{s^b} + (x - x_{s^b})\left(\frac{\partial f}{\partial x}\right)_{s^b} + (y - y_{s^b})\left(\frac{\partial f}{\partial y}\right)_{s^b} \right) dA \right]. \tag{C9}$$

Here, $A_{s^b d^a}$ indicates the area of overlap between the source mesh triangles $s^b$ and the destination mesh Voronoi cells $d^a$. This is illustrated in Fig. C1.

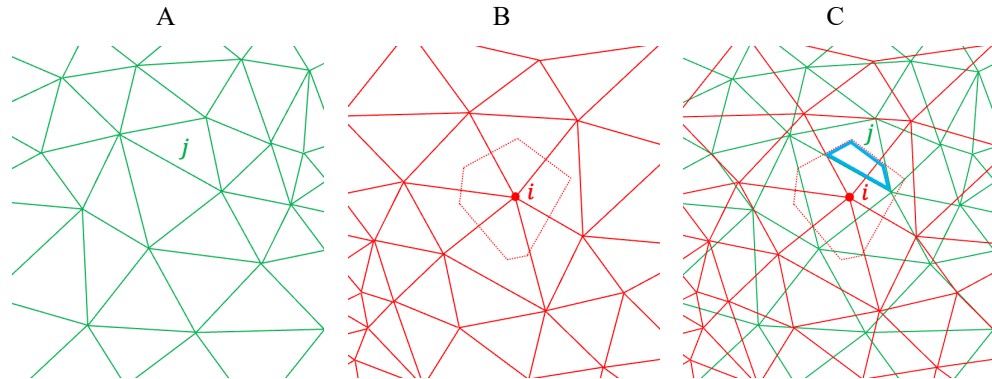

**Figure C1: A) the source mesh, with triangle $j$ indicated. B) the destination mesh, with the Voronoi cell of vertex $i$ indicated. C) the area of overlap $A_{s^{bj} d^{ai}}$ between source mesh triangle $j$ and destination mesh vertex $i$ is indicated by the thick blue line. The perimeter of this area consists of sections of the perimeter of source mesh triangle $j$, and the Voronoi cell of destination mesh vertex $i$.**

Eq. C9 can be rearranged to read:

$$f_{d_a} = \frac{1}{A_{d^a}} \sum_{s^b} \left[ f_{s^b} \iint_{A_{s^b d^a}} dA + \left(\frac{\partial f}{\partial x}\right)_{s^b} \iint_{A_{s^b d^a}} (x - x_{s^b}) dA + \left(\frac{\partial f}{\partial y}\right)_{s^b} \iint_{A_{s^b d^a}} (y - y_{s^b}) dA \right],$$

$$\frac{1}{A_{d^a}} \sum_{s^b} \left[ \left( f_{s^b} - x_{s^b}\left(\frac{\partial f}{\partial x}\right)_{s^b} - y_{s^b}\left(\frac{\partial f}{\partial y}\right)_{s^b} \right) \iint_{A_{s^b d^a}} dA + \left(\frac{\partial f}{\partial x}\right)_{s^b} \iint_{A_{s^b d^a}} x dA + \left(\frac{\partial f}{\partial y}\right)_{s^b} \iint_{A_{s^b d^a}} y dA \right]. \tag{C10}$$

Since the area of overlap $A_{s^b d^a}$ between a triangle of the source mesh and a Voronoi cell of the destination mesh will generally be an irregularly-shaped polygon, Eq. C10 is generally not easy to evaluate. However, the problem can be simplified by applying the divergence theorem, rewriting the three surface integrals in Eq. C10 into line integrals:

$$\iint_A dA = \oint_{\partial A} x dy, \tag{C11}$$

$$\iint_A x dA = -\oint_{\partial A} xy dx, \tag{C12}$$

$$\iint_A y dA = \oint_{\partial A} xy dy. \tag{C13}$$

Note that, as the perimeters of both the source mesh triangles and the destination mesh Voronoi cells are piecewise linear curves, the perimeter of the area of overlap $A_{s^b d^a}$ must therefore also be a piecewise linear curve. The expressions for the three line integrals along a straight line from $p = [x_p, y_p]$ to $q = [x_q, y_q]$ are given by:

$$\int_p^q x \, dy = x_p \Delta y - y_p \Delta x + \frac{\Delta x}{2 \Delta y}(y_q^2 - y_p^2),$$  (C14)

$$\int_p^q -xy \, dx = \frac{1}{2}\left(x_p \frac{\Delta y}{\Delta x} - y_p\right)(x_q^2 - x_p^2) - \frac{1}{3}\frac{\Delta y}{\Delta x}(x_q^3 - x_p^3),$$  (C15)

$$\int_p^q xy \, dy = \frac{1}{2}\left(x_p - y_p \frac{\Delta x}{\Delta y}\right)(y_q^2 - y_p^2) + \frac{1}{3}\frac{\Delta x}{\Delta y}(y_q^3 - y_p^3).$$  (C16)

Here, $\Delta x = x_q - x_p$, $\Delta y = y_q - q_p$. Substituting Eqs. C11 – 13 into Eq. C10 yields:

$$f_{d^a} = \frac{1}{A_{d^a}}\sum_{s^b}\left[\left(f_{s^b} - x_{s^b}\left(\frac{\partial f}{\partial x}\right)_{s^b} - y_{s^b}\left(\frac{\partial f}{\partial y}\right)_{s^b}\right)\oint_{\partial A_{s^b d^a}} x \, dy - \left(\frac{\partial f}{\partial x}\right)_{s^b}\oint_{\partial A_{s^b d^a}} xy \, dx + \left(\frac{\partial f}{\partial y}\right)_{s^b}\oint_{\partial A_{s^b d^a}} xy \, dy\right].$$  (C17)

This implies that, in order to find the remapped value of $f$ on a destination vertex, we need to find all the source triangles overlapping with that vertex' Voronoi cell, and calculate the three line integrals around the perimeter of the area of overlap

between that source triangle and the destination Voronoi cell.

As can be seen from Eq. C17, the remapped function $f_{d^a}$ is a linear combination of the triangle source function values $f_{s^b}$ and its gradients $\left(\frac{\partial f}{\partial x}\right)_{s^b}$, $\left(\frac{\partial f}{\partial y}\right)_{s^b}$, which are in turn linear combinations of the vertex source function values $f_{s^a}$. We can therefore rewrite Eq. C17 as a matrix equation. First, we define the three matrices $B_{xdy}$, $B_{-xydx}$, and $B_{xydy}$, which contain the line integrals around the areas of overlap between the source triangles $s^b$ and the destination Voronoi cells $d^a$:

$$B_{xdy}^{ij} = \oint_{\partial A_{s^{bj} d^{ai}}} x \, dy,$$  (C18)

$$B_{-xydx}^{ij} = -\oint_{\partial A_{s^{bj} d^{ai}}} xy \, dx,$$  (C19)

$$B_{xydy}^{ij} = \oint_{\partial A_{s^{bj} d^{ai}}} xy \, dy.$$  (C20)

Note that $B_{xdy}^{ij}$, $B_{-xydx}^{ij}$, and $B_{xydy}^{ij}$ are non-zero if and only if source triangle $j$ and destination Voronoi cell $i$ overlap.

These three matrices can be combined to yield the three remapping weights matrices $W_0$, $W_{1,x}$, and $W_{1,y}$:

$$W_0^{ij} = \frac{B_{xdy}^{ij}}{A_{d^{ai}}},$$  (C21)

$$W_{1,x}^{ij} = \frac{B_{-xydx}^{ij}}{A_{d^{ai}}} - W_0^{ij} x_{s^{bj}},$$  (C22)

$$W_{1,y}^{ij} = \frac{B_{xydx}^{ij}}{A_{d^{ai}}} - W_0^{ij} y_{s^{bj}}.$$  (C23)

Substituting Eqs. C21 – 23 into Eq. C17 yields:

$$f_{d^a} = W_0 f_{s^b} + W_{1,x}\left(\frac{\partial f}{\partial x}\right)_{s^b} + W_{1,y}\left(\frac{\partial f}{\partial y}\right)_{s^b}.$$  (C24)

Substituting Eqs. C5 – 7 into Eq. C24 yields:

$$f_{d^a} = \left(W_0 M_{m,s^a,s^b} + W_{1,x} M_{x,s^a,s^b} + W_{1,y} M_{y,s^a,s^b}\right) f_{s^a} = M_{s^a,d^a} f_{s^a}. \tag{C25}$$

Here, $M_{s^a,d^a} = W_0 M_{m,s^a,s^b} + W_{1,x} M_{x,s^a,s^b} + W_{1,y} M_{y,s^a,s^b}$ is an $nV_d$-by-$nV_s$ matrix that represents the second-order conservative remapping operation from the source mesh vertices to the destination mesh vertices.

## C2: Implementation

In order to calculate the remapping matrix $M_{s^a,d^a}$, the three line integrals in Eqs. C11 – 13 need to be calculated around the areas of overlap between all source mesh triangles and destination mesh Voronoi cells. While the line integrals themselves are simple enough (Eqs. C14 – 16), determining which sources triangles overlap with which destination Voronoi cells is not straightforward. Given the large numbers of vertices and triangles involved in high-resolution meshes (easily several tens of thousands of both), it is necessary to pay attention to computational efficiency.

The perimeter $\partial A_{s^{bj} d^{ai}}$ of the area of overlap $A_{s^{bj} d^{ai}}$ between source triangle $s^{bj}$ and destination Voronoi cell $d^{ai}$ consists of part of the perimeter $\partial A_{s^{bj}}$ of source triangle $s^{bj}$, and part of the perimeter $\partial A_{d^{ai}}$ of destination Voronoi cell $d^{ai}$. This means that, in order calculate the coefficients of the three matrices in Eqs. C18 – 20, it suffices to integrate once around every source triangle and around every destination Voronoi cell, carefully keeping track of the triangle or Voronoi cell of the opposite mesh with which it overlaps.

In UFEMISM, this is done using a collection of "line tracing" subroutines. Given a line $[p, q]$, the model "traces" that line through a mesh, and returns a list of all the Voronoi cells or triangles through which that line passes, and the line integrals for all the individual line segments lying within them. Great care is taken to detect cases where the perimeters of source triangles and destination Voronoi cells coincide, to prevent double-counting. By actively "tracing" the line, finding the index of the next triangle or Voronoi cell it crosses into from the connectivity lists of the triangle or cell it departs, instead of performing a mesh-

wide search operation every time, computational expense is greatly reduced. Thus, calculating the remapping matrix only takes a fraction of the computation time required to create a new mesh.

## Appendix D: Terrain-following coordinate transformation

In order to solve the BPA, the heat equation, and conservation of mass, the vertical dimension must be discretised as well. In UFEMISM, this is done by introducing a terrain-following coordinate transformation:

$$\hat{x}(x, y, z, t) = x, \tag{D1a}$$
$$\hat{y}(x, y, z, t) = y, \tag{D1b}$$
$$\zeta(x, y, z, t) = \frac{s(x,y,t) - z}{H(x,y,t)}, \tag{D1c}$$
$$\hat{t}(x, y, z, t) = t. \tag{D1d}$$

Eq. D1c implies that $\zeta = 0$ at the ice surface, and $\zeta = 1$ at the ice base. Note that, in order to transform the heat equation, the time dimension is transformed as well. Applying this coordinate transformation results in the following expressions for the gradient operators:

$$\frac{\partial}{\partial x} = \frac{\partial}{\partial \hat{x}} + \frac{\partial \zeta}{\partial x}\frac{\partial}{\partial \zeta}, \tag{D2a}$$

$$\frac{\partial}{\partial y} = \frac{\partial}{\partial \hat{y}} + \frac{\partial \zeta}{\partial y}\frac{\partial}{\partial \zeta}, \tag{D2b}$$

$$\frac{\partial}{\partial z} = \frac{\partial \zeta}{\partial z}\frac{\partial}{\partial \zeta}, \tag{D2c}$$

$$\frac{\partial}{\partial t} = \frac{\partial}{\partial \hat{t}} + \frac{\partial \zeta}{\partial t}\frac{\partial}{\partial \zeta}. \tag{D2d}$$

Applying the chain rule to Eq. D1c yields the following expressions for the gradients of $\zeta$:

$$\frac{\partial \zeta}{\partial x} = \frac{1}{H}\left(\frac{\partial s}{\partial x} - \zeta\frac{\partial H}{\partial x}\right), \tag{D3a}$$

$$\frac{\partial \zeta}{\partial y} = \frac{1}{H}\left(\frac{\partial s}{\partial y} - \zeta\frac{\partial H}{\partial y}\right), \tag{D3b}$$

$$\frac{\partial \zeta}{\partial z} = \frac{-1}{H}. \tag{D3c}$$

$$\frac{\partial \zeta}{\partial t} = \frac{1}{H}\left(\frac{\partial s}{\partial t} - \zeta\frac{\partial H}{\partial t}\right), \tag{D3b}$$

The gradient operators in Eqs. D2a – d can be represented by matrices as derived in Appendix B, by multiplying their untransformed equivalents with the gradients of $\zeta$, e.g.:

$$M_{x,a,b} = M_{\hat{x},a,b} + D\left(\frac{\partial \zeta}{\partial x}\right)M_{\zeta,a,b}. \tag{D4}$$

Here, $D(f)$ represents a diagonal matrix with the elements of the vector $f$ on the diagonal, i.e. $D^{ij} = \partial^{ij}f^i$ (with $\partial^{ij}$ being the Kronecker delta). By thus calculating the matrices for all the gradient operators, the stiffness matrix representing the momentum balance can be assembled.

The scaled vertical coordinate $\zeta$ is discretised using an irregular, log-linear grid:

$$\zeta^k = 1 - \frac{R^{\left(\frac{n-k}{n-1}\right)}-1}{R-1}, k \in [1, n]. \tag{D5}$$

This implies that the ratio between the grid spacings at the ice surface and ice base is approximately equal to $R$, which is a configurable number with a default value of $R = 10$. This scheme results in improved accuracy of the solution near the ice base, where the strain rates (in the BPA) and the temperature gradients (in the heat equation) are highest, without requiring additional vertical grid points. The number of vertical layers is configurable, and is by default set to 12.

## Appendix E: vertical ice velocities

Applying the terrain-following coordinate transformation from Appendix D to the expression for conservation of mass in Eq. 14 yields:

$$\frac{\partial u}{\partial \hat{x}} + \frac{\partial \zeta}{\partial x}\frac{\partial u}{\partial \zeta} + \frac{\partial v}{\partial \hat{y}} + \frac{\partial \zeta}{\partial y}\frac{\partial v}{\partial \zeta} + \frac{\partial \zeta}{\partial z}\frac{\partial w}{\partial \zeta} = 0. \tag{E1}$$

The terms $\frac{\partial u}{\partial \hat{x}} + \frac{\partial v}{\partial \hat{y}}$ describe the divergence in the two-dimensional plane, in scaled coordinates:

$$\frac{\partial u}{\partial \hat{x}} + \frac{\partial v}{\partial \hat{y}} = \hat{\nabla}u. \tag{E2}$$

Averaging this divergence over the Voronoi cell of a mesh vertex yields:

$$\overline{\hat{\nabla}u} = \frac{1}{A} \iint_A (\hat{\nabla}u) dA. \tag{E3}$$

By applying the divergence theorem, this integral can be transformed to a loop integral around the boundary of the Voronoi cell:

$$\overline{\hat{\nabla}u} = \frac{1}{A} \oint_{\partial A} (u \cdot \hat{n}) dS. \tag{E4}$$

Here, $\hat{n}$ is the outward normal vector to the Voronoi cell boundary. Substituting this expression into Eq. 15 yields:

$$\frac{\partial w}{\partial \zeta} = \frac{-1}{\partial \zeta / \partial z} \left[ \frac{1}{A} \oint_{\partial A} (u \cdot \hat{n}) dS + \frac{\partial \zeta}{\partial x}\frac{\partial u}{\partial \zeta} + \frac{\partial \zeta}{\partial y}\frac{\partial v}{\partial \zeta} \right]. \tag{E5}$$

This expression can then be integrated over the transformed vertical dimension to find $w$:

$$w(\zeta) = w(\zeta = 1) - \int_1^\zeta \frac{\partial w}{\partial \zeta} d\hat{\zeta}. \tag{E6}$$

Note that the minus sign in Eq. E6 arises from the fact that $\zeta$ runs from 0 at the ice surface, to 1 at the ice base, meaning that

integrating upwards from the ice base means integrating in the negative $\zeta$ direction. The vertical velocity at the base is given by:

$$w(\zeta - 1) = w_b = u_b \left( \frac{\partial s}{\partial x} - \frac{\partial H}{\partial x} \right) + v_b \left( \frac{\partial s}{\partial y} - \frac{\partial H}{\partial y} \right) + \frac{\partial s}{\partial t} - \frac{\partial H}{\partial t}. \tag{E7}$$

## Appendix F: calculating the ice flux divergence operator

Conservation of ice mass for a shallow layer of ice in the 2-D plane is expressed mathematically as:

$$\frac{\partial H}{\partial t} = -\nabla \cdot (\boldsymbol{u}H) + m. \tag{F1}$$

Here, $m$ is the net mass balance, including terms at the ice base, the ice surface, and the lateral boundaries. This equation is discretised spatially using the finite volume scheme that lent UFEMISM its name. Averaging Eq. F1 over the Voronoi cell of vertex $i$ (the control volume of the finite volume scheme) yields:

$$\frac{\partial H^i}{\partial t} = \frac{-1}{A^i} \iint_{A^i} \nabla \cdot (\boldsymbol{u}H) dA + m^i. \tag{F2}$$

Using the divergence theorem, the double integral in Eq. F2 can be transformed:

$$\frac{\partial H^i}{\partial t} = \frac{-1}{A^i} \oint_{\partial A^i} (\boldsymbol{u}H) \cdot \widehat{\boldsymbol{n}} dS + m^i. \tag{F3}$$

Here, $\widehat{\boldsymbol{n}}$ is the outward unit normal to the boundary $\partial A^i$ of the Voronoi cell of vertex $i$. Let $(\boldsymbol{u}H)^{ij}$ be average ice flux on the shared Voronoi cell boundary of vertices $i$ and $j$. Then the loop integral Eq. F3 can be transformed to a sum:

$$\frac{\partial H^i}{\partial t} = \frac{-1}{A^i} \sum_{j=1}^{n} [(\boldsymbol{u}H)^{ij} \cdot \widehat{\boldsymbol{n}}^{ij} L^{ij}] + m^i. \tag{F4}$$

Here, $\widehat{\boldsymbol{n}}^{ij}$ is the unit normal vector pointing from vertex $i$ to vertex $j$, $L^{ij}$ is the length of their shared Voronoi cell boundary, and $\sum_{j=1}^{n}$ sums over only those vertices $j$ that are connected to $i$. We then introduce an upwind scheme for the ice flux $\boldsymbol{u}H$:

$$(\boldsymbol{u}H)^{ij} = \begin{cases} \boldsymbol{u}^{ij} H^i & \text{if } \boldsymbol{u}^{ij} \cdot \widehat{\boldsymbol{n}}^{ij} > 0, \\ \boldsymbol{u}^{ij} H^j & \text{otherwise.} \end{cases} \tag{F5}$$

This implies that, if the ice flows from vertex $i$ to vertex $j$, the ice thickness in vertex $i$ determines the flux, and vice versa. This scheme offers better numerical stability than using the average ice thickness of $i$ and $j$ regardless of the flow direction.

Eqs. F4 and F5 imply that $\frac{\partial H^i}{\partial t}$ is a linear combination of the ice thicknesses $H^i$. Eq. F4 can therefore be represented by a matrix equation:

$$\frac{\partial H^i}{\partial t} = -M_{\text{divQ}} H^i + m^i. \tag{F6}$$

Here, $M_{divQ}$ is a matrix whose coefficients depend on the mesh geometry and the ice velocities, which can be multiplied with the ice thickness vector $H^i$ to find the ice flux divergence $\nabla \cdot (\boldsymbol{u}H)$. The coefficients of $M_{divQ}$ are given by:

$$M_{\text{divQ}}^{ij} = \begin{cases} \frac{1}{A^i} \sum_{j=1}^{n} [L^{ij} \max(\boldsymbol{u}^{ij} \cdot \widehat{\boldsymbol{n}}^{ij}, 0)] & \text{if } i = j, \\ \frac{L^{ij}}{A^i} \min(\boldsymbol{u}^{ij} \cdot \widehat{\boldsymbol{n}}^{ij}, 0) & \text{if } i \text{ is connected to } j, \\ 0 & \text{otherwise.} \end{cases} \tag{F7}$$

*Acknowledgements.* We will thank any reviewers who are kind enough to provide constructive criticism of the manuscript.

*Author contributions.* CJB wrote the new model code except for the distributed-memory code, which was developed by VA.
CJB performed the experiments. VA and JB set up the new version control system. CJB wrote the draft of the manuscript; all authors contributed to the final version.

*Code and data availability.* The source code of UFEMISM v2.0, scripts for compiling and running the model on a variety of computer systems, and the configuration files for all simulations presented here, are freely available on GitHub:
https://github.com/IMAU-paleo/UFEMISM2.0. The exact version of the code that was used to produce the results presented here is archived at zenodo.org (Berends et al., 2023c), though aspiring users are advised to check out the latest version from GitHub.

*Competing interests.* The authors declare that they have no competing interests.

*Financial support.* CJB was supported by PROTECT. This project has received funding from the European Union's Horizon 2020 research and innovation programme under grant agreement no. 869304 (PROTECT; [==PROTECT article number will be assigned upon acceptance for publication!==]). JB and VA were supported by NWO, grant no. OCENW.KLEIN.515.

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
