# Peer review of "The Utrecht Finite Volume Ice-Sheet Model (UFEMISM version 2.0) – part 1: description and idealised experiments"

_Geoscientific Model Development, 2024_

## Referee Comment (RC1)

**Review of *The Utrecht Finite Volume Ice-Sheet Model (UFEMISM version 2.0) – part 1: description and idealised experiments**

**General Impression**

The manuscripts describes a new version of a finite-volume ice-flow model solving different approximations to the Stokes equations, namely, 1st order (a.k.a. Blatter-Pattyn), shallow ice approximation (SIA) in combination with shallow shelf approximation (SSA) and a higher order, vertically integrated scheme, DIVA.

The article lists in great detail the differential equations representing the aforementioned approximations, the basal sliding parametrizations and the solution of the vertically integrated equation of mass conservation. This is followed by a chapter that addresses parallel execution of the code (including aspects of performance and scalability), a recently introduced interface with the NetCDF library and the code's development ecosystem. Thereafter, attention is focused on – according to the authors – verification of the code by comparison with the ensemble outputs from several inter-comparison projects, namely, ISMIP-HOM, (an altered and reduced version of) MISMIP EXP1 and MISMIP+ experiment ice1r. After a brief conclusion chapter, a rather extensive Appendix presenting certain aspects of mesh-generation and remapping and numerical discretization-schemes concludes the article.

Topically, this article appears to be well suited for this journal. I can, to a large extend follow the concept of the manuscript. Graphs are generally well implemented not overcrowded and easy to interpret (something I value). I find the mesh-adaptivity in this new version a particular interesting feature.

Nevertheless, I need to point out a few issues I found in the manuscript. What I do not really understand, is, why a new model variant has to be presented in two different papers? In my opinion – e.g., by reducing the details on presenting Stokes approximations that can be looked up in standard literature (e.g., Greve and Blatter, 2009) and numerical concepts in the Appendix – it would be more convenient for the reader to access all information (whatever is planned for part 2) from within a single manuscript, in particular as part 2 to me seems to contain information that appears to me as essential to understand statements on performance and scalability (or the lack of the latter). Furthermore, in parts, I cannot understand motivation of altered intercomparison experiments (MISMIP) or the interpretation of results (ISMIP-HOM) in terms of model and approximation accuracy. I would see the necessity to address these points (to which I will refer in the next chapter) in details, before I would consider this article to be ready for publication.

**Main points of critics**

Here I summarize the main points I see necessary to be addressed in an elaborated fashion. Please, find references to them also in the part of *Detailed list of requested changes and elaborations*. The main topics I see necessary to be addressed in the manuscript are:

**1 In parts inaccurate description of parallel implementation**

This criticism mainly refers to section 3, and in particular sub-section 3.1. I find a few claims that I simply think are not correct or conclusions that are in my opinion unjustified, related to the

information given therein.  The authors seem to have Message Passing paradigm (MPI) introduced into UFEMISM v2. I can find little information on how exactly this has been achieved. From the sentence "… data is distributed over many memory chips" I assume that they are applying something like a domain-decomposition to distribute the mesh over different tasks. I understand that the solution step (should one apply approximations that demand solution of a matrix system) is taken over by PETSc (which comes with an MPI interface), but, for instance, in SIA the algorithm will have to evaluate hydrostatic pressure gradients across domain boundaries – how is this achieved? What kind of MPI communication (blocking, non-blocking) has been implemented for exchanging data?  What also confuses me, is the quote "*UFEMISM offers a set of standardised routines that interface with the OpenMPI library (Gabriel et al., 2004) to facilitate this*".  MPI is a standard and there are several implementations on library-level, such as OpenMPI, MPICH or MVAPICH and also vendor specific MPI-libraries. What part in the code makes it necessary to particular utilize OpenMPI? This is in my opinion not unimportant, in particular in view of the initial claim in the abstract, that UFEMISM should be ready to be integrated into earth system models (ESM). ESM's generally (by the high demand from their atmospheric and ocean components) are run on dedicated supercomputers that in many cases restrict the usage of anything else than a vendor MPI (implied by the interconnection network in place) that not necessarily is derived from OpenMPI. I would ask to **add more information on how exactly MPI has been implemented for the different approximations, how PETSc is integrated on MPI level and why there is a suggested restriction to a single MPI implementation (OpenMPI)?**

**2 In parts inaccurate description and conclusions of scaling and performance analysis**

Another point I see the need to be improved is the description of performance and scalability. To me the only relevant information on performance can be derived from Figure 2, where some performance measure in terms of seconds (I guess wall clock time?) per 1000 simulated years is to be found in the annotation of the y-axis. Yet, the information there, in my opinion, is too limited to draw any clear conclusion on performance and scalability of UFEMISM v2.0. First, what model is run (1st order, hybrid SIA/SSA or DIVA) for the reported scalability test? Similar issue with the claim to run millennial Antarctic "basin-scale" simulations on a laptop – what approximation was used there and what size of problem are we looking at? Secondly, what computational platform are the tests from Fig. 2 ran on? Judging by the amount of cores, to me it appears to be a single node of a computing-cluster. From my own experience, often there are situations with memory-bound codes (and generally finite volume, finite difference, finite element fall under that category) on, e.g., AMD-EPYC systems (with versions that exactly have 2 sockets a 64 = 128 cores per node), where performance on the single node drops first at 8 cores (due to L3 cache misses) and then at 32 cores because of insufficient or insufficiently used memory bandwidth (the architecture has 4 NUMA domains) – something I also can observe in Fig. 2.,  except that intra-node scaling completely breaks down past 32 cores, which raises my doubts that inter-node scaling can be achieved at all. Also, intra-node performance highly depends on the implementation (e.g., Byckling et al., 2017). A single non-performing serial section in an OpenMP threaded application has the potential to destroy scalability (Amdahl, 1960).  A further problem (should Fig 2 demonstrate runs confined to a single node)  I have to point out that one cannot deduce inter-nodal from intra-nodal scalability – to answer this, I would ask the authors to provide scalability results run over several distributed memory nodes. Third and final point of critics is that scaling of a dimensionally reduced flow-line problem as MISMIP in my opinion is not representative of a full Antarctic setup, assuming that applications like that are the final goal to achieve scaling with. The authors must have performed MISMIP+ spinup (*ice0r*) to get a starting point for the reported melt experiment

(*ice1r*) – why not reporting performance/scaling numbers from that setup? To summarize, I would see the **necessity of elaborating the circumstances (platform, compiler, compiler-optimization flags, and most important applied approximation to Stokes) that lead to those** scalability results **and (provided we are looking at intra-nodal scaling here) extend to inter-nodal scalability tests to be in a position to evaluate code scalability for distributed memory applications, if possible for applications that solve (parts of) whole ice-sheets rather than flow-line setups.** Furthermore, **I ask to provide wall-clock times concerning the** performance**, in particular** of the most versatile approximation (**1st order/Blatter-Pattyn**) – which by ISMIP-HOM results to me appears to be the only option if sufficient accuracy is sought - **on large scale applications, such as Antarctica or MISMIP+.**

**3 Missing information on constraints of applicability of the applied approximations in view of intercomparison results and code verification**

The manuscript contains intercomparison results for ISMIP-HOM, MISMIP and MISMIP+ runs with the new code version of UFEMISM. The authors use these comparison of the results of different approximations (1st order, DIVA and SIA/SSA) as the means of verification of the new implementation of UFEMISM. Arguments have been brought forward that exact solutions rather than intercomparison should be used to verify a code (Bueler, 2005). But even in case of the latter, I would ask for a more detailed analysis. To me it appears that for high-frequency disturbances in ISMIP-HOM Exp. A, the SIA/SSA as well as the DIVA approximation are significantly deviating from the ensemble (both the HO and even more the Stokes) and in case of the bedrock friction experiment (ISMIP-HOM C) somewhat (to me surprisingly) in the lower row of Fig. 4 the hydrostatic first order (Blatter-Pattyn) solution – despite the authors claiming that it is contained in the ensemble-range for all domain scales. In particular, with respect to conclusions of inaccuracies arising in both, the SIA/SSA and DIVA approximation at smaller disturbance length scale and to me also the Blatter-Pattyn solution showing deviations in Exp. C, I would ask to provide **a discussion on the expected accuracy and the acclaimed verification of these approximation applied to ice-sheet simulation, also beyond synthetic intercomparison setups**. For the marine ice-sheet examples (MISMIP and MISMIP+) I could not find the information what approximation to the Stokes equation has been used to compute the results. I am confused by the output in Figure 5 (MISMIP), where a timescale of 30 kyr = $3.0 \times 10^2$ yr is depicted. If this should resemble Exp 2 in Pattyn et al. (2012), I am missing several further step-wise increases. Also, the (here only two varying) values of the factor *A*, to me appear several orders of magnitude larger than those used in the MISMIP experiments. I do not know the reasons to deviate that much from the MISMIP protocol, but I would ask the authors to **provide a reason for those to me rather strong deviations from the original MISMIP experiment**, such that the reader can correctly interpret the meaning of "*an experiment along the lines of the Marine Ice-Sheet Intercomparison Project*" not to try to make an attempt to compare with the results reported in Pattyn et al. (2012**). Or - in view of the claimed code verification - run the MISMIP Exp 1 and Exp 2 according to the protocol,** such that the reader can get a clear picture on how UFEMISM v2.0 behaves in view of the MISMIP ensemble.

**4 Missing information on thermo-mechanical coupled problems**

All the investigated intercomparison setups focus on pure mechanics. Yet, a changing temperature field by the Arrhenius law has a significant impact on ice-dynamics (Schoof and Hewit, 2021). The manuscript is not mentioning the inclusion or even computation of heat transfer in a coupled

thermo-mechanical context at all in the text. **I would ask the authors to add a paragraph if and how temperature (or even damage) is accounted for or included in UFEMISM?**

**Detailed list of requested changes or elaborations**

The list of issues is in order of the text indicated by page and line-number of the submitted manuscript. Quotes from the manuscript are kept in blue text.

**page 1 – line 20:** The i/o now follows the NetCDF-4 standard, including automated remapping between regular grids and irregular meshes, reducing user workload for pre- and post-processing.
What exactly do the authors mean by NetCDF standard? A certain convention, like CF?

**page 1 – line 23:** … and which is ready to be used for coupling within earth system models.
What constitutes the readiness for inclusion in ESM's? Does the code have coupler interfaces for online coupling to atmospheric or ocean models (e.g., Gladstone et al., 2021) implemented? From the scalability figures given in the text, I see a problem to run the code on large supercomputers, which I see as a necessity for inclusion in ESM's. The in my opinion unclear restriction with respect to MPI implementation (OpenMPI) most likely constitute a hurdle to run on Tier 0 or Tier 1 HPC facilities. From what is presented in this paper, I would not derive a readiness of UFEMISM v2.0 to be incorporated into ESM's. See my argumentation under major points #1 and #2.

**page 2 – line 34** Part of this large uncertainty stems from poorly constrained physical properties and processes in the Antarctic ice sheet system, including subglacial conditions (e.g. Kazmierczak et al., 2022; Berends et al., 2023a), basal sliding (Sun et al., 2020), interactions between the ice shelf and the ocean in the sub-shelf cavity (e.g. Burgard et al., 2022; Berends et al., 2023b), and ice-dynamical processes (e.g. Rückamp et al., 2022).
What about calving-induced instabilities, like MICI (e.g. Crawford et al., 2021) - or are those subsumed under ice-dynamical processes?

**page 2 – line 46:** Here, we present version 2.0 of the Utrecht Finite Volume Ice-Sheet Model (UFEMISM). Version 1.0 (Berends et al., 2021) was the second ice-sheet model to use a dynamic adaptive mesh, the first being BISICLES (Cornford et al., 2013).
May I point out that in the context of calving-front computations in Greenland, Todd et al. (2018) introduced a dynamic remeshing algorithm into *Elmer/Ice*. Additionally, in the first flowline marine ice-sheet full-Stokes experiments by Durand et al. (2009) adaptive meshing around the grounding line was already introduced 15 years ago in connection to the also in this manuscript discussed MISMIP setups. Furthermore, also *ISSM* includes mesh-adaptation, according to their web-site (https://issm.jpl.nasa.gov/documentation/mesh/).

**page 3 – line 80:** Earlier research in ice-sheet modelling has shown that the accuracy of a numerical model is particularly resolution of the grid around the grounding line (Gladstone et al., 2012; Pattyn et al., 2012)..
For flow-line problems, mesh sensitivity of grounding line positions – even in the context of full-Stokes - was described in even earlier works by Durand et al. (2009).

**page 4 – line 104:** The most complete is the Blatter-Pattyn approximation (BPA; Pattyn, 2003), which is described in Sect. 2.2.1.
"Most complete" in what sense? And I would add Blatter (1995) as a citation here.

**page 5 – line 112:** The BPA arises from the Stokes equations …
In my opinion, if the applied approximations to the Stokes equations are discussed in such details, it would be best to write out the complete Stokes equations to relate the approximations. But in

my opinion – as I also mentioned in the *General Impression* - a reference to standard literature (e.g. Greve & Blatter, 2009) or presentation of the equations from section 2.2 in an appendix would be sufficient and significantly shorten the article.

**page 5 – line 114:** The BPA produces ice velocities that are generally very close to those from the Stokes equations (Pattyn et al., 2008).

The wording "generally very close" in my view is hard to interpret and easy to misinterpret. If taken by its spatial extend of validity on large ice-sheets, one could claim that also SIA is "generally" very close to Stokes, but it completely fails under ice-domes and ridges and in fast flow regions and shelves. The ISMIP-HOM reference (Pattyn et al., 2008) is a set of idealized synthetic benchmark cases and in my opinion does not justify a statement that could be interpreted that first-order approximation is a sufficient substitute to the complete Stokes solution in every situation – which does not apply, in particular where variations in vertical advection are of essence, like in thermo-mechanically coupled problems of ice-streams (Schoof and Mantelli, 2021),  advection problems of tracers  (Jouvet et al., 2021) and flow at ridges and domes (Seddik et al., 2011).

**page 5 – line:  117**, … owing to the fact that, where those approximations either parameterise or neglect vertical variations in velocities or strain rates, the BPA solves for such variations explicitly.

To me it appears that this could be interpreted that BPA solves for all vertical variations of strain-rates. My suggestion: *… where those approximations parametrise or simply ignore the in BPA not neglected vertical derivatives of the horizontal velocity components*.

**page 5 – line 118:** This requires the model to discretise the vertical dimension as well, whereas the DIVA and the hybrid SIA/SSA operate in the two-dimensional plane, yielding a system of linear equations that is larger by a factor of the number of vertical layers in the model.

I would understand SIA to be solving column wise quadrature on a three-dimensional mesh (e.g., Greve and Blatter, 2009), so not being confined to a plane mesh.

**page 5  – line 126; Equ. (2):**

This links to major points of critics #4 that the authors seem to completely neglect the thermo-mechanical aspect of ice-sheet modelling, which in my opinion is of essence (Schoof and Hewit, 2013). In my opinion, one at least should mention that the rate factor, *A(T,p)* , is a function of the temperature and the pressure (and damage, if one wants to extend to that).

**page  6 – line 132:** The similar zero-stress boundary condition at the ice base includes a basal friction term.

To me this sentence is a contradiction: either there is a zero stress boundary or there is friction with a resulting tangential stress applied. Suggestion: *A similar dynamical boundary condition …*

**page  6 – line 144:** In the Ice-Sheet Model Intercomparison Project for Higher-Order Models (ISMIP-HOM; Pattyn et al., 2008) experiments, the DIVA produces velocities that agree well with the Stokes solution down to horizontal scales for basal topographical features of about 20 km (Berends et al., 2022; Robinson et al., 2022; this study,  Sect. 4.1).

This links to my major point #3. As DIVA is introduced to be the default solver in UFEMISM, does that mean in a complementary conclusion that DIVA should not be deployed to mesh sizes below this threshold, as then accuracy is compromised? I would like to see some sort of deeper discussion in the with respect to the rest of the manuscript extremely brief section 5.

**page 6 – line 154:** The integral term $F_2$, which can be thought of as the depth-integral of the square of the inverse viscosity, is defined as:

To me that does not come clear from (8) (i.e., there is no exponent *n=2* over the viscosity). Also, for consistency, the product  $\beta \cdot F_2$  in (7) should be dimensionless, which to me does not work out if $F_2$ is proportional to the square of the inverse viscosity.

**page 6 – line 156:** Note that, in Eq. 7, $n = 2$; Eq. 8 lists the general form because elsewhere in the DIVA, $F_1$ appears as well.

The reader might ask themselves where that would be. Are there other equations entering the system? In view of a more complex derivation of the equations of motion, I would suggest to drop everything around eqts. (7) and (8) and directly refer to look things up in Lipscomb et al. (2019)

**page 7 – line 178:** Substituting Eq. 12 into Eq. 11, and assuming a stress-free boundary condition at the ice surface and a no-slip boundary condition at the ice base, leads to the following analytical solution for the vertical profile of the horizontal ice velocity:.

I would like to have the assumption of no-slip motivated. I do not even understand it in case of hybrid SIA/SSA, as to my understanding there the sliding velocity should be provided by the SSA solution (hence non-zero).

**page 8 – line 187:** … but starts to deviate significantly from the Stokes solution earlier than the DIVA as the length scale decreases (Berends et al., 2022; this study).

This links to major point of critics #3. I would see it necessary to quantify this in terms of grid sizes that can be addressed with hybrid SIA/SSA.

**page 8 – line 205:** Here, $N$ is the effective pressure between the ice and the bedrock, which is equal to the ice overburden pressure minus the subglacial water pressure.

That leaves me (and perhaps some readers) with the question on how water pressure is determined in UFEMISM v2.0? Is there a sub-glacial hydrology model included (e.g., Gagliardini and Werder, 2018) to provide that variable?

**page 9 – line 229:** Therefore, in UFEMISM v2.0 the flux condition has been replaced by a sub-grid friction scaling scheme, following the approach used in PISM (Feldmann et al., 2014), CISM (Leguy et al., 2021), and IMAU-ICE (Berends et al., 2022).

May I point out that sub-grid friction parametrizations at grounding lines are wider spread in the ice-sheet model community. Also ISSM (Seroussi et al, 2014,) and Elmer/Ice (Gagliardini et al., 2016) deploy a sub-grid friction parametrization.

**page 9 – line 237:** Conservation of ice mass for a shallow layer of incompressible ice in the 2-D plane is expressed mathematically as:

$$\frac{\partial H}{\partial t} = -\nabla \cdot (\boldsymbol{u}H) + m$$

Equation (21) seems to be vertically integrated mass balance. Hence, $\boldsymbol{u}H$ being the vertically integrated, horizontally vector-valued, volume flux. Please, to inform the readers, add a definition of it to the text, also if/how the definition of this term differs between the available approximations to the Stokes equations.

**page 10 – line 264:** v2.0 uses the predictor/corrector (PC) time-stepping scheme by Robinson et al. (2020).

How does the predictor-corrector scheme link with the time-discretization schemes presented in section 2.5? I have the suspicion that the symbols $\Delta t$ have different meanings in 2.5 and 2.6. Please, elaborate.

**page 11 – line 191:** … a shared-memory architecture, where all data is stored in the same memory chip which all processors can access ...

To me, this is gives an over-simplified picture. Even in a shared-memory machine/node, generally, it is not a single chip containing the data. And, not all processing cores (which I would use as a term rather than processors) can access a certain chip in a similar fast way (NUMA domains). A better formulation in my view would be: … *in shared memory architecture, where all parts of the memory are accessible via a common bus to all computing cores, in contrary to distributed memory*

*architecture that demands communication between by memory separated computing nodes*.

**page 11 – line 293:** The distributed-memory architecture is slower than a shared-memory program running on the same number of processors, as data frequently needs to be exchanged between the different processor.

I would understand architecture as the hardware, which cannot directly be compared to a program (software implementation). For the latter, it very much depends on how the code is implemented. It is correct that on a pure hardware-level distributed memory access across nodes is slower (how much depends also on the performance of the interconnect-network and the memory-layout of the shared memory node) than the one of shared memory, yet, even shared memory parallelism (talking again about software) obeys Amdahl's law (Amdahl, 1965) and performance mainly hangs on the serial sequences of the code (which generally exist). Suggestion: *Memory access within shared memory nodes outperforms message passing across shared-memory nodes.*

**page 11 – line 298:** Solving the matrix equation representing the momentum balance is currently the most computationally demanding part of the model by far, often accounting for more than 80 % of the total computation time of a simulation

To my understanding, not every approximation needs a matrix system to be solved, SIA does not. And the SSA/DIVA matrix must be significant smaller than the 1st-order system. I wonder: Does the number given apply to all discussed approximations? If not, I would ask to be more specific.

**page 12 – line: 303:** … UFEMISM offers a set of standardised routines that interface with the OpenMPI library (Gabriel et al., 2004) to facilitate this..

This links to #1 of major issues. Why the constraint to the OpenMPI flavour? Does this mean UFEMISM cannot compile with another MPI-standard library, like IntelMPI or vendor specific MPI implementation? If so, please explain why.

**page 12 – line 306:** This likely has to do with the way data communication between processes is handled by PETSc, which could be improved by paying more attention to the way the model domain is partitioned over the processes, and the way PETSc decides which data should be communicated.

This also links to #1 of major issues. From the manuscript, I do not get enough information to be in a position to understand how the MPI parallelism in UFEMISM is organized. It would be interesting to the HPC inclined reader to learn how partitioning is done and - in particular with respect to the remeshing - the load balancing is guaranteed. To my knowledge, PETSc is well tuned to perform on multi-node clusters - what in particular do the authors suggest to be changed therein?

**page 12 – line 308:** However, it should be noted that v2.0 in its current form is capable of performing multi-millennial simulations of the Antarctic ice sheet, using a grounding-line resolution of < 5 km across selected basin-scale regions, on a dual-core, consumer-grade laptop (Bernales et al., in prep.). Large-scale practical applications of the model are therefore already feasible even without these future improvements.

This links to #2 of major issues. Firstly, I am missing the information what approximation applied allows one to run "basin-scale" (not sure what it means in terms of grid-sizes and spatial dimensions) on a laptop. Secondly, simulations on a laptop to me have a remote relevance to parallel performance/scaling on large machines, particular on distributed memory setups, which I understand this chapter to be about if the authors refer to "Large-scale practical applications".

**page 12 – line Figure 2:**

[Figure]

This figure is the main reason for point #2 in the list of major critics. This graph, in my opinion, needs way more explanation and discussion – also in the text, not only the caption. Like in other parts in the text, it is missing information on the approximation to the Stokes equations that is being studied here. Secondly, the informed reader might want to know on which computational platform this was run on and if we look at a single- or multiple node run. I already mentioned that a flow-line model in my view is a non-representative example for scalability if one wants to get a picture on how the code would perform and scale being applied to full ice-sheet problems. Yet, this seems to be the only place in the manuscript where the reader can get an idea on a performance baseline in terms of solved time-steps/wall clock time. I already suggested to do scalability tests with MISMIP+ if not on the full Antarctic setup. Furthermore, I would like to learn more on how the authors determine and separate ice dynamics and non-ice-dynamics parts in this figure. From a pure computational science point of view, I interpret the fact that a run on 64 and even 128 cores consumes a comparable wall-clock time as a 2 core run points to a serious issue in the parallel implementation that in my opinion prohibits production runs on compute clusters.

**page 12 – line 322:** … detect the type of grid from the dimensions of the NetCDF file …
Do the authors mean that there is some automatic parsing of the meta-data of the NetCDF file (CF convention?) that deduces the coordinate system? Further question:  is UGRID format meant when referring to triangular meshes?

**page 14 – line 365**  In experiment C (Fig. 4), which concerns sliding over a bed with spatially varying roughness, all three approximations result in velocities that agree well with the ensemble.
To me, in the lower row in Fig 4 displays the 1st order results (not tremendously, yet visible) surface velocities to exceed these of the ensemble. I would ask to explain why this is the case and – since the authors do not seem to raise any concern in the text – why it can be neglected.

**page  15 – line 273:**  The experiment describes a circular, cone-shaped island, subjected to a spatially uniform positive mass balance.
I could not find any mentioning in Pattyn et al. (2012) of a lateral circular symmetry to apply to MISMIP flowline setups.

**page 15  – line 381:**  We performed simulations with grounding-line resolutions of 10, 8, 5, and 4 km.
Like in the caption of Figure 5 on page 16, the information on what approximation to the Stokes-equation has been used for this resolution-experiments is missing.

**page 15  – line 381:**  We start with a 10,000-yr spin-up phase, with a uniform flow factor of $A$ = $10^{-16} Pa^{-3} yr^{-1}$. We then decrease the flow factor to $A$ = $10^{-17} Pa^{-3} yr^{-1}$ for a period of  10,000 years, resulting in an advance of the grounding line by about 200 km.
As mentioned in the major points of critics #3, I would ask the authors to relate parameter choice and the reduced time-span of the experiments to the original MISMIP protocol and explain – also in light of the argument of verification – this deviation.

**page 16 – line 405:**  We have performed MISMIP+ experiment "ice1r" (100 years of increased-melt forcing) with UFEMISM v2.0, using the Schoof sliding law (Eq. 20) …

Like in the MISMIP chapter, information on the applied approximation to run the MISMIP+ experiments seems to be missing, also in the caption of Fig. 5. Please add this information.

**page 17 – line 417:** We have presented version 2.0 of UFEMISM and verified its performance in a number of benchmark experiments with idealised geometries.

If this is about computational performance, I have to disagree. From this paper I am lacking information to really judge the computational performance of the code. Deducing from Fig. 2, I would even conclude that there is an unresolved issue what comes to scalability of the code. If it is about code verification, I previously mentioned that the MISMIP tests to me do not provide the means to deliver on that aspect, as they deviate from the original protocol which prohibits comparison, which leaves ISMIP-HOM (with some approximation showing strong deviations) and MISMIP+, where I could not deduce what approximation has been used for intercomparison with ensemble results.

**page 17 – line 423:** The numerical stability and computational performance of the model have been greatly improved. This includes the new time-stepping scheme, as well as the switch from a simple successive over-relaxation scheme to PETSc for solving the matrix equations. As a result, v2.0 is much faster than v1.0, capable of either running the same simulations in a fraction of the required computation time, or running a simulation in the same amount of time as before, but at a much higher resolution.

As mentioned earlier (point #2 of major points), in my view this paper is lacking information to really judge about performance of the code, as the reader is not provided with a baseline value. In my opinion, wording like "much faster" are not conveying enough information to the reader that would enable quantification of the code's performance. To really judge about performance, the reader would need to get an idea on a performance-baseline. For instance, how much the solution of one ISMIP 6 scenario run for Antarctic ice sheet using a particular approximation (preferably the optimal one, hence BPN) needs wall clock time on one, two or even more nodes of a computing cluster. From figure 2 I would draw the conclusion that the code does not scale beyond 8 cores (of whatever platform it was run on).

**page 17 – line 431:** However, solving the BPA can easily require 50 times more computation time than solving the DIVA, which would be unfeasible for many practical applications.

As I mentioned before, statements like this to me are impossible to evaluate without providing a baseline value. Just the fact that some algorithm takes 50 times longer (presumably using the same amount of computational resources) in my view does not imply that it is impossible to solve it – in particular if the code is claimed to run parallel (should in theory be able to use 50 times more resources, provided it scales) and comes with mesh-adaptation scheme.

**page 18 – line 435:** In preparation for such an approach, the code of UFEMISM's routines for solving the ice thickness equation has been written in such a way as to easily allow the user to define regions where the ice thickness should not change.

I wonder, should there be larger thickness changes in the active region, how does the model deal with the artificially imposed hydrostatic pressure gradients at boundaries with a one-sided fixed ice-thickness?

**page 18 – line :** While the model is already capable of performing high-resolution (< 5 km), multi-millennial simulations of the Antarctic ice sheet (Bernales et al., in prep.), moving to even higher resolutions would currently still require the user to wait for several days for the simulation to complete.

To me this again lacks information to really get a clear picture on what type of simulations with what approximation and what accuracy can be achieved with what computational resources. There is only information on minimum resolution, but not the approximation used. Multi-day

simulations are to me nothing that renders a computational problem impossible. But to me this sentence backs my suggestion that a fusion of the two papers would be beneficial to the reader to pick information that I conclude exists in the other manuscript (Bernales et al., in prep) to be able to relate statements presented in this one.

**page 19 – line Figure A1:**

[Figure]

Looking at the distances of the polygon-points at the zoom-in over Ronne-Filchner ice-shelf, I conclude that this is far away from the acclaimed 5 km resolution. If this is really the resolution to start from, I would ask the authors to include a sentence on how the accuracy of the coastline is increased when increasing the mesh density: is it just linearly interpolated between existing points or is additional geometrical information added to the polygon?

**page 20 – Figure A2:**

This is a large figure. I would try to simplify the composition resulting now into entry e) and thereby reduce (entries a – e) the size of the figure. Instead, what I would welcome to see included is a visual demonstration on how the remeshing algorithm could enhance mesh densities in areas of large derivatives of the velocity field, which turned out to be essential to resolve (thermo-)dynamics of fast outlet glaciers (e.g., Zhao et al. 2018) in other applications.

**page 21 – Appendix B:**

This whole part appears abstract to me and without looking things up in Syrakos et al. (2017), difficult to understand. I would even suggest that, equally, a simple reference to the paper above and removing whole part B would shorten the manuscript. If the authors want to keep it, in my opinion it would help to have some figure of local grid configurations annotated with the node-indices and showing the most important features, like distances ($\Delta x_j, \Delta y_j$) and value entries ($f_a^j$ ) to better help illustrating the formulation of the stencils as presented in this appendix, also, with respect to the different discussed mesh-types and the coordination numbers of nodes/variables therein.

**page 21 – Eqs B9 and B10:**

In my view, it would be beneficial to the reader to explain what the third case "otherwise" means. With respect to definition in the text that indices $j \in [1, n]$ indicate all neighbours of $i$, I must miss something by interpreting that neighbours are connected by definition and "otherwise" being irrelevant for A-grids. Like mentioned before, some graphical display of a local mesh arrangement in my opinion could aid the understanding and prevent misinterpretations.

**page 24 – Appendix C:**

As remeshing/-mapping seems to be one of the main new features of UFEMISM v2.0, I would suggest to move this part (at least the non-mathematical) into the main section of the paper and rather take out other parts from there (I already suggested earlier).

**page 25 – line 561:** Let there exist two meshes that both cover the same domain Ω: a source mesh (indicated from here by the subscript $s$) and a destination mesh (subscript $d$). Suppose the source mesh is the one that existed before a mesh update, and the destination mesh is the newly generated mesh.

Like before, I find this section relatively abstract and difficult to read. Similar as before, I would be of the opinion that some graphics on the mesh-to-mesh projections showing the domain, its boundary and the two (source and destination) meshes to get a picture what this is about.

**page 25 – line 566:** where $A_d^i$ are the Voronoi cells of the vertices of the destination mesh

Some readers that have not dealt with dual graphs might not know what a Voronoi cell is. Displaying this in a graph (see point above) and defining it in the text, in my view, would improve the readability of this chapter.

**List of less important issues**

Please, find here a list of minor things I only suggest to be changed.

**page 2 – line 55:** This includes a change from a shared-memory to a distributed-memory architecture (Sect. 3.1), ...
I guess the authors want to express that the code can now be run on distributed memory architecture (referring to architecture as being the hardware, rather than the program itself)? If referring to code, I would suggest to change to: *… from a shared- to a distributed memory implementation.*

**page 4 – line 91:** Some models solve this problem by using a mesh with a high resolution over a wider area, ...
In my opinion, a statement like this would demand references to be included.

**page 6 – line 141:** The DIVA, which is the default option for the momentum balance approximation in v2.0, arises by neglecting vertical variations in the membrane stresses in the BPA (i.e. $\frac{\partial}{\partial z}\left(\frac{\partial u}{\partial x}, \frac{\partial u}{\partial y}, \frac{\partial v}{\partial x}, \frac{\partial v}{\partial y}\right) \approx 0$) …
The brackets show the vertical derivatives of the components constituting the horizontal strain-rates (and not membrane stresses).

**page 11 – line 297:** processors  - I would use *cores*

**page 13  – line 344:** These experiments describe a slab of ice on a flat, sloping bed.
One group of experiments does include flat beds (C,D,F), the other (A,B - as correctly pointed out in the sentence to follow) compute on an undulated bed and Experiment E on a glacier flowline (which is not flat, either).

**page 15 – line 388:** Note that all these simulations were performed with the dynamic adaptive mesh; whereas in v1.0, a mesh update would result in a small but noticeable "jump" in the grounding-line position (Berends et al., 2021, their Fig. 390 10b), improvements to the remapping scheme in v2.0 have greatly reduced this problem..
Since remapping is mentioned here, I would suggest to make a reference to Appendix C for the reader's convenience to quickly look things up.

**page 19 – line 488:** Here, $Mx,a,a$ is an *nV*-by-*nV* matrix.
*nV* at this stage appears to me as being undefined (guess, it relates to some correlation number)

**page 21  – line 505, eq. B5; page 23 eq. B13; page 24, eq B20:**
In all equations the lower symmetry entries in the matrix are omitted. Despite the redundancy, in my view it is preferable to either spell things out in the equation or mention the unusual notation in the text.

**page 28 – line 665:** Just a suggestions, but one could elegantly use the Kronecker delta to define $D^{ij} = \delta_i^j\, f^i$ to achieve a better typesetting result in that line.

**References**

Amdahl, G (1967): *Validity of the single processor approach to achieving large scale computing capabilities*, In AFIPS '67 (Spring): Proceedings of the April 18-20, 1967, spring joint computer conference April 1967, pp 483–485 doi:10.1145/1465482.1465560

Blatter, H. (1995): *Velocity and stress fields in grounded glaciers: a simple algorithm for including deviatoric stress gradients*. Journal of Glaciology, 41 (138), 333–344

Bueler E, Lingle CS, Kallen-Brown JA, Covey DN, Bowman LN (2005): *Exact solutions and verification of numerical models for isothermal ice sheets*. Journal of Glaciology, 51(173):291-306. doi:10.3189/172756505781829449

Byckling, M., J. Kataja, M. Klemm, and T. Zwinger (2017): *OpenMP SIMD Vectorization and Threading of the Elmer Finite Element Software*, In: de Supinski B., Olivier S., Terboven C., Chapman B., Müller M. (eds) Scaling OpenMP for Exascale Performance and Portability. IWOMP 2017. Lecture Notes in Computer Science, vol 10468. Springer, doi: 10.1007/978-3-319-65578-9_9

Crawford, A.J., Benn, D.I., Todd, J, Åström, J.A, Bassis, J.N. and Zwinger, T. (2021): *Marine ice-cliff instability modeling shows mixed-mode ice-cliff failure and yields calving rate parameterization.* Nat Commun 12, 2701, doi:10.1038/s41467-021-23070-7.

Durand G., O. Gagliardini, T. Zwinger, E. Le Meur and R.C.A. Hindmarsh (2009): *Full Stokes modeling of marine ice sheets: influence of the grid size*, Ann. Glaciol., 50(52), 109–114.

Gagliardini, O., Brondex, J., Gillet-Chaulet, F., Tavard, L., Peyaud, V., and Durand, G. (2016): *Brief communication: Impact of mesh resolution for MISMIP and MISMIP3d experiments using Elmer/Ice*, The Cryosphere, 10, 307–312, doi:10.5194/tc-10-307-2016, 2016.

Gagliardini O., and Werder, M. (2018): *Influence of increasing surface melt over decadal timescales on land-terminating Greenland-type outlet glaciers*, Journal of Glaciology, 64(247), 700-710, doi:10.1017/jog.2018.59

Gladstone, R., Galton-Fenzi, B., Gwyther, D., Zhou, Q., Hattermann, T., Zhao, C., Jong, L., Xia, Y., Guo, X., Petrakopoulos, K., Zwinger, T., Shapero, D., and Moore, J. (2021): *The Framework For Ice Sheet–Ocean Coupling (FISOC) V1.1.* Geosci. Model Dev., 14, 889–905. doi:10.5194/gmd-14-889-2021

Greve, R. and Blatter, H (2009): *Dynamics of Ice Sheets and Glaciers*, Springer, Berlin, Heidelberg doi:10.1007/978-3-642-03415-2

Jouvet, G., Röllin, S., Sahli, H., Corcho, J., Gnägi, L., Compagno, L., Sidler, D., Schwikowski, M., Bauder, A., and Funk, M. (2021): *Mapping the age of ice of Gauligletscher combining surface radionuclide contamination and ice flow modelling*, The Cryosphere, 14, 4233–4251, doi:10.5194/tc-14-4233-2020, 2020.

Pattyn, F., Schoof, C., Perichon, L., Hindmarsh, R. C. A., Bueler, E., de Fleurian, B., Durand, G., Gagliardini, O., Gladstone, R. M., Goldberg, D., Gudmunsson, G. H., Huybrechts, P., Lee, V., Nick, F. M., Payne, A. J., Pollard, D., Rybak, O., Saito, F., and Vieli, A. (2012): *Results of the Marine Ice Sheet Model Intercomparison Project, MISMIP*, The Cryosphere 6, 573-588, doi:10.5194/tc-6-573-2012

Seddik H., R. Greve, T. Zwinger and L. Placidi (2011)**:** *A full-Stokes ice flow model for the vicinity of Dome Fuji, Antarctica, with induced anisotropy and fabric evolution*, The Cryosphere, 5, 495-508, doi:10.5194/tc-5-495-2011

Schoof, C, and Hewit, I. (2013): *Ice-Sheet Dynamics*, Ann. Rev. of Fluid Mech., 45: 217-239, doi: 10.1146/annurev-fluid-011212-140632

Schoof, C., and Mantelli, E. (2021): *The role of sliding in ice stream formation*. Proc. R. Soc. A 477: 20200870, doi:10.1098/rspa.2020.0870

Seroussi, H., Morlighem, M., Larour, E., Rignot, E., and Khazendar, A. (2014): *Hydrostatic grounding line parameterization in ice sheet models*, The Cryosphere, 8, 2075–2087, doi:10.5194/tc-8-2075-2014

Syrakos, A., Varchanis, S., Dimakopoulos, Y., Goulas, A., and Tsamopoulos, J. (2017): A critical analysis of some popular methods for the discretisation of the gradient operator in finite volume methods, Physics of Fluids 29, 127103, doi:10.1063/1.4997682

Todd, J., P. Christoffersen, T. Zwinger, P. Råback, N. Chauché, D. Benn, A. Luckman, J. Ryan, N. Toberg, D. Slater, and A. Hubbard (2018): *A Full-Stokes 3D Calving Model applied to a large Greenlandic Glacier*. Journal of Geophysical Research: Earth Surface, 123, 410–432, doi:10.1002/2017JF004349

Zhao, C., Gladstone, R. M., Warner, R. C., King, M. A., Zwinger, T., and Morlighem, M. (2018): *Basal friction of Fleming Glacier, Antarctica – Part 1*: *Sensitivity of inversion to temperature and bedrock uncertainty*, The Cryosphere, 12, 2637–2652, doi:10.5194/tc-12-2637-2018

---

## Author Comment (AC1)

We thank the reviewer for their constructive criticism of our manuscript, and would hereby like to respond to their concerns. Their comments are shown in italics, our response in regular type.

**Main points of critics**

> *What I do not really understand, is, why a new model variant has to be presented in two different papers? In my opinion – e.g., by reducing the details on presenting Stokes approximations that can be looked up in standard literature (e.g., Greve and Blatter, 2009) and numerical concepts in the Appendix – it would be more convenient for the reader to access all information (whatever is planned for part 2) from within a single manuscript, in particular as part 2 to me seems to contain information that appears to me as essential to understand statements on performance and scalability (or the lack of the latter).*

We think reviewer #2 summarised the reason for writing out the different Stokes approximations nicely: "...the authors put a lot of effort in writing out equations and discretizations. This is truly valuable, as the implementation on an unstructured triangular mesh adds quite some complexity." Indeed, while the equations for the Stokes approximations themselves can be found in existing literature, the discretisations of these equations as they have been implemented in UFEMISM cannot. We find it convenient to have these gathered in one place, in a consistent form which also matches the model code. The reason that we have decided to split the presentation of our new model into two parts, is that a single manuscript would become unreadably long. The 'nudging' approaches that we implemented in UFEMISM v2.0 are based on, but not identical to, earlier methods, so that they too require a comprehensive presentation of the underlying equations. While the manuscript of Part II has unfortunately been delayed (as two of the authors have had to switch contracts in the meantime, a consequence of the failing system of scientific funding in the Western world), even the current draft is already long enough to merit a separate publication.

> *The authors seem to have Message Passing paradigm (MPI) introduced into UFEMISM v2. I can find little information on how exactly this has been achieved. From the sentence "... data is distributed over many memory chips" I assume that they are applying something like a domain-decomposition to distribute the mesh over different tasks.*

Correct, v2.0 uses a domain decomposition. We will state this explicitly in the manuscript, and we will explain that v1.0 also used MPI to do parallel computations, but that in that version, only the *computations* were distributed over the processing cores (with the data gathered on a memory node that can be accessed by all cores), while in v2.0 the *data* is distributed too (thus necessitating the use of MPI communication at a lot more places in the code).

> *I understand that the solution step (should one apply approximations that demand solution of a matrix system) is taken over by PETSc (which comes with an MPI interface), but, for instance, in SIA the algorithm will have to evaluate hydrostatic pressure*

*gradients across domain boundaries – how is this achieved? What kind of MPI communication (blocking, non-blocking) has been implemented for exchanging data?*

The calculation of gradients (such as the gradient of the surface elevation that is needed to calculate ice velocities in the SIA) is done using (sparse) matrix multiplications. These are handled by PETSc, which we believe uses non-blocking MPI communication internally unless specified otherwise by the user. We will mention this in the manuscript.

*What also confuses me, is the quote "UFEMISM offers a set of standardised routines that interface with the OpenMPI library (Gabriel et al., 2004) to facilitate this". MPI is a standard and there are several implementations on library-level, such as OpenMPI, MPICH or MVAPICH and also vendor specific MPI-libraries. What part in the code makes it necessary to particular utilize OpenMPI? This is in my opinion not unimportant, in particular in view of the initial claim in the abstract, that UFEMISM should be ready to be integrated into earth system models (ESM). ESM's generally (by the high demand from their atmospheric and ocean components) are run on dedicated supercomputers that in many cases restrict the usage of anything else than a vendor MPI (implied by the interconnection network in place) that not necessarily is derived from OpenMPI. I would ask to add more information on how exactly MPI has been implemented for the different approximations, how PETSc is integrated on MPI level and why there is a suggested restriction to a single MPI implementation (OpenMPI)?*

We have clearly not explained ourselves well, for which we apologise. The UFEMISM code by itself simply calls routines from the MPI API. Whether a user has installed OpenMPI, MPICH, or another standard-compliant library should not matter (in the Nix set-up we provide, we use OpenMPI because that's what we use ourselves). The set of standardised routines we mentioned, is a collection of subroutines we wrote ourselves that provide some more convenience for aspiring developers. For example, gathering distributed data to a single processing core involves allocating the appropriate amount of memory (possibly after a reduction operation to determine how much memory is required), calling one of the MPI_gather routines, and (optionally) deallocating the memory of the distributed array. Our set of routines combine these operations to make a developer's life that much easier. We will explain this more clearly in the manuscript.

*First, what model is run (1st order, hybrid SIA/SSA or DIVA) for the reported scalability test?*

We ran the scalability tests with the spin-up phase of the (modified, plan-view) MISMIP experiment, using the DIVA with an 8-km resolution at the grounding line, for a period of 10,000 years. These simulations were run on the Snellius supercomputer on the AMD Rome 7H12 nodes (of 128 cores each). We will mention this in the manuscript.

*Similar issue with the claim to run millennial Antarctic "basin-scale" simulations on a laptop – what approximation was used there and what size of problem are we looking at?*

These are pan-Antarctic simulations of 20,000 model years that can run in less than 24 wall clock hours on 2 cores of a Macbook Pro M2 2023 using the GNU Fortran compiler version 13.2.0. They use the DIVA at a relatively low resolution between 16 and 50 km (e.g. grounding lines and East Antarctic interior, respectively), except over target basins (e.g. Pine Island and Thwaites areas) where the mesh resolution smoothly increases up to 3-5 km around the grounding line. We will mention this in the manuscript.

*Secondly, what computational platform are the tests from Fig. 2 ran on? Judging by the amount of cores, to me it appears to be a single node of a computing-cluster. From my own experience, often there are situations with memory-bound codes (and generally finite volume, finite difference, finite element fall under that category) on, e.g., AMD-EPYC systems (with versions that exactly have 2 sockets a 64 = 128 cores per node), where performance on the single node drops first at 8 cores (due to L3 cache misses) and then at 32 cores because of insufficient or insufficiently used memory bandwidth (the architecture has 4 NUMA domains) – something I also can observe in Fig. 2., except that intra-node scaling completely breaks down past 32 cores, which raises my doubts that inter-node scaling can be achieved at all. Also, intra-node performance highly depends on the implementation (e.g., Byckling et al., 2017). A single non-performing serial section in an OpenMP threaded application has the potential to destroy scalability (Amdahl, 1960).*

We agree that these results require some more information and discussion. Apart from the possible issues with PETSc, we add some discussion that strong scaling here also seems to be dependent on the architecture and communication latencies between cores. We will also mention that it is possible the experiment used in this test was too 'small', i.e with too few vertices, so that the communication time at large umbers of cores starts to dominate over the computation time. The reviewer correctly guessed the system the tests where run on (amd-epyc on snellius). We will mention this in the manuscript.

*A further problem (should Fig 2 demonstrate runs confined to a single node) I have to point out that one cannot deduce inter-nodal from intra-nodal scalability – to answer this, I would ask the authors to provide scalability results run over several distributed memory nodes.*

The code was reimplemented (from UFEMISM v1.0) with fully distributed memory MPI. This means no shared memory accesses are used as explained in section 3.1. Therefore the communication paradigm used is the same for intra and inter-nodal communication. However, the reviewer is correct to point out that the scaling will be different because for example intra-node communication is much slower than communication between two cpus on one node.  We will mention this in the manuscript.

*Third and final point of critics is that scaling of a dimensionally reduced flow-line problem as MISMIP in my opinion is not representative of a full Antarctic setup, assuming that applications like that are the final goal to achieve scaling with. The authors must have performed MISMIP+ spinup (ice0r) to get a starting point for the reported melt experiment 3 (ice1r) – why not reporting performance/scaling numbers from that setup?*

This is the same 'modified' MISMIP experiment we describe later on, which extrudes the flowline set-up from the original paper radially to create a circular, cone-shaped island. A plan-view experiment, not a flowline. We will mention this in the manuscript.

*To summarize, I would see the necessity of elaborating the circumstances (platform, compiler, compiler-optimization flags, and most important applied approximation to Stokes) that lead to those scalability results and (provided we are looking at intra-nodal scaling here) extend to inter-nodal scalability tests to be in a position to evaluate code scalability for distributed memory applications, if possible for applications that solve (parts of) whole ice-sheets rather than flow-line setups. Furthermore, I ask to provide wall-clock times concerning the performance, in particular of the most versatile approximation (1st order/Blatter-Pattyn) – which by ISMIP-HOM results to me appears to be the only option if sufficient accuracy is sought - on large scale applications, such as Antarctica or MISMIP+.*

Following our earlier responses, we will add the requested information about our scaling experiments.

We have not done any dedicated performance tests with the Blatter-Pattyn Approximation. Preliminary tests, as well, as the ISMIP-HOM experiments, indicate that the BPA is easily up to 50 times slower than the DIVA, making it impractical to use in realistic applications – at least, until the scalability problems are sorted out.

*To me it appears that for high-frequency disturbances in ISMIP-HOM Exp. A, the SIA/SSA as well as the DIVA approximation are significantly deviating from the ensemble (both the HO and even more the Stokes) and in case of the bedrock friction experiment (ISMIP-HOM C) somewhat (to me surprisingly) in the lower row of Fig. 4 the hydrostatic first order (Blatter-Pattyn) solution – despite the authors claiming that it is contained in the ensemble-range for all domain scales. In particular, with respect to conclusions of inaccuracies arising in both, the SIA/SSA and DIVA approximation at smaller disturbance length scale and to me also the Blatter-Pattyn solution showing deviations in Exp. C, I would ask to provide a discussion on the expected accuracy and the acclaimed verification of these approximation applied to ice-sheet simulation, also beyond synthetic intercomparison setups.*

We will state more clearly in the manuscript that UFEMISM's solutions to the BPA lie outside the ISMIP-HOM model ensemble for some experiments. While we do not have a clear explanation for this deviation, we do think that the deviations are quite small (relative to the ensemble range); had UFEMISM v2.0 been included in the Pattyn 2008 intercomparison exercise, we do not think it would have been excluded from the ensemble.

Regarding the accuracy of the SIA/SSA and DIVA solutions, we do not think it is within the scope of this manuscript to discuss the relative merits of these different physical approximations. We aim to focus this manuscript on *verifying* our solution of the equations, rather than *validating* the equations themselves.

> *For the marine ice-sheet examples (MISMIP and MISMIP+) I could not find the information what approximation to the Stokes equation has been used to compute the results.*

All these experiments were performed with the DIVA. We will mention this in the manuscript.

> *I am confused by the output in Figure 5 (MISMIP), where a timescale of 30 kyr = 3.0 × 10$^2$ yr is depicted. If this should resemble Exp 2 in Pattyn et al. (2012), I am missing several further step-wise increases.*

We only did the single step-wise increase/decrease, as this is enough to assess the (unwanted) grounding-line hysteresis (or "numerical path- dependency", as the other reviewer would like it to be called) present in the model. We will mention this in the manuscript.

> *Also, the (here only two varying) values of the factor A, to me appear several orders of magnitude larger than those used in the MISMIP experiments.*

The values given by Pattyn et al. (2012) are in Pa^-3 s^-1, while ours are in Pa^-3 yr^-1.

> *Or - in view of the claimed code verification - run the MISMIP Exp 1 and Exp 2 according to the protocol, such that the reader can get a clear picture on how UFEMISM v2.0 behaves in view of the MISMIP ensemble.*

The reason we have opted to extrude the 1-D geometry radially, rather than transforming the original 1-D flowline into a 2-D flowband, is that, while this means the resulting grounding-line position no longer matches the (semi-)analytical solution provided by Pattyn et al. (2012), it offers the advantage of checking the full 2-D stress balance (instead of only the x-component). This particularly allows us to check the symmetry of the grounding line. A well-known (but, as far as we are aware, never published) issue with flux condition schemes in square-grid models is the "octagonal" grounding line. A similar undesirable dependency on the grid geometry could sometimes be seen in UFEMISM v1.0 (which used a flux condition scheme), but has since been fixed with the introduction of the sub-grid friction scaling scheme in v2.0. We will mention this in the manuscript.

*All the investigated intercomparison setups focus on pure mechanics. Yet, a changing temperature field by the Arrhenius law has a significant impact on ice-dynamics (Schoof and Hewit, 2021). The manuscript is not mentioning the inclusion or even computation of heat transfer in a coupled thermo-mechanical context at all in the text. I would ask the authors to add a paragraph if and how temperature (or even damage) is accounted for or included in UFEMISM?*

The thermodynamics module of UFEMISM v2.0 is unchanged from the version in v1.0, which solves the heat equation inside the ice (excluding horizontal diffusion and possible liquid water inside the ice column). Ice damage is currently not accounted for in any way. We will mention this in the manuscript.

**Detailed list of requested changes or elaborations**

*What exactly do the authors mean by NetCDF standard? A certain convention, like CF?*

We mean that the model only produces NetCDF-4 output files (whereas v1.0 produced NetCDF-3, as well as some .txt files), and only reads NetCDF input files (whereas v1.0 required a number of .txt input files). We will clarify this in the manuscript.

*What constitutes the readiness for inclusion in ESM's? Does the code have coupler interfaces for online coupling to atmospheric or ocean models (e.g., Gladstone et al., 2021) implemented? From the scalability figures given in the text, I see a problem to run the code on large supercomputers, which I see as a necessity for inclusion in ESM's. The in my opinion unclear restriction with respect to MPI implementation (OpenMPI) most likely constitute a hurdle to run on Tier 0 or Tier 1 HPC facilities. From what is presented in this paper, I would not derive a readiness of UFEMISM v2.0 to be incorporated into ESM's. See my argumentation under major points #1 and #2.*

While we acknowledge that the computational performance of UFEMISM still needs some attention, we do not think this is an issue for coupling to ESM's. In that context, the computational load of an ice-sheet model is typically negligible compared to the atmospheric and oceanic components.

As for the coupling itself, we were thinking more of a script-based coupling than of integrating UFEMISM with the code of a GCM into a single executable. The current code already goes quite some way to making the latter possible, including clean wrapper routines for running a single, externally defined coupling time-step, and a separate library for the various remapping routines to reproject data between an ESM's (supposedly) global lon/lat-grid and UFEMISM's adaptive mesh. However, in our experience a script-based coupling is typically much easier to realise, especially for exploratory projects. To make this kind of coupling easier, UFEMISM outputs data on a (user-defined) square grid, is flexible in accepting input files from different locations and grid types, and allows for "perfect restarting" (e.g. saving the auxiliary fields that are used to

calculate the dynamic time step) so that the repeated terminations and (re)initialisations do not affect the model results.

We will mention this in the manuscript.

> *What about calving-induced instabilities, like MICI (e.g. Crawford et al., 2021) - or are those subsumed under ice-dynamical processes?*

We will add calving to this list of possible physical sources of uncertainty.

> *May I point out that in the context of calving-front computations in Greenland, Todd et al. (2018) introduced a dynamic remeshing algorithm into Elmer/Ice. Additionally, in the first flowline marine ice-sheet full-Stokes experiments by Durand et al. (2009) adaptive meshing around the grounding line was already introduced 15 years ago in connection to the also in this manuscript discussed MISMIP setups. Furthermore, also ISSM includes mesh-adaptation, according to their web-site ([https://issm.jpl.nasa.gov/documentation/mesh/](https://issm.jpl.nasa.gov/documentation/mesh/)).*

We will add references to Todd et al. (2018), Durand et al. (2009), Gladstone et al. (2010), and to dos Santos et al. (2019, for ISSM) to the manuscript, and change the phrasing accordingly.

> *For flow-line problems, mesh sensitivity of grounding line positions – even in the context of full-Stokes - was described in even earlier works by Durand et al. (2009)*

We will add this reference to this manuscript.

> *"Most complete" in what sense? And I would add Blatter (1995) as a citation here.*

In the sense that it neglects the fewest terms from the Stokes equations; we will mention this in the manuscript, and include the reference to Blatter (1995).

> *In my opinion, if the applied approximations to the Stokes equations are discussed in such details, it would be best to write out the complete Stokes equations to relate the approximations. But in my opinion – as I also mentioned in the General Impression - a reference to standard literature (e.g. Greve & Blatter, 2009) or presentation of the equations from section 2.2 in an appendix would be sufficient and significantly shorten the article.*

We will refer the reader to Greve and Blatter (2009) for a comprehensive description of the Stokes equations and the derivation of the different approximations.

> *The wording "generally very close" in my view is hard to interpret and easy to misinterpret. If taken by its spatial extend of validity on large ice-sheets, one could claim that also SIA is "generally" very close to Stokes, but it completely fails under ice-domes*

*and ridges and in fast flow regions and shelves. The ISMIP-HOM reference (Pattyn et al., 2008) is a set of idealized synthetic benchmark cases and in my opinion does not justify a statement that could be interpreted that first-order approximation is a sufficient substitute to the complete Stokes solution in every situation – which does not apply, in particular where variations in vertical advection are of essence, like in thermo-mechanically coupled problems of ice-streams (Schoof and Mantelli, 2021), advection problems of tracers (Jouvet et al., 2021) and flow at ridges and domes (Seddik et al., 2011).*

We will clarify that the BPA, and to a lesser extent the DIVA and the hybrid SIA/SSA, are generally able to describe the large-scale evolution of continental ice-sheet-shelf systems such as the Antarctic ice sheet, with the caveats mentioned by the reviewer.

*To me it appears that this could be interpreted that BPA solves for all vertical variations of strain rates. My suggestion: … where those approximations parametrise or simply ignore the in BPA not neglected vertical derivatives of the horizontal velocity components.*

We will clarify that this only applies to the vertical derivatives of the horizontal velocities and strain rates.

*I would understand SIA to be solving column wise quadrature on a three-dimensional mesh (e.g., Greve and Blatter, 2009), so not being confined to a plane mesh.*

In UFEMISM, as in (as far as we are aware) all models that offer an SIA-only option, the analytical solution to the SIA in the vertical column is used, so that it only needs to *evaluate* an expression, rather than *solve* an equation. However, the hybrid SIA/SSA, the SSA part of the solution still needs to be solved in the 2-D plane.

*This links to major points of critics #4 that the authors seem to completely neglect the thermo-mechanical aspect of ice-sheet modelling, which in my opinion is of essence (Schoof and Hewit, 2013). In my opinion, one at least should mention that the rate factor, A(T,p) , is a function of the temperature and the pressure (and damage, if one wants to extend to that).*

We will refer to the UFEMISM v1.0 model description paper, where the equations for the heat equation and the thermomechanical coupling are provided.

*To me this sentence is a contradiction: either there is a zero stress boundary or there is friction with a resulting tangential stress applied. Suggestion: A similar dynamical boundary condition …*

Accepted.

*This links to my major point #3. As DIVA is introduced to be the default solver in UFEMISM, does that mean in a complementary conclusion that DIVA should not be deployed to mesh sizes below this threshold, as then accuracy is compromised? I would like to see some sort of deeper discussion in the with respect to the rest of the manuscript extremely brief section 5.*

Technically, it is not the higher mesh resolution that invalidates the DIVA (or the SIA/SSA, or the BPA), but rather the appearance of smaller topographical features that can be resolved by a finer mesh. That said, we do not wish to move too far into the ice-dynamical territory where discussions about the validity and applicability of different Stokes approximations belong. While papers about these discussions are highly valuable, we think our manuscript is already long enough when limited to describing only a computer program that *solves* these approximations. Given this deliberate choice of scope, we believe the existing paragraph in Sect. 5 ("The ISMIP-HOM experiments presented here … thickness should not change.") adequately covers the subject.

*To me that does not come clear from (8) (i.e., there is no exponent n=2 over the viscosity). Also, for consistency, the product $\beta \cdot F2$ in (7) should be dimensionless, which to me does not work out if $F2$ is proportional to the square of the inverse viscosity.*

The description of the equation is incorrect, it should simply be "a (scaled) depth-integral of the inverse viscosity". We will change this in the manuscript. We have also checked the units and verified that the product beta*F_n is indeed dimensionless.

*The reader might ask themselves where that would be. Are there other equations entering the system? In view of a more complex derivation of the equations of motion, I would suggest to drop everything around eqts. (7) and (8) and directly refer to look things up in Lipscomb et al. (2019).*

We appreciate the suggestion, but we will keep the presentation of the physical equations in their current form.

*I would like to have the assumption of no-slip motivated. I do not even understand it in case of hybrid SIA/SSA, as to my understanding there the sliding velocity should be provided by the SSA solution (hence non-zero).*

In the hybrid SIA/SSA case (which is described in the next paragraph), the sliding velocities from the SSA are indeed added to the vertical shear velocities from the SIA. However, when UFEMISM is used in SIA-only mode (which is possible, but is not done in any of the experiments presented here), it is not possible to include a sliding law (meaning that we did not include the option in UFEMISM). We will clarify this in the manuscript.

*This links to major point of critics #3. I would see it necessary to quantify this in terms of grid sizes that can be addressed with hybrid SIA/SSA.*

See above for our reply regarding the scope of the manuscript.

> *That leaves me (and perhaps some readers) with the question on how water pressure is determined in UFEMISM v2.0? Is there a sub-glacial hydrology model included (e.g., Gagliardini and Werder, 2018) to provide that variable?*

There is not. Currently, the sub-glacial water pressure is defined as 96 % of the ice overburden pressure (following Winkelmann et al., 2011), optionally scaled with a bedrock elevation-dependent parameterization developed for Antarctica (following Martin et al., 2011), which we will mention in the manuscript. The inclusion of a sub-glacial hydrology model is high on the priority list for future updates.

> *May I point out that sub-grid friction parametrizations at grounding lines are wider spread in the ice-sheet model community. Also ISSM (Seroussi et al, 2014,) and Elmer/Ice (Gagliardini et al., 2016) deploy a sub-grid friction parametrization.*

We will add these references to the manuscript.

> *Equation (21) seems to be vertically integrated mass balance. Hence, $\boldsymbol{u}H$ being the vertically integrated, horizontally vector-valued, volume flux. Please, to inform the readers, add a definition of it to the text, also if/how the definition of this term differs between the available approximations to the Stokes equations.*

We will mention that **u** here is the vertically integrated, horizontally vector-valued ice velocity. The equations presented in this paragraph do not differ between the available approximations to the Stokes equations; since UFEMISM always assumes a uniform, constant ice density, only the vertically averaged ice velocity is needed in the continuity equation.

> *How does the predictor-corrector scheme link with the time-discretization schemes presented in section 2.5? I have the suspicion that the symbols $\Delta t$ have different meanings in 2.5 and 2.6. Please, elaborate.*

Correct, the are some subtleties to the use of $\Delta$t. The different schemes in Sect. 2.5 require a value of $\Delta$t in order to yield H(t+$\Delta$t), while the P/C-scheme in Sect. 2.6 requires the rate of change dH/dt. Currently, the model solves this mismatch by taking H(t+$\Delta$t) (which is provided by whatever scheme, explicit, implicit, the user chooses), subtracting H(t) and dividing by the $\Delta$t that went into the scheme. This $\Delta$t is later on adapted by the P/C-scheme to yield the "final" $\Delta$t that is used by the model. We will mention this in the manuscript.

> *To me, this gives an over-simplified picture. Even in a shared-memory machine/node, generally, it is not a single chip containing the data. And, not all processing cores (which I would use as a term rather than processors) can access a certain chip in a similar fast way (NUMA domains). A better formulation in my view would be: … in shared memory*

*architecture, where all parts of the memory are accessible via a common bus to all computing cores, in contrary to distributed memory architecture that demands communication between by memory separated computing nodes.*

Accepted.

*I would understand architecture as the hardware, which cannot directly be compared to a program (software implementation). For the latter, it very much depends on how the code is implemented. It is correct that on a pure hardware-level distributed memory access across nodes is slower (how much depends also on the performance of the interconnect-network and the memory-layout of the shared memory node) than the one of shared memory, yet, even shared memory parallelism (talking again about software) obeys Amdahl's law (Amdahl, 1965) and performance mainly hangs on the serial sequences of the code (which generally exist). Suggestion: Memory access within shared memory nodes outperforms message passing across shared-memory nodes.*

Accepted.

*To my understanding, not every approximation needs a matrix system to be solved, SIA does not. And the SSA/DIVA matrix must be significant smaller than the 1st-order system. I wonder: Does the number given apply to all discussed approximations? If not, I would ask to be more specific.*

In SIA-only mode, this number would likely be a lot smaller. However, we do not foresee UFEMISM being used this way in any applications where computation time is a limiting factor (in fact, the only time we've ever used it is in the EISMINT-1 benchmark experiments described in the v1.0 paper). The 80% number is for the DIVA, which is (in our experience) the 'fastest' of all approximations except the SIA. For the hybrid SIA/SSA and the BPA, this number would be higher still.  We will mention this in the manuscript.

*This links to #1 of major issues. Why the constraint to the OpenMPI flavour? Does this mean UFEMISM cannot compile with another MPI-standard library, like IntelMPI or vendor specific MPI implementation? If so, please explain why.*

We agree this phrasing was confusing; while we use OpenMPI on our system, UFEMISM should be able to compile and run with any other MPI-standard library. We will clarify this in the manuscript.

*This also links to #1 of major issues. From the manuscript, I do not get enough information to be in a position to understand how the MPI parallelism in UFEMISM is organized. It would be interesting to the HPC inclined reader to learn how partitioning is done and - in particular with respect to the remeshing - the load balancing is guaranteed. To my knowledge, PETSc is well tuned to perform on multi-node clusters - what in particular do the authors suggest to be changed therein?*

We do not think the problem lies with PETSc itself, but rather with the way we have implemented PETSc within UFEMISM. Mesh partitioning is currently done by simply partitioning the rectangular domain into equal-sized 'strips' along a single dimension. Optionally, the width of the resulting sub-domains can be tuned to make sure each process gets (approximately) equal numbers of vertices. We have done some (simple) load balancing experiments for version 1.0, which suggested that this approach works reasonably well. Where we suspect the current performance problem lies, is with how PETSc determines (or is told) the connectivity between the vertices, and by implication, the non-zero structure of the sparse matrices it must work with. In the current implementation, in every non-linear viscosity iteration (where the sparse matrix equation must be solved), PETSc is (re)initialized, the sparse matrices are constructed, the resulting matrix equation is solved, and PETSc is finalised. While our own time measurements show that it is the solving step that accounts for 99% of the computation time (of these combined steps), we still suspect that somehow storing the non-zero structure of the sparse matrices (which remains unchanged until the next mesh update) could help with performance. The reason we suspect this is because UFEMISM v1.0 was much faster in this regard, and the only significant change in this particular part of the code is the change from shared memory to distributed memory.
We will reflect these thoughts in the Discussion section of the manuscript.

> *This links to #2 of major issues. Firstly, I am missing the information what approximation applied allows one to run "basin-scale" (not sure what it means in terms of grid-sizes and spatial dimensions) on a laptop. Secondly, simulations on a laptop to me have a remote relevance to parallel performance/scaling on large machines, particular on distributed memory setups, which I understand this chapter to be about if the authors refer to "Large-scale practical applications".*

"Basin-scale" in this context means a single ice-sheet drainage basin. Apart from the ISMIP-HOM experiments, all simulations presented here are run with the DIVA. We will mention this in the manuscript.

Regarding the laptop vs. cluster question: we believe performance to be equally important in both settings. In our experience, a large fraction of a model developer's time is spent running relatively short, simple simulations on their laptop to debug and test new code. Only when that new code is judged mature enough do we move on to larger systems to run large-scale tests. However, already in the development phase, a significant fraction of our time is spent waiting for these simple test simulations to finish (which can take anywhere between a few seconds to a few hours, depending on the kind of work). Improving the model's performance, and thereby reducing this waiting time, would already be a big help to us. When we do eventually move on to 'application' simulations, about half of the time these are run on a laptop as well; only when the scale of the experiments (either in terms of resolution or duration, or in terms of number of simulations) becomes too large to be feasible run on a laptop do we move on to running on a cluster. Therefore, even for laptop settings, performance is important to us.

*This figure is the main reason for point #2 in the list of major critics. This graph, in my opinion, needs way more explanation and discussion – also in the text, not only the caption. Like in other parts in the text, it is missing information on the approximation to the Stokes equations that is being studied here. Secondly, the informed reader might want to know on which computational platform this was run on and if we look at a single- or multiple node run. I already mentioned that a flow-line model in my view is a non-representative example for scalability if one wants to get a picture on how the code would perform and scale being applied to full ice-sheet problems. Yet, this seems to be the only place in the manuscript where the reader can get an idea on a performance baseline in terms of solved time-steps/wall clock time. I already suggested to do scalability tests with MISMIP+ if not on the full Antarctic setup. Furthermore, I would like to learn more on how the authors determine and separate ice dynamics and non-ice-dynamics parts in this figure. From a pure computational science point of view, I interpret the fact that a run on 64 and even 128 cores consumes a comparable wall-clock time as a 2 core run points to a serious issue in the parallel implementation that in my opinion prohibits production runs on compute clusters.*

We agree that the text accompanying this figure needs to be more informative. In response to earlier comments, we will add the relevant information to the manuscript (platform, type of nodes, experimental set-up), plus some additional discussion regarding the poor scaling.

*Do the authors mean that there is some automatic parsing of the meta-data of the NetCDF file (CF convention?) that deduces the coordinate system? Further question: is UGRID format meant when referring to triangular meshes?*

The model parses the dimensions of the NetCDF file. If, for example, the file contains a dimension called "x" (or "X", "x-dimension", "x-coordinate", etc.) and a dimension called "y", it assumes the file contains data on a square grid, and tries to read it accordingly. If not, it looks for "lon" and "lat" (again with a set of alternatives for both) and if those are found, it assumes the file contains data on a global grid. Lastly, it looks for the set of dimensions used for the UFEMISM mesh format (which is not Ugrid; the option to output data in Ugrid format is planned for future work). We will mention this in the manuscript.

*To me, in the lower row in Fig 4 displays the 1st order results (not tremendously, yet visible) surface velocities to exceed these of the ensemble. I would ask to explain why this is the case and – since the authors do not seem to raise any concern in the text – why it can be neglected.*
*...*
*I could not find any mentioning in Pattyn et al. (2012) of a lateral circular symmetry to apply to MISMIP flowline setups.*
*...*
*Like in the caption of Figure 5 on page 16, the information on what approximation to the Stokes-equation has been used for this resolution-experiments is missing.*
*...*

*As mentioned in the major points of critics #3, I would ask the authors to relate parameter choice and the reduced time-span of the experiments to the original MISMIP protocol and explain – also in light of the argument of verification – this deviation.*
*…*
*Like in the MISMIP chapter, information on the applied approximation to run the MISMIP+ experiments seems to be missing, also in the caption of Fig. 5. Please add this information.*

See our previous answers to Major point of critique #3. All the requested information will be added to the manuscript.

*If this is about computational performance, I have to disagree. From this paper I am lacking information to really judge the computational performance of the code. Deducing from Fig. 2, I would even conclude that there is an unresolved issue what comes to scalability of the code. If it is about code verification, I previously mentioned that the MISMIP tests to me do not provide the means to deliver on that aspect, as they deviate from the original protocol which prohibits comparison, which leaves ISMIP-HOM (with some approximation showing strong deviations) and MISMIP+, where I could not deduce what approximation has been used for intercomparison with ensemble results.*

Our previous answers will see some additional information added to the manuscript to aid the reader in interpreting the results.

We agree that the poor computational performance, especially when moving to multiple nodes, is an unresolved issue. Since our budget for IT support has regrettably run its course, we do not foresee this being resolved in the near future.

*As mentioned earlier (point #2 of major points), in my view this paper is lacking information to really judge about performance of the code, as the reader is not provided with a baseline value. In my opinion, wording like "much faster" are not conveying enough information to the reader that would enable quantification of the code's performance. To really judge about performance, the reader would need to get an idea on a performance-baseline. For instance, how much the solution of one ISMIP 6 scenario run for Antarctic ice sheet using a particular approximation (preferably the optimal one, hence BPN) needs wall clock time on one, two or even more nodes of a computing cluster. From figure 2 I would draw the conclusion that the code does not scale beyond 8 cores (of whatever platform it was run on).*

It is difficult to come up with an experiment that would fairly compare the computational performance of v 1.0 and v2.0. v1.0 solved a simplified version of the SSA, where the gradients in the effective viscosity were neglected. This greatly improved the model's computational performance, but could significantly reduce the physical accuracy of the solution in certain cases (including, as it turned out, geometries with migrating grounding lines). We did, in the early stages of development, perform some preliminary experiments where we solved the same

simplified SSA in both v1.0 and v2.0, which is what the claim that v2.0 is "much faster" (which we agree is too vague) is based on. However, we regrettably did not save those experiments, and the option to use the simplified SSA has since been removed from v2.0 (due to its poor physical accuracy), so we cannot repeat them at this time. Additionally, v1.0 defined ice velocities on the grid edges (similar to an Arakawa-C grid), whereas v2.0 defines them on the triangle centers (similar to an Arakawa-B grid), which introduces an additional difference.

Accepting all these differences and simply comparing the total time for e.g. an ISMIP projection is difficult, because v1.0 lacks the modules for inverting for basal friction or sub-shelf melt, or for reading in external files describing those fields. Turning the problem around and taking an Antarctic experiment from v1.0 (which would do a non-nudged spin-up, use a greatly simplified ocean, and thereby result in a very different present-day ice geometry) would likewise be unfair, since the resulting geometry would be much 'smoother' and more stable than one resulting from a nudged spin-up, thereby artificially improving the model's stability and inflating its performance.

We will add a paragraph to the Discussion section reflecting these thoughts.

> *As I mentioned before, statements like this to me are impossible to evaluate without providing a baseline value. Just the fact that some algorithm takes 50 times longer (presumably using the same amount of computational resources) in my view does not imply that it is impossible to solve it – in particular if the code is claimed to run parallel (should in theory be able to use 50 times more resources, provided it scales) and comes with mesh-adaptation scheme.*

As mentioned in this paragraph, we do believe that using the BPA will become feasible if and when the scaling issue is resolved.

> *I wonder, should there be larger thickness changes in the active region, how does the model deal with the artificially imposed hydrostatic pressure gradients at boundaries with a one-sided fixed ice-thickness?*

If the thickness change in the active region would become large enough to significantly affect the surface slopes near the (artificially fixed) ice divide, then the assumption of a fixed divide would be invalid anyway. However, to be honest, we do not expect much use of this new option, as we believe the approach of simulating the entire ice sheet at a coarse resolution, and only the drainage basin one is interested in at a higher resolution (which UFEMISM can do), is much more practical.

> *To me this again lacks information to really get a clear picture on what type of simulations with what approximation and what accuracy can be achieved with what computational resources. There is only information on minimum resolution, but not the approximation used. Multi-day simulations are to me nothing that renders a computational problem impossible. But to me this sentence backs my suggestion that a*

*fusion of the two papers would be beneficial to the reader to pick information that I conclude exists in the other manuscript (Bernales et al., in prep) to be able to relate statements presented in this one.*

We hope the additional information provided in both our responses above and the revised manuscript satisfies the reviewer, as merging the two manuscripts is not a feasible option for us.

*Looking at the distances of the polygon-points at the zoom-in over Ronne-Filchner ice-shelf, I conclude that this is far away from the acclaimed 5 km resolution. If this is really the resolution to start from, I would ask the authors to include a sentence on how the accuracy of the coastline is increased when increasing the mesh density: is it just linearly interpolated between existing points or is additional geometrical information added to the polygon?*

The figure shows the coastline resulting from the BedMachine Antarctica v3 dataset at 40 km resolution, so using this to generate a 5-km mesh would not be useful. Choosing a higher resolution DEM would fill up the figure with dots, making it difficult to read. We will clarify this in the manuscript.

It is also important to note that a mesh, by itself, has no concept of 'bedrock elevation' or 'coastline'. When it is refined, it is simply presented with a set of line segments in the 2-D plane; where those segments originated does not matter. It is only when the data fields representing the Earth's geometry (bedrock elevation, sea surface elevation, and ice thickness) are projected onto the mesh, that the 'coastline' is defined. Setting up an ice model with a high-resolution coastline therefore requires 1) an input DEM with a suitably high resolution, 2) extracting the coastline from that DEM, 3) using that coastline as a refinement criterion for a mesh, and 4) projecting the DEM data fields to the resulting mesh. If the input DEM were to have a 40 km resolution, then we could theoretically use that to refine a mesh with a 5 km resolution around the coastline. Naively, we could then say that the resulting ice model resolves the coastline to 5 km, but the bedrock elevation itself in that region would be very smooth, as the input DEM did not have any information below the 40 km scale in the first place.

*This is a large figure. I would try to simplify the composition resulting now into entry e) and thereby reduce (entries a – e) the size of the figure. Instead, what I would welcome to see included is a visual demonstration on how the remeshing algorithm could enhance mesh densities in areas of large derivatives of the velocity field, which turned out to be essential to resolve (thermo-)dynamics of fast outlet glaciers (e.g., Zhao et al. 2018) in other applications.*

We agree that steps d and e are perhaps unnecessary; we will remove them.

UFEMISM currently does not include the option to use the velocity gradients as a mesh refinement criterion. However, the basic components to create this functionality are there. First you would need to define the 2-D polygons that envelop the regions where the strain rates

exceed a certain threshold (simply done with the existing calc_mesh_contours routine), and then simply use those polygons as refinement criteria (Fig. A2 only illustrates refinement around 0-D points and 1-D lines, but 2-D polygons are supported too; these are currently used to define resolutions over individual ice drainage basins).

> *This whole part appears abstract to me and without looking things up in Syrakos et al. (2017), difficult to understand. I would even suggest that, equally, a simple reference to the paper above and removing whole part B would shorten the manuscript. If the authors want to keep it, in my opinion it would help to have some figure of local grid configurations annotated with the node-indices and showing the most important features, like distances ($\Delta x_j$, $\Delta y_j$) and value entries ($f_{a_j}$) to better help illustrating the formulation of the stencils as presented in this appendix, also, with respect to the different discussed mesh-types and the coordination numbers of nodes/variables therein.*

Our approach expands upon the work by Syrakos et al. (2017) by adding the second-order derivatives, and by including the derivation for a staggered grid, so that a reference to that paper would not be sufficient.

We will add an illustration to help the reader with the different geometrical quantities.

> *In my view, it would be beneficial to the reader to explain what the third case "otherwise" means. With respect to definition in the text that indices $j \in [1, n]$ indicate all neighbours of i, I must miss something by interpreting that neighbours are connected by definition and "otherwise" being irrelevant for A-grids. Like mentioned before, some graphical display of a local mesh arrangement in my opinion could aid the understanding and prevent misinterpretations.*

There was a typo in Eq.s B9, B10, and B15-17. In all these equations, the subscripts in the second case (if j is connected to i) should be j, not c. Together with the new figure illustrating the local mesh geometry, this fix should clear up the confusion.

> *As remeshing/-mapping seems to be one of the main new features of UFEMISM v2.0, I would suggest to move this part (at least the non-mathematical) into the main section of the paper and rather take out other parts from there (I already suggested earlier)*

We believe the main body of the manuscript should focus on the ice dynamics and user experience of the model, and that the underlying math can remain in the appendix.

> *Like before, I find this section relatively abstract and difficult to read. Similar as before, I would be of the opinion that some graphics on the mesh-to-mesh projections showing the domain, its boundary and the two (source and destination) meshes to get a picture what this is about.*

*Some readers that have not dealt with dual graphs might not know what a Voronoi cell is. Displaying this in a graph (see point above) and defining it in the text, in my view, would improve the readability of this chapter.*

We will add an illustration of the geometry of the two meshes to aid the reader.

**List of less important issues**

*I guess the authors want to express that the code can now be run on distributed memory architecture (referring to architecture as being the hardware, rather than the program itself)? If referring to code, I would suggest to change to: ... from a shared- to a distributed memory implementation.*

Accepted.

*In my opinion, a statement like this would demand references to be included.*

Although we definitely remember seeing this approach in several papers (and conference talks, etc.) we cannot seem to find the references. We will adjust the statement in the manuscript.

*The brackets show the vertical derivatives of the components constituting the horizontal strain rates (and not membrane stresses).*

We will change the phrasing.

*page 11 – line 297: processors - I would use cores*

Accepted.

*One group of experiments does include flat beds (C,D,F), the other (A,B - as correctly pointed out in the sentence to follow) compute on an undulated bed and Experiment E on a glacier flowline (which is not flat, either).*

We will change the phrasing.

*Since remapping is mentioned here, I would suggest to make a reference to Appendix C for the reader's convenience to quickly look things up.*

Accepted.

*nV at this stage appears to me as being undefined (guess, it relates to some correlation number)*

nV is the number of vertices in the mesh; we will mention this in the manuscript.

*In all equations the lower symmetry entries in the matrix are omitted. Despite the redundancy, in my view it is preferable to either spell things out in the equation or mention the unusual notation in the text.*

We have encountered this notation regularly.

*Just a suggestions, but one could elegantly use the Kronecker delta to define $D_{ij} = \delta_{ij} f_i$ to achieve a better typesetting result in that line.*

Accepted.

---

## Author Comment (AC2)

We thank the reviewer for their constructive criticism of our manuscript, and would hereby like to respond to their concerns. Their comments are shown in italics, our response in regular type.

**General assessment**

*I would encourage the authors to better motivate the acceptance level with respect to the Stokes or higher-order ensemble results (why is 20km still accepted, but 10km is not?)*

*L145: "DIVA produces velocities that agree well with the Stokes solution down to horizontal scales for basal topographical features of about 20 km" How can "agree well" be quantified? In the mentioned references, the solution for 20km is not within the higher order or Stokes ensemble.*

*L368: "The DIVA remains accurate to spatial scales of about 20 km,..."*
*Gain, please define "accurate". What deviation from the FS ensemble is accepted? Why is the 20km still accepted but 10km not?*

The relative errors that are deemed acceptable are, ultimately, a subjective judgment. In ISMIP-HOM Experiment A with L = 20 km, the DIVA overestimates the surface velocity of the fast-flowing ice by about 25 %. In the experiment with L = 10 km, this number increases to about 40 %. Based on the inter-model spread of intercomparison exercises such as ISMIP6, and also considering the fact that ISMIP-HOM presents an extreme case (with quite dramatic subglacial topography), we found a 25 % deviation to be large but workable. Of course, so long as running simulations with the BPA is not computationally feasible, the DIVA remains the lesser of two evils, providing the least inaccurate solution (compared to the hybrid SIA/SSA, and to several other depth-integrated approximations, according to Robinson et al. (2021). We will add a few lines to the manuscript to reflect these thoughts.

*However, DIVA can be thought of a modified SSA approximation (considering membrane stresses in the plane), in which the effective viscosity is treated in a different way (also accounting for vertical velocity gradients). I suspect that the non-linearly diffusive component of the ice sheet flow (e.g. Bueler et al., 2007), which is relevant in many (purely shear-stress-driven) parts of the ice sheets, may be underrepresented in DIVA.*

*L164: Maybe it is worth mentioning the differences and similarity of SSA to DIVA.*

*L186: "but starts to deviate significantly from the Stokes solution earlier than the DIVA as the length scale decreases (Berends et al., 2022; this study).*
*"In the ISMIP-HOM experiment". Generally SIA describes a (non-linear) diffusive process that is characteristic for large parts of the ice sheet flow and not represented in SSA, and likely not (or only limited) in DIVA. Please refer to the respective section in this study.*

It can be shown that, in the absence of horizontal strain rates (so pure shear flow over a flat plane), the DIVA is identical to the SIA. We have at some point run the EISMINT-1 experiments (an idealised, roughly Greenland-sized ice sheet lying on a flat plane, achieving steady state through a simple spatially variable mass balance) with the DIVA instead of the SIA; the resulting ice sheet is nearly identical to that resulting from the SIA model (a few meters thickness difference near the ice divide and near the margin). ISMIP-HOM Experiment A has a no-slip condition at the base and so too is dominated by vertical shear, and there too the DIVA performs well. We will clarify this in the description of the DIVA in Sect. 2.2.2.

Regarding the SSA: the similarity between the linearised equations (Eq. 6 and Eq. 9) is actually a great benefit in practice, as it makes it very easy to adapt SSA models to be able to solve the DIVA. We will mention this in the manuscript.

> *Maybe, benefits of DIVA over SIA/SSA will become clearer in real world (or regional) application, as planned for the mentioned follow-up study.*

We have no such experiments planned for the Part II paper, which will instead focus mostly on model initialisation, nudging of basal friction, melt, and calving, and on comparing to Greenland & Antarctic retreat model ensembles (e.g. ISMIP6).

> *I am also missing in the description how (lateral/marginal) boundary conditions have been treated.*

> *L135ff: What about the lateral boundaries (margins, calving fronts)?*

UFEMISM uses the "infinite slab" approach, where the momentum balance is solved even in the ice-free part of the domain (by assigning a small – 10 cm – ice thickness to those cells), and applying a simple Neumann boundary condition to the domain boundary. We will mention this in the manuscript.

> *Apart from the DIVA solver, some model components (e.g. thermodynamics validated with EISMINT) have already been described in the v1.0 paper by Berends et al., 2021. It would be good to mention more clearly what has not changed since v1.0.*

The list of things that have not changed is quite extensive, but consists largely of model components that are not used in the idealised experiments described here (surface mass balance, climate forcing, GIA, etc.), and which were also not described in the v1.0 paper (as they were mostly adapted from earlier models). We agree that thermodynamics is an important one since it is so closely related to the ice dynamics itself, so we will mention it in the manuscript.

> *Also in the v1.0 paper, the same MISMIP-inspired experiment has been performed, but with a SSA/SIA hybrid stress balance, flux correction, an explicit first-order finite volume upwind scheme for the ice thickness evolution and much coarser resolution. Readers may*

*be interested in a more quantitative comparison of the new model version compared to the older version, or to other similar models (e.g. flow-line MISMIP or MISMIP2d).*

A direct comparison to v1.0 is difficult because of the many changes in the model. The change from the flux condition to the sub-grid friction scaling, the change in time stepping, and the change in resolution, all can be viewed as "mathematical", but there have also been some "physical" changes. One in particular (which was perhaps not immediately obvious from the manuscript, so we will state this explicitly in the revised version) is that v1.0 solved a simplified version of the SSA, where the gradients of the effective viscosity were neglected. Back then, we based this on (much) earlier work, but we later performed some experiments that indicated that this approach, while making the model substantially more numerically stable and thereby faster, introduced significant errors in the velocity solution, particularly in geometries with migrating grounding lines. We have therefore removed this simplification in v2.0 (which necessitated the change from defining the velocities on the grid edges, to the triangle centres, in order to achieve a numerically stable solution).

Since v1.0 has not been used in any practical applications yet (as our own focus has been more on developing v2.0), and there are therefore no earlier results that can be retroactively assessed, we do not think there is much added value in such a detailed comparison.

We also do not think performing the flowline MISMIP or MISMIP2d experiments would be of much added value. The two main model properties that these experiments investigate, namely (the lack of) path-dependency in the grounding-line position and the rate of grounding-line advance/retreat, are already assessed in the experiments we already did (path-dependency in the modified, plan-view MISMIP experiments, and retreat rate in MISMIP+).

> *The authors also mention that they switched for the discretization from neighbor functions to a least squares-based scheme. The v1.0 paper already uses an averaged-gradient (numerical stencils) approach similar to an unweighted least-squares approach (Syrakos et al., 2017). Is this the same as in v1.0?*

In the v1.0 paper, we compared the numerical convergence of the neighbour functions approach (which could indeed be described as an averaged-gradient approach, and which is what was implemented in v1.0) to that of the least-squares approach by Syrakos et al. (2017), finding that they produced very similar results. Because of the change in the SSA/DIVA discretisation mentioned before (no longer neglecting gradients in the effective viscosity), and the subsequent change in definition of the ice velocities (triangles instead of edges), v2.0 makes a lot more use of staggering than did v1.0. As the least-squares approach can be more easily generalised to staggered grids (and also has a much more elegant derivation), we opted to remove the neighbour functions approach entirely and use the least-squares approach instead. We will clarify this in the manuscript.

> *I tried to install UFEMISM (without nix), but failed for the PETSc dependency, while searching for help in an installation manual or README.*

We are aware that it can be tricky getting the different libraries and compilers to work on different platforms. Unfortunately, since our funding for IT support from the eScience Centre (in the person of co-author V. Azizi) has expired, we do not foresee significant improvements in this regard in the near future.

**Detailed comments**

*L10: "... irreducible uncertainty in many of these processes..." Not sure what this means.*

For many of these physical processes, there is a limit to how accurately we can predict their future behaviour, based on currently available observations. Typically, this limit is determined experimentally by sampling the phase space of possible values of the parameters governing these processes, which is where the models we mentioned appear. We agree that describing this uncertainty as "irreducible" might be confusing; we will remove this word.

*L43: "...have recently directed their efforts at creating new, more powerful ice-sheet models (e.g. Pattyn, 2017; Hoffman et al., 2018; Quiquet et al., 2018; Lipscomb et al., 2019; Robinson et al., 2020; Berends et al., 2022)" I am not sure if all models on this list had the intention to become "more powerful", as some are used for improving process understanding etc.*

We agree this was probably not their only intention, but we have deliberately referenced models that have included high-performance computing and/or user-friendliness in their model design, so we believe this phrasing to be appropriate here.

*L47: "Version 1.0 (Berends et al., 2021) was the second ice-sheet model to use a dynamic adaptive mesh..." There should be previous experience with AMR, e.g. dos Santos, 2019 for ISSM or Gladstone et al., 2010a, just to give two examples. Better rephrase or provide a complete list.*

We will add references to Todd et al. (2018) for Elmer/ice, Durand et al. (2009), Gladstone et al. (2010), and to dos Santos et al. (2019, for ISSM) to the manuscript, and change the phrasing accordingly.

*L62: "Part 2, which is submitted for review and publication separately (Bernales et al, in prep.)..." It would have been great to had both as companion.*

We agree this would have been great. Unfortunately, the manuscript of Part II has been delayed, as two of the authors have had to switch contracts in the meantime (a consequence of the failing system of scientific funding in the Western world).

*L98: "...with no significant loss of accuracy"*

*Hard to find quantitative numbers here or in Berends et al. (2021). If this remapping step is performed regularly also a small information loss (diffusion) can accumulate.*

In the v1.0 manuscript, this was studied in the Halfar and Bueler dome experiments, where we could compare to an analytical, time-dependent solution. Both these experiments involved dozens of remeshes (over a hundred in the higher-resolution cases), and the solution was not visibly affected. In the current manuscript, probably the most solid evidence comes from the MISMIP+ experiment. In the 500 m resolution experiment, the mesh was updated about a hundred times, and yet the solution stays well within the model ensemble. We suspect that, since the flow of ice generally already has a substantial diffusive term, the small amount of numerical diffusion added by the remapping does not have a large effect (although in the MISMIP+ experiment, it might be possible that the solution is somewhat affected, which could explain why the grounding-line positions in the 1 km, 750 m and 500 m experiments differ more than expected).

We will add a few lines to the manuscript to reflect these thoughts.

*L228: "… flux condition has been replaced by a sub-grid friction scaling scheme, following the approach used in PISM (Feldmann et al., 2014),".*
*Is this really a "replacement", as both methods may have different effects in controlling grounding line flux? I think the linear GL interpolation goes back to Gladstone et al., 2010b, and then there are 2D extension (bi-linear), e.g. Feldmann et al., 2014. Would it be possible to align the mesh with the grounding line, such that meshes are either grounded or floating/icefree?*

We will add a few lines to briefly state the (dis)advantages of the two approaches.

While aligning the mesh with the grounding line is technically possible, it would imply remeshing the model in every time step. Not only would this be computationally very expensive, but in that case, our earlier claim that the numerical diffusion resulting from the remapping is not a problem, would likely not hold anymore.

*L247: " an implicit scheme"*
*The is an interesting paper which may be cited as well (Bueler, 2023). Is the (semi-) implicit scheme still mass conservative (as it should be as a natural property of finite volume models)?*

This is indeed a very interesting paper, we thank the reviewer for pointing it out and will definitely cite it.

The implicit scheme is mass-conserving within machine precision. The semi-implicit scheme introduces errors that are larger than can be explained by machine precision, although still very small (if I recall, around 10^-12 relative error).

*L284: Eq. 33: I understand the notation is consistent with Robinson et al. 2022 and Cheng et al. 2017, but τ is often associated with stresses or time scales. Better use another epsilon consistent with Eq. 34. From Eq. 33 it seems that τ has dimensions of m/s. How does this fit with the default value for ε =3 m in L287? I would have expected it to be dimensionless.*

We understand the possible confusion, but we much prefer to keep the notation consistent with the existing literature.

We had not noticed the mismatching dimensions before. Neither Cheng et al. (2017) nor Robinson et al. (2020) mention the units of the tolerance epsilon. As tau should indeed have units of m/s (or m/yr in UFEMISM), this suggests epsilon should too (which actually seems more plausible than simply m, as m/yr implies a tolerance in the thinning rate). We will mention this in the manuscript.

*L293ff: "memory chips" and "processors"*
*For HPC architectures terms like "multi-core CPU nodes" are used, and the standard is rather 2x64 cores per node.*

*L299: "32 processors"*
*Do you mean CPU cores here, as mentioned in the Fig. 2 caption?*

We will change "processors" to "cores" throughout the manuscript and clarify the statement regarding the typical number of processors per node.

*L299: "...often accounting for more than 80 % of the total computation time of a simulation."*
*This very much depends on the application and the size of the computational domain. For instance, I/O can be a bottleneck for high-resolution application.*

In UFEMISM, even in relatively low-resolution simulations, solving the momentum balance accounts for > 80 % of computation time, even when using the DIVA (which is the fastest to solve). I/O has never been a significant contributor in our experiments.

*L307: "..., which could be improved by paying more attention to the way the model domain is partitioned over the processes, and the way PETSc decides which data should be communicated."*
*In deed, domain decomposition as well as matrix factorization and preconditioning have quite some potential for performance speed-up. Berends et al., 2021 describes in v1.0 already a load- balanced processor domain decomposition, which would be worth the refer to.*

We suspect the scaling problems lies with the way we have implemented PETSc within UFEMISM. Mesh partitioning is currently done the same way as in v1.0. Where we suspect the

current performance problem lies, is with how PETSc determines (or is told) the connectivity between the vertices, and by implication, the non-zero structure of the sparse matrices it must work with. In the current implementation, in every non-linear viscosity iteration (where the sparse matrix equation must be solved), PETSc is (re)initialized, the sparse matrices are constructed, the resulting matrix equation is solved, and PETSc is finalised. While our own time measurements show that it is the solving step that accounts for 99% of the computation time (of these combined steps), we still suspect that somehow storing the non-zero structure of the sparse matrices (which remains unchanged until the next mesh update) could help with performance. The reason we suspect this is because UFEMISM v1.0 was much faster in this regard, and the only significant change in this particular part of the code is the change from shared memory to distributed memory.

We will reflect these thoughts in the Discussion section of the manuscript.

> *L322: "appropriate remapping function"*
> *You could already refer to the kind of remapping (bilinear, conservative) in Sect. ?. What input files not covering all of the computational domain, are there missing values or extrapolation applied?*

"Appropriate" here means the function appropriate for the type of source grid (x/y, lon/lat, unstructured), not the kind of remapping. Currently, nearest-neighbour extrapolation is used for input files not covering the entire model domain (although routines for assigning a user-defined missing value or doing e.g. linear or Gaussian extrapolation exist and should be easy to integrate here. We will mention this in the manuscript.

> *L329: "full list of the 100+ fields.."*
> *Does fields" imply that only 2D variables are available as diagnostic output? I guess for the current size of the applications, parallel I/O is not yet considered?*

3-D fields of e.g. velocity, temperature, and (effective) viscosity are available as well. Additionally, UFEMISM generates a NetCDF output file with domain-integrated values of e.g. mass balance components, ice volume, etc. All of this is indeed done serially, as we have not yet encountered any significant computational load from this. We will mention this in the manuscript.

> *L338: "The UFEMISM Github repository also features integration with the nix package manager..."*
> *Does also imply also other options? I found an EasyBuild example in the "templates" folder in the Git repo. Is there a general installation manual or a website with instructions? What open source license is used, I found in the "CITATIONS.cff" the entry for "Apache-2.0"?*

The nix package manager option was added as a convenience to have a reproducible build of UFEMISM2.0 and a canonical way to compile the code. However, since the nix package manager is not widely used, the main way to install UFEMISM2.0 is by using make. UFEMISM2.0 has

minimal and easily installable external library dependencies (petsc, netcdf). We will add this information in the README of the GitHub repository and also update the CITATIONS.cff with the correct license.

> *L365: "In experiment C (Fig. 4), which concerns sliding over a bed with spatially varying roughness, all three approximations result in velocities that agree well with the ensemble."*
> *Please define "agree well", what is the acceptance range with respect to the FS ensemble? SIA/SSA seems to perform better than DIVA and BPA , why?*

We will change this phrasing to clarify that all results lie within the ensemble range except the BPA, which overestimates the full Stokes solution by up to 13 %.

We have spent a considerable amount of time looking into the difference between UFEMISM's BPA results, and that of the Pattyn et al. (2008) higher-order ensemble. While we cannot be sure, we think there is the possibility that UFEMISM's solution slightly more accurate than that of the ensemble. We find that, in the small-wavelength versions of Experiment C, the non-linear viscosity iteration converges extremely slowly compared to the other experiments, particularly when using the BPA (the SIA/SSA and the DIVA show the same behaviour, but to a lesser extent). Using a tolerance of e.g. a difference between subsequent velocity iterations of 1e-7 m/yr (which is very small; for most other experiments, values of about 1e-4 m/yr are sufficient) still results in a significantly different solution from using an even smaller value of 1e-9 m/yr. If the other models in the ensemble used a stop criterion for this iteration that was not strict enough, it is possible that their modelled velocities are too low, as the non-linear iteration terminated before having converged. This could also explain why the full Stokes ensemble shows higher velocities than the higher-order ensemble (as, according to Pattyn et al., the full Stokes models generally use pre-existing, general-purpose finite element packages, while the higher-order models more often used code written by the researchers themselves). As the Pattyn et al. ensemble was published one-and-a-half decades ago, there is no convenient way to check this. Other than this, the only plausible explanation is a coding error on our side – which is, of course, always possible.
We will leave it up to the editor if this consideration should be written down in the manuscript.

> *L372: "...performed an experiment along the lines of the Marine Ice-Sheet Intercomparison Project (MISMIP; Pattyn et al., 2012). The experiment describes a circular, cone-shaped island, ..."*
> *Please, better motivate how the experiment deviated from the original MISMIP (and from Berends et al., 2021, Sect. 3.3) I assume that it also covers two-dimensional aspects of the stress balance and geometry evolution. However, this prohibits a comparison to the semi-analytical solution provided for the flowline SSA case. The authors should indicate that they only considered the experiment with downward-sloping bed (EXP1+2, without overdeepening EXP3).*

The reason we have opted to extrude the 1-D geometry radially, rather than transforming the original 1-D flowline into a 2-D flowband, is that, while this means the resulting grounding-line position no longer matches the (semi-)analytical solution provided by Pattyn et al. (2012), it offers the advantage of checking the full 2-D stress balance (instead of only the x-component). This particularly allows us to check the symmetry of the grounding line. A well-known (but, as far as we are aware, never published) issue with flux condition schemes in square-grid models is the "octagonal" grounding line. A similar undesirable dependency on the grid geometry could sometimes be seen in UFEMISM v1.0 (which used a flux condition scheme), but has since been fixed with the introduction of the sub-grid friction scaling scheme in v2.0. We will mention this in the manuscript.

> L378: "grounding-line hysteresis"
> In order to not confuse the reader here, the authors should name this effect "numerical path- dependency" or similar, as this has nothing to do with the (intrinsic) hysteresis associated with EXP3 in MISMIP.

Accepted.

> L405: "using the Schoof sliding law"
> How is this choice motivated?

We chose the Schoof sliding law because, of the three options offered by the MISMIP+ protocol, we find that this one results in the best numerical stability in our model. We will mention this in the manuscript.

> L425: "...in a fraction of the required computation time, or running a simulation in the same amount of time as before, but at a much higher resolution."
> I encourage the authors to provide some rough numbers, in comparison to v1.0 and maybe to other similar models? PISM with 4km resolution in Antarctica (1521 x 1521 x 221) would need for one model year about 10 wall clock minutes on one 2x64 CPU node, with 16km and 64 CPU cores (one socket) it would be about 30 wall clock hours (from Albrecht et al., in review).

Our earlier response explains the difficulty in directly comparing the performance of v1.0 and v2.0. We will add some more detail about the "basin-scale" experiments mentioned in the manuscript, which consist of 20,000-year pan-Antarctic experiments with a 4-km grounding-line resolution in the Pine Island and Thwaites basins, which can be run in ~24 hours on a 2-core Macbook Pro M2 2023.

> L465: "The polygon-based routine can be used to increase the mesh resolution over a certain ice- sheet section, e.g. the Pine Island Glacier drainage basin. This is illustrated in Fig. A2."
> Where is the polygon-based refinement in PIG illustrated in Fig. A2?

The polygon-based refinement is not illustrated; we will clarify this in the manuscript.

> L674: "The number of vertical layers is configurable, and is by default set to 12."
> But this would mean for 3 km thick ice more than 600m at the top and about 80m at the base? In order to resolve temperate ice at the base, this is quite coarse (Kleiner et al., 2015).

This is correct.

> L714: "matrix whose coefficients depend on the mesh geometry and the ice velocities"
> Is Lij in Eq. F7 related to mesh geometry?

It is, see line 874 (right after Eq. F4): "L^ij is the length of their shared Voronoi cell boundary"

> L8: "...man-made climate-change-caused mass loss...", also L26 better say "human-made" or "anthropogenic"

Accepted.

> L18: "The version control system.." you mean git? Not mentioned before.

We will state that this is indeed git.

> L20: "The i/o...", also L55, L334 Define and better write capital I/O.

Accepted.

> L23: "earth" Earth

Accepted.

> L29: comma before "range"

Accepted.

> L56: "It also includes a version control system that includes..." uses

We do not think this suggestion makes the sentence more readable.

> L68: " It solves an approximation of the Stokes equations..." "different approximations", or "for approximations"

Accepted.

*L94: "…thus defeating the purpose of the adaptive mesh."*
*Sounds a bit harsh, maybe use "offsetting the benefits of the adaptive mesh"?*

Accepted.

*L100: "… adapted into computer code…" "the way it has been implemented"*

Accepted.

*L104: "The most complete is the Blatter-Pattyn approximation…"*
*Maybe mention L109 earlier, "can all be derived by neglecting increasingly more terms in the Stokes equation"*

Accepted.

*L180: Eq. 13 and Eq. 20 use the horizontal nabla operator, should be introduced at some point, maybe in Table 1? It is defined later in Eq. E2 in the appendix.*

We believe the nabla operator to be such a basic mathematical tool that we can trust the majority of readers to be familiar with it. The reason we mention its use explicitly in Appendix E is to clarify its two-dimensional (rather than three-dimensional) use in that context.

*L216: "… (which is not the square of the basal friction coefficient β, but a confusingly named separate entity, which we maintain for the sake of consistency with earlier literature) for the Weertman-type part."*
*Then better use β\* or some different index (e.g. T).*

We believe the confusion arising from this terminology is to be preferred over the confusion of having different letters for the same variable across different papers.

*L264: Better not start a sentence with an abbreviation.*

Accepted.

*L319: "square grids"*
*Do you mean a regular "Cartesian grids"? Can the domain be also rectangular, or is a square with Lx = Ly required?*

We do indeed mean Cartesian grids; we will mention this in the manuscript. The domain can also be rectangular, but the code does require that $\Delta x = \Delta y$ (if this is not the case, it will throw an error).

*L321: "…UFEMISM will automatically detect the type of grid from the dimensions of the NetCDF file"*

*Does this mean, the model checks if lon/lat or x/y are used as dimensions? Or do you also use projection parameters from the metadata in the HDF5 headers, proj string?*

It does indeed check only the names of the dimensions (though it accepts multiple variants, e.g. 'x', 'X', 'x-coordinate', etc.). Projection parameters are not read; it assumes the provided grid has the same projection as UFEMISM (i.e. the ISMIP standard projections for Greenland and Antarctica). We will mention this in the manuscript.

*L324: "The sparse matrices representing the remapping operators..." This is often called remapping "weights" (e.g. YAC based on CDO).*

We will state this widespread alternative name in the manuscript.

*L332: "Github Actions"*
*Better use "GitHub" with capital "H", and refer to the website for the GitHub Actions software development framework (https://docs.github.com/en/actions).*

Accepted.

*L417: " verified its performance"*
*What does this mean? I would associate "performance" with numerical efficiency.*

We will change this to "verified it in a number of different benchmark experiments"

*L436: "define regions where the ice thickness should not change."*
*For instance, along flow divides. What about defined boundary velocities?*

Currently this is not possible, although the basic tools are there. The velocity solver has as optional input arguments a mask defining where velocities are defined, and the accompanying velocity field; right now, the field is simply zero everywhere, but it would be trivially easy to e.g. read a NetCDF with satellite-derived velocities, and provide those to the solver instead.

*L488: " nV-by-nV matrix" with V the number of vertices?*

nV is the number of vertices in the mesh; we will mention this in the manuscript.

*Fig. 1: Maybe provide a scale as measure, or mention the length of the domain for reference. What are the used distance measures here, mentioned later as config variables?*

The caption states that the domain covers the entire Antarctic continent, as is clearly visible in the figure. The caption also already states the size of the triangles in km.

*Fig. 2: I would assume you used the default DIVA? What kind of architecture did you use, CPU nodes with 2x32 cores? Hence, 128 cores would be associated with 2 CPU nodes using inter communication?*

We have will add some additional information about the preliminary scaling tests, concerning both the experimental set-up and the system the simulations were run on. The tests were performed on the Snellius supercomputer, on 128-core nodes. We have also added an additional test with 256 cores to the plot

*Figs. 3+4: In the labels, I would expect the FS/HO mean to be the line, and in transparent the ensemble range?*

Correct, the legends have the entries for the ensemble mean and ensemble range reversed. We will fix this.

*Fig. 5 How does this experiment differ from Berends et al., 2021, Fig10?*

Berends et al. (2021) used the hybrid SIA/SSA instead of the DIVA, the flux condition instead of the sub-grid friction scaling, and substantially coarser resolutions. We will mention this in the manuscript.

*Fig. 6: The dashed lines in panel a are hard to distinguish, e.g. where is the red 5km solution? The solutions of MISMIP+ ice1r seem to converge for increasing resolution against a solution above the ensemble mean from Cornford et al., 2020. I am assuming the authors used the DIVA stress balance approximation here? This is interesting, as the 750m solution seems to be an exceptions of this convergence? What could be the reason for this resolution dependence. Fig. 11b in Cornford et al., 2020 suggests that higher resolution may provide solutions below the mean, same for the HO contributions in Fig. 9b in Cornford et al., 2020.*

The lines in panel A are indeed hard to distinguish because the solutions are very close to each other. We will mention that we did indeed use the DIVA for this experiment.

We suspect it could be possible that the solutions in the higher-resolution simulations (1 km, 750 and 500 m) are starting to show some accumulating diffusion from the many remeshes (which happened more than a hundred times in the 500 m simulation). Possibly this could be solved by setting a wider band around the grounding line where the high resolution should be applied to the mesh generation, which would reduce the frequency of the mesh updates. We will mention this in the manuscript.

*Fig. A2: I personally like the humor in this figure but 3 rows would be sufficient to show how line-based, point-based and polygon-based routines work. Also, the refinement seems to be more dense in the half-circle case compared to the circle? I guess this refers*

*to the different config parameter named in L476? If yes, this information would be helpful in the figure caption.*

We agree that steps d and e, while humourous, are perhaps unnecessary; we will remove them. We will also mention that, indeed, the half-circle was given a higher resolution than the circle.

---

## Referee Report (RR1)

**Review of The Utrecht Finite Volume Ice-Sheet Model (UFEMISM version 2.0) – part 1: description and idealised experiments**

**by Constantijn J. Berends, Victor Azizi, Jorge A. Bernales and Roderik S. W. van de Wal**

**Summary:**

The revised manuscript by Berends et al. addresses most of the points raised by the reviewers. The authors added many technical details (e.g on MPI in Sect. 3.1 or on I/O in Sect 3.2), the recommended references, additional explanations for the experimental design (e.g. on MISMIP 2D extension in Sect 4.2) and more detailed discussion on the limitations of the model (e.g. the frequent mesh update induced numerical diffusion in Sect 4.3 or the weak scalability for inter-nodal applications in the discussion Sect. 5), which seems to be subject of future work.
For instance, the authors added a new section 2.5 on energy conservation (l 304ff) based on model v1.0, a paragraph on ice front boundary conditions (l. 166ff) and similarities of DIVA with SIA/SSA (l. 197ff and l. 220ff), as well as a paragraph on elaborating the difficulties in comparing the two model versions (l. 619ff). In the appendix also some figures (Figs. B1 and C1) were added to better explain the discretization and remapping schemes.

**General assessment:**

While the authors address most reviewer concerns in the response and the revised manuscript, the additional speculations and discussions are still based on the original set of experiments. I was hoping that some additional test simulations were possible to quantify the effects of the size of the problem (see below) or the mesh refinement frequency on scalability (Sect. 3.1) or of a wider mesh around the grounding line (Sect 4.3).

The authors now emphasize that DIVA is the default stress balance approximation used in the experiments (and not BPA). As outlined earlier in the review, there is some fundamental critique in parts of the ice model community on the applicabilty of DIVA in general, and it is therefore good to gain some more experience. However, the authors mention in their response, that "validating the equations themselves" would not be the focus of their manuscript.

**Detailed comments:**

l. 166: "UFEMISM currently does not include a stress boundary condition at the ice front for any of the momentum balance approximations. Instead, it uses the "infinite slab" approach…"

Doesn't this imply numerical diffusion of the front? And if yes, how would you then define a calving front (line segment) in the model, e.g. for calving experiments?

l. 424: "Another contributing factor could be that the model set-up used for the scaling test was too 'small' (i.e. had too few vertices), so that the communication latencies between cores begin to dominate the total computation time."

This would be worth a few test simulations, not over the full length.

**Typos and recommendations:**

l 141:  Antarctic ice sheet →  Antarctic Ice Sheet (also l 427)
l. 198. we used  DIVA
l. 468: extrapolation→ extrapolated
l. 485:  "perfect restart" → this would be a prerequisite of reproducibility from (intermediate) restart states
Fig C1: panel C may not be needed

---

## Author Response (AR2)

We thank the reviewer for their constructive criticism of our manuscript, and would hereby like to respond to their concerns. Their comments are shown in italics, our response in regular type.

**General assessment**

While the authors address most reviewer concerns in the response and the revised manuscript, the additional speculations and discussions are still based on the original set of experiments. I was hoping that some additional test simulations were possible to quantify the effects of the size of the problem (see below) or the mesh refinement frequency on scalability (Sect. 3.1) or of a wider mesh around the grounding line (Sect 4.3).

We have performed an additional set of simulations of the scaling experiment (i.e. the spinup phase of the modified MISMIP experiment), this time with a grounding-line resolution of 2 km instead of 8 km. The poor scaling persists, suggesting that there is indeed an underlying problem with the parallelisation. As we currently do not have funding for IT support anymore, investigating this issue further will be reserved for future work. We will mention this in the manuscript.

We have re-done the MISMIP+ ice1r simulations with a wider high-resolution band around the grounding line for the high-resolution (

**Fig. 1: results of the MISMIP+ ice1r experiment with the same flow factor for every resolution.**

The authors now emphasize that DIVA is the default stress balance approximation used in the experiments (and not BPA). As outlined earlier in the review, there is some fundamental critique in parts of the ice model community on the applicability of DIVA in general, and it is therefore good to gain some more experience. However, the authors mention in their response, that "validating the equations themselves" would not be the focus of their manuscript.

We maintain that a fundamental study of the relative merits of different approximations to the Stokes equations, is beyond the scope of this study.

*l.* 166: "UFEMISM currently does not include a stress boundary condition at the ice front for any of the momentum balance approximations. Instead, it uses the "infinite slab" approach..."

Doesn't this imply numerical diffusion of the front? And if yes, how would you then define a calving front (line segment) in the model, e.g. for calving experiments?

While we have not performed such experiments with UFEMISM, earlier experiments with our older model IMAU-ICE showed that using the infinite-slab approach resulted in very small differences in the modelled velocities, compared to using a stress boundary condition at the calving front. We will mention this in the manuscript.

Although the conservation-of-momentum solver uses the infinite-slab approach (by 'imagining' a very thin – 10 cm – ice shelf in areas that in reality are open ocean), the conservation-of-mass solver does see the difference between ice-covered and ice-free vertices. The calving front can then be identified per edge, and calving can be applied there. We have contributed simulations with UFEMISM to the ongoing CalvingMIP project, and our results seem to agree well with the other models' submissions so far.

*l.* 424: "Another contributing factor could be that the model set-up used for the scaling test was too 'small' (i.e. had too few vertices), so that the communication latencies between cores begin to dominate the total computation time." This would be worth a few test simulations, not over the full length.

See above.

l 141: Antarctic ice sheet → Antarctic Ice Sheet (also l 427)

Fixed.

l. 198. we used to DIVA

Fixed.

*l.* 468: extrapolation → extrapolated

Fixed.

l. 485: "perfect restart"  $\rightarrow$  this would be a prerequisite of reproducibility from (intermediate) restart states

Yes.

Fig C1: panel C may not be needed

We will remove panel C.

---

## Author Response (AR3)

Dear Philippe,

Thank you for the final corrections. I have changed the colour schemes of the figures (following the suggested set of colourmaps from the Copernicus submission guidelines), that should be sorted now. I have also changed put the DOI of the zenodo archive with the model code directly in the text.

Kind regards,
Tijn Berends